# Reducing information dependency does not cause training data privacy. Adversarially non-robust features do.

**Rasmus Torp**[*]
Department of Computer Science
Dartmouth College
rasmus.torp.gr@dartmouth.edu

**Shailen K. Smith**[*]
Department of Computer Science
Dartmouth College
shailen.k.smith.gr@dartmouth.edu

**Adam Breuer**[†]
Department of Computer Science & Department of Government
Dartmouth College
adam.breuer@dartmouth.edu

## ABSTRACT

In this paper, we challenge the prevailing view that information dependency (including rote memorization) drives training data exposure to image reconstruction attacks. We show that extensive exposure can persist without rote memorization and is instead caused by a tunable connection to adversarial robustness. We begin by presenting three surprising results: (1) recent defenses that inhibit reconstruction by Model Inversion Attacks (MIAs), which evaluate leakage under an idealized attacker, do *not* reduce standard measures of information dependency (HSIC); (2) models that maximally memorize their training datasets remain robust to MIA reconstruction; and (3) models trained without seeing 97% of the training pixels, where recent information-theoretic bounds give arbitrarily strong privacy guarantees under standard assumptions, can still be devastatingly reconstructed by MIA.

To explain these findings, we provide causal evidence that privacy under MIA arises from what the adversarial examples literature calls "non-robust" features (generalizable but imperceptible and unstable features). We further show that recent MIA defenses obtain their privacy improvements by unintentionally shifting models toward such features. To establish this causal relationship, we introduce **An**t**i A**dversarial **T**raining (**AT-AT** 🏛), a training regime that intentionally learns non-robust features to obtain both superior reconstruction defense and higher accuracy than state-of-the-art defenses. Our results revise the prevailing understanding of training data exposure and reveal a new privacy-robustness tradeoff.

## 1 INTRODUCTION

A foundational goal of contemporary machine learning is to learn *generalizable* model parameters without encoding the private training dataset in a manner that *exposes* it to leakage. However, a rapidly expanding class of practical privacy attacks has demonstrated reconstruction of private training data in a variety of learning settings and model architectures. Recent variants of training data privacy attacks apply not only to CNNs (Struppek et al., 2022; Carlini et al., 2022; Mehnaz et al., 2022; Yeom et al., 2018; Shokri et al., 2017), but also to LLMs (Shi et al., 2024; Nasr et al., 2023; Carlini et al., 2021) and diffusion models (Carlini et al., 2023).

Across this attack landscape, model inversion attacks (MIAs) have emerged as a mainstream tool for studying and evaluating training data exposure, particularly in the high-resolution image classifica-

---

[*]**Equal contribution.**

[†]**Breuer Lab gratefully acknowledges the support of the OpenAI Cybersecurity Grant.**
Replication code is available at https://github.com/BreuerLabs/Anti-Adversarial-Training

tion domain. Unlike more conservative attacks that merely probe for the presence of some exposed training examples, MIAs test the degree to which a powerful attacker armed with white-box model access and significant computational and data resources can reconstruct information from the training examples. While early MIAs applied SGD to directly optimize reconstructions of individual training examples (Fredrikson et al., 2015), contemporary MIAs use sophisticated gradient techniques and external data to attempt to infer the full set of class-level characteristics for each class in the target model's training data (Qiu et al., 2024; Struppek et al., 2022; Haim et al., 2022).

However, while MIAs lower-bound the extent to which contemporary vision models expose their training data to reconstruction, they do not explain what aspect of a model encodes this vulnerability, or how to prevent it. This raises a fundamental question:

*What properties of learned representations encode vulnerability to training data leakage and reconstruction, and how can they be controlled?*

◇ ◇ ◇

Understanding these properties has far-reaching implications for learning that extend beyond the privacy domain. For example, influential recent work conjectures that obtaining stronger learning performance may require *more* extensive model-to-training-data dependencies, including rote memorization and exposure of the training set, not only for vanilla CNNs (Brown et al., 2021; Feldman, 2020), but also for recent paradigms including LLMs (Du et al., 2025; Tirumala et al., 2022), diffusion (e.g., Shah et al., 2025; Carlini et al., 2023), and self-supervised learning (Wang et al., 2024).

**The prevailing 'training data information dependency→training data reconstruction' theory.** An influential recent line of algorithms has obtained practically performant defenses against high-resolution MIA image reconstruction. Their practical performance is widely attributed to their reasonable theoretic hypothesis that training data privacy leakage arises from direct and excessive information dependency between training inputs and the model's internal representations or outputs (Dibbo et al., 2024; Peng et al., 2022; Wang et al., 2020). To mitigate this hypothesized dependency, Mutual Information Defense (MID) (Wang et al., 2020) introduces a regularizer that reduces mutual information between the training examples and intermediate feature representations, arguing that this "limit[s] the redundant information about the model input contained in the prediction" so that MIAs cannot "exploit this correlation." Building on this idea, the popular BiDO algorithm (Peng et al., 2022) penalizes a kernel-based information dependency measure (HSIC) between training examples and intermediate embeddings, arguing that this "limits redundant information propagated from the inputs to the latent representations that can be exploited by the [MIA] adversary."

The same hypothesis appears again in SCA (Dibbo et al., 2024), which argues that the relevant MIA training data reconstruction vulnerability is reduced by directly "jettison[ing] unnecessary private information in the input image," e.g. by sparse-coding training images before they are seen by the model, and by also "jettison[ing]... unnecessary private information from downstream layers" to attempt to preclude high-fidelity memorization of the inputs, rather than disincentivizing it via the loss function as in Peng et al. (2022) and Wang et al. (2020).

Similarly, the highly performant Transfer Learning (TL-DMI) approach (Ho et al., 2024) advances a main hypothesis that "by reducing the number of parameters fine-tuned with private dataset, [TL-DMI] could reduce the amount of private information encoded in the model, making it more difficult for adversaries to reconstruct private training data." This hypothesis is widely credited with the strong practical performance of TL-DMI, an approach that uses training data only for fine-tuning.

In sum, state-of-the-art privacy defenses advance a shared and intuitively reasonable theoretic argument: they argue that they increase the generic privacy of their training data by reducing information dependency between training examples and the target model's internal representations or outputs.

**Our contribution I: Challenging the theoretic bases of MIA defenses.**

In this paper, we challenge this foundational hypothesis. Specifically, we motivate our techniques by first presenting three surprising experiments that indicate that the information dependency view is inadequate to explain MIA reconstruction vulnerability:

**1.** We show that recent defenses that successfully inhibit reconstruction by MIAs do *not* reduce the approximation of information dependency, HSIC, used in MIA research. In fact, when we modify defenses such that they do reduce HSIC, they cease to provide effective reconstruction defense.

**2.** We train models to perfectly rote-memorize the training data such that their data-to-model information dependencies are intuitively maximal (at least in one intuitive view of dependency), yet we show that such 'max-information-dependency' models are *robust* to MIA reconstructions.

**3.** We train models without seeing >97% of each training image's pixels, such that information dependencies are minimal (at least in one intuitive view of dependency). Recent information-theoretic bounds give these >97% pixels arbitrarily strong privacy guarantees under standard assumptions. Yet, we show that this training data can still be devastatingly reconstructed in practice: deleting >97% of the training data yields *no significant privacy increase* against MIA reconstruction.

In sum, our motivating experiments formally show that the main theoretic approach in privacy/MIA defense is incorrect: reducing information dependency does not cause privacy as widely believed. SOTA defenses thus obtain their impressive performance by a different theoretic mechanism.

**Our contribution II: The privacy-robustness feature tradeoff.**

To explain these results, we provide evidence that privacy under MIA arises instead from what the adversarial examples literature calls "non-robust" features (generalizable but imperceptible and unstable features), and recent MIA defenses obtain their privacy improvements by unintentionally shifting models toward such features. To accomplish this, we conduct the first systematic evaluation of the robustness of privacy/MIA defenses to adversarial examples (Szegedy et al., 2014; Ilyas et al., 2019). We find that all recent defenses that attempt to reduce information dependency (MID, BiDO, TL-DMI) obtain a reduction in privacy leakage under MIA that is directly proportional to a corresponding increase in their vulnerability to adversarial examples.

We then explicitly test our hypothesis that these defenses' privacy leakage reduction (measured by AttAcc@1 under PPA or IF-GMI attacks) is a linear function of their robust accuracy (or, equivalently, a linear function of their reliance on non-robust features). We find that we can almost perfectly predict each of these defenses' MIA robustness via a simple linear function of their vulnerability to adversarial examples ($R^2$=0.95). As such, we can compute the *adversarial robustness cost of MIA privacy gain.* For example, reducing MIA leakage by 1 percentage point corresponds to a statistically significant 0.31-5.4 percentage point decrease in robust accuracy, depending on the magnitude of the adversarial example chosen, where MIA leakage is measured via the standard AttAcc@1 metric using a standard PPA attack.

As such, MIA defenses unintentionally make a tradeoff: by shifting towards non-robust features, they increase privacy by stripping semantically meaningful information from the model, but this opens significant new vulnerabilities to adversarial examples.

**Our contribution III: Anti-Adversarial Training (AT-AT 🏯 ).**

If recent privacy/MIA defenses appear to work by unintentionally 'nonrobustifying' the model's feature space, then it stands to reason that we can *causally induce* superior defense performance by deliberately exploiting this mechanism. To establish this causal relationship, we introduce Anti Adversarial Training (AT-AT 🏯 ), a novel generic privacy/MIA defense training regime that balances model accuracy with gradient steps that reverse standard adversarial training (Madry et al., 2017). Rather than promoting robustness to adversarial perturbations, **AT-AT** applies targeted gradient updates that suppress reliance on robust features. This compels the model to instead learn non-robust (but still generalizable) features that are semantically meaningless to humans and thereby hinder visual reconstruction under MIA.

In a causal framework, we show that 'treating' a standard ResNet-152 with **AT-AT**'s non-robust feature reward causes a reduction in reconstruction rate (AttAcc@1) from 84% to 6.5% ($p < 10^{-16}$).

We then show that our anti-adversarial approach obtains superior MIA defense at higher levels of accuracy than baselines across the standard architectures, datasets, and reconstruction metrics in the high-resolution MIA domain. As with other generic defenses, **AT-AT**'s performance comes at the cost of increased vulnerability to adversarial examples. However, in our case, this vulnerability cost is a tunable parameter, offering a new design axis for private models.

**Implications for private learning.** Our results have four implications. First, among the most celebrated results in ML is the finding that adversarial examples do not reveal bugs, but rather non-robust features that are generalizable and discriminatory of the classes we want to learn, yet visually imperceptible to humans (Ilyas et al., 2019). Our argument in this paper is that these properties

make them ideal instruments for privacy-preserving learning, where our goal is to classify accurately without encoding visually meaningful information that is vulnerable to visual reconstructions.

Second, eliminating training data information dependency does not guarantee privacy, as leakage can persist in its absence. Indeed, if recent work on learning performance is correct in suggesting that dependencies up to and including rote memorization can confer accuracy benefits, then such benefits may be attainable without incurring all of the privacy risks it is widely assumed to entail.

On the other hand, our results corroborate the recent idea that there may be an inherent privacy vulnerability associated with robust learned representations. As such, robust models may be inherently privacy compromising, beyond any specific aspects of how they encode data. This intuition is consistent with recent works analyzing connections between privacy and robustness (see App. F.2).

Finally, our results suggest that the assumptions of recent theoretical privacy guarantees may be optimistic, rendering their protections practically inapplicable against recent MIAs.

## 2 EXPERIMENTAL SETUP, SCOPE, AND REPLICATION

The experiments throughout this paper focus on the white-box high-resolution vision setting that is central to contemporary empirical privacy and MIA research (see App. C for further scope details). Specifically, our experiments evaluate the facial recognition setting on the standard hi-res image MIA benchmark datasets (*FaceScrub* (Ng & Winkler, 2014), *CelebA* (Liu et al., 2015)) under the established benchmarks for hi-res white-box attacks (*PPA* (Struppek et al., 2022), plus recent, high-performance *IF-GMI* (Qiu et al., 2024) and *PPDG* (Peng et al., 2024)), on the standard high-resolution image architectures (*ResNet-152*, *ResNet-18*, *DenseNet-169*) that are the focus of recent experiments in the literature (Struppek et al., 2022; Qiu et al., 2024; Struppek et al., 2024; Ho et al., 2024; Koh et al., 2024; Liu & Chen, 2024; Peng et al., 2022). Section 4 adds robustness metrics from four SOTA adversarial example attacks (Croce & Hein, 2020a;b; Andriushchenko et al., 2020), each under various attack strengths $\epsilon$. App. F.3 gives experimental details for adversarial attacks. In App. I (Table 26b) we also consider an extra *Stanford Dogs* dataset (Khosla et al., 2011) for variety.

**Defense Baselines: generic information defenses and gradient-suppressing defenses.** As discussed above, our main focus is on class of state-of-the-art *generic defenses* that seek to hinder reconstruction via private information reduction. This includes *MID* (Wang et al., 2020), *BiDO* (Peng et al., 2022), and the recent *TL-DMI* (Ho et al., 2024). For the sake of comparison, we also add four main recent *attack gradient suppressing* MIA defenses, *NegLS* (Struppek et al., 2024), *RoLSS*, *RoLSS-SSF* (Koh et al., 2024), *Trap-MID* (Liu & Chen, 2024). These defenses do not seek generic privacy, but rather seek to inhibit the gradient steps taken by modern MIAs during their optimization stages. See App. A for experimental details of target models, datasets, and defense baselines.

**Privacy Metrics.** As in recent work, we measure privacy leakage via the main *Attack Acc@1* metric, which reports the accuracy of an external Inception-v3 model at guessing the identity of the attacker's reconstructed image as its #1 guess. Where appropriate, we also report the similar *Attack Acc@5* metric; *L2-FaceNet Distance* (the minimum $\ell_2$-norm FaceNet embedding distance between the attacker's reconstruction and any training image in the attacked class); and the *Average Evaluation Confidence* of the external Inception-v3 model that the reconstructed image is from the respective original image's class. We also show *qualitative reconstruction* comparisons.

This yields 7 defenses×3 attacks×3 architectures×3 datasets×(4 robustness+4 privacy metrics).

**Replication code.** Full replication codes for all experiments are available at https://github.com/BreuerLabs/Anti-Adversarial-Training.

## 3 THREE EXPERIMENTS OVERTURN THE DEPENDENCY → LEAKAGE VIEW

If training data exposure to MIA were explained by information dependency/memorization, then three intuitive hypotheses would hold: (1) Effective defenses would *reduce* information dependency metrics; (2) Models that fully rote-memorize the training data would be *vulnerable* to MIA; (3) Models trained without seeing 97.5% of each training image's pixels (i.e. where memorization is upper bounded at 2.5%) would be *robust* to MIA reconstruction. Our goal in this section is to show

**Table 1:** HSIC$(X, Z_j)$ for defenses on FaceScrub (ResNet-152). PPA **AttAcc@1** measures reconstruction.

| Arch. | Defense | TestAcc↑ | AttAcc@1↓ | HSIC$(X, Z_0)$ | HSIC$(X, Z_1)$ | HSIC$(X, Z_2)$ | HSIC$(X, Z_3)$ |
|---|---|---|---|---|---|---|---|
| ResNet-152 | NoDef | 0.958 | 0.882 | 71.66 | 72.32 | 72.72 | 72.72 |
| | MID | 0.950 | 0.391 | 71.52 | 72.57 | 72.84 | 61.98 |
| | BiDO | 0.928 | 0.780 | 73.68 | 73.68 | 73.68 | 73.63 |
| | TL-DMI | 0.911 | 0.190 | 71.73 | 72.52 | 72.93 | 72.58 |
| | NegLS | 0.906 | 0.160 | 71.64 | 72.33 | 72.29 | 73.63 |
| | RoLSS | 0.886 | 0.474 | 72.09 | 72.30 | 29.69 | 61.92 |
| | RoLSS-SSF | 0.869 | 0.343 | 71.88 | 72.02 | 32.75 | 64.94 |
| | Trap-MID | 0.926 | 0.401 | 69.80 | 71.88 | 72.62 | 69.28 |
| | BiDO** | 0.940 | 0.815 | 2.159 | 1.256 | 9.310 | 71.81 |

experimentally that each of these hypotheses is false. Thus, the prevailing view that information dependency drives training data exposure to reconstruction is incomplete.

**1. Effective defenses do *not* reduce information dependency metrics.** First, we show that contemporary defenses that are widely believed to work by reducing training-data-to-model dependency do *not* in fact reduce this dependency, and may in fact increase it, in some cases even according to their own metrics. In the defense literature, the Hilbert–Schmidt Independence Criterion (HSIC) (Gretton et al., 2005) is the standard dependency metric that can be computed for an arbitrary model. Defenses such as BiDO and BiDO+ (Peng et al., 2022; 2025) explicitly penalize (approximated) HSIC between inputs $X$ and intermediate embeddings $Z_j$:

$$\text{HSIC}(X, Z_j) = \left\| \mathbb{E}[\phi(X)\psi(Z_j)^\top] - \mathbb{E}[\phi(X)]\mathbb{E}[\psi(Z_j)]^\top \right\|_{HS}^2$$

Where nonlinear feature transformations $\phi, \psi$ are created such that $\phi(X)$, $\psi(Z_j)$ are members of reproducing kernel Hilbert spaces, and $\|\cdot\|_{HS}^2$ denotes the Hilbert-Schmidt norm.

We train all defenses described in Section 2, then measure their average HSIC across the train set at all standard intermediate embeddings $Z_i$ using the exact HSIC implementation from Peng et al. (2022). We then run standard PPA attacks to test their leakage (AttAcc@1) under MIA.

**HSIC Results: reducing HSIC does not cause privacy.** Table 1 reports results (also see the additional results in App. D). Neither BiDO nor the defenses TL-DMI and NegLS that obtain the largest reductions in reconstruction (AttAcc@1) significantly decrease HSIC$(X, Z_i)$ according to the standard HSIC approximation used in MIA.

BiDO also allows users to directly manipulate the weight $\lambda_x$ on HSIC in its loss. We use this to experimentally lower HSIC by training a BiDO** model with a different $\lambda_x$ (see App. D.1). This experimental BiDO** 'HSIC treatment' model succeeds in lowering HSIC$(X, Z_i)$ compared to standard BiDO parameters, indicating significantly reduced information dependency, yet it obtains *worse* defense. In sum, reducing HSIC as measured in MIA does not cause privacy under MIA.

**2. A model that fully rote-memorizes the dataset is *not* vulnerable to reconstruction.** Second, if training-data-to-model information dependency drives leakage, then a model that *maximally* memorizes should be maximally vulnerable. The seminal permuted-label setting of

**Table 2:** PPA results with and without label permutation.

| Arch. | Labels | TrainAcc ↑ | TestAcc ↑ | L2-Face ↑ |
|---|---|---|---|---|
| RN-152 | Not permuted | 1.000 | 0.958 | 0.768 |
| | Permuted | 0.996 | 0.001 | 1.249 |
| RN-18 | Not permuted | 1.000 | 0.950 | 0.781 |
| | Permuted | 0.994 | 0.001 | 1.251 |

Zhang et al. (2017) provides a clean test of this hypothesis: when training labels are randomized, networks are known to achieve perfect training accuracy by rote-memorizing the entire training dataset without learning any meaningful features. Table 2 shows that MIAs completely fail in this setting: reconstructions collapse, yielding high L2 reconstruction distance (note that other standard metrics such as AttAcc@1 are not well-defined in the permuted setting). Despite full rote memorization, leakage is minimal. This suggests that leakage is not driven by rote training data information dependency, but rather by some other property. See App. A.3.1 for further details.

**3. Unseen training data can still be reconstructed under MIA, contra recent theoretic bounds.** Third, if information dependency/memorization drives privacy leakage, then deleting the training data would prevent leakage (notwithstanding the effect on accuracy). However, we now show that models trained without seeing >97% of the training data pixels, for which recent information-theoretic privacy results give arbitrarily large bounds against reconstruction under standard assump-

**Table 3:** Unseen-Pixels trained on FaceScrub obtains very limited defense against reconstruction by PPA.

| Arch. | Defense | %Pix-deleted | TestAcc ↑ | AttAcc@1 ↓ | AttAcc@5 ↓ | L2-Face ↑ | EvalConf ↓ |
|---|---|---|---|---|---|---|---|
| RN-152 | NoDef | 0 | 0.958 | 0.881 | 0.982 | 0.781 | 0.859 |
| | NoDef-EarlyStop | 0 | 0.917 | 0.634 | 0.875 | 0.852 | 0.607 |
| | TL-DMI | 0 | 0.911 | 0.163 | 0.398 | 1.072 | 0.152 |
| | **Unseen-Pixels** | **97.8** | 0.910 | 0.592 | 0.851 | 0.819 | 0.566 |
| RN-18 | NoDef | 0 | 0.950 | 0.903 | 0.985 | 0.778 | 0.884 |
| | NoDef-EarlyStop | 0 | 0.890 | 0.620 | 0.877 | 0.840 | 0.596 |
| | TL-DMI | 0 | 0.906 | 0.219 | 0.487 | 1.031 | 0.206 |
| | **Unseen-Pixels** | **97.2** | 0.892 | 0.546 | 0.813 | 0.844 | 0.522 |

tions, can still be devastatingly reconstructed by MIAs. We construct a novel lasso-based technique that deletes over 97% of the pixels in each training image *prior* to training (so they are completely unseen by the model), yet allows us to train models with test accuracy that is within a few points of the accuracy of a vanilla model trained on uncensored images. Fig. 1 shows examples of pixel-deleted images used for training. App. E.2 describes our deletion technique.

**Unseen pixel leakage is theoretically impossible under standard unbiasedness assumptions.** A recent stream of theory research seeks information-theoretic privacy guarantees against image training data reconstruction by lower-bounding the variance of any unbiased estimator of the train set via, e.g., the Cramér-Rao bound and Fisher Information Loss (Hannun et al., 2021; Guo et al., 2022), or the HCR bound (Guo et al., 2022; Chaudhuri et al., 2024). Interestingly, in the same unbiased estimator setting, our model trained on >97% pixel-deleted data enjoys a perfect privacy guarantee (unbounded reconstruction variance) on these >97% deleted pixels (see App. E.3).

>**97% Unseen-Pixels experiment.** Table 3 compares PPA reconstructions on models trained on the >97% pixel-deleted training dataset versus various models trained on the uncensored data, including a vanilla undefended model (NoDef) and a SOTA MIA defense (TL-DMI). We also compare against an undefended model trained with early-stopping to provide an extra baseline with accuracy comparable to our Unseen-Pixel model's accuracy, which is motivated by the common observation that less accurate models are naturally more robust to MIAs (see Struppek et al. (2024)).

**Unseen-Pixels results.** Surprisingly, deleting >97% of the pixels before training fails to prevent MIA reconstructions. Table 3 shows that PPA attacks on the Unseen-Pixels model still obtain reconstructions that are classified as their original class >50% of the time (per AttAcc@1). For comparison, that means the Unseen-Pixels model trained on >97% censored data is *not* significantly more private than an undefended model with about the same accuracy that is trained on uncensored data. More specifically, it is only marginally more private in terms of AttAcc@1, and marginally *less* private in terms of the L2 metric on RN-152. See addl. results in App. E. This also suggests that the assumptions of recent privacy bounds are overly optimistic, rendering their protections practically inapplicable against recent attacks (see App. E.3).

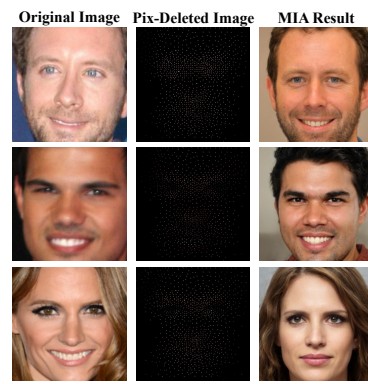

Original Image    Pix-Deleted Image    MIA Result

**Figure 1:** Orig. & pixel-deleted images (*zoom in for* 2.2% *training pixels*).

## 4 THE PRIVACY-ADVERSARIAL EXAMPLE ROBUSTNESS TRADEOFF

Our goal in this section is to describe a previously unobserved privacy-adversarial example robustness tradeoff that characterizes the performance of generic information dependency-based MIA defenses on the main architectures and datasets used to benchmark hi-res privacy attacks/defenses in recent work. We show that the defense performance obtained by the class of defenses widely believed to work via information dependency reduction is highly correlated with vulnerability to adversarial examples and non-robust feature bases. We accomplish this by presenting the first systematic tests of the robustness of MIA defenses to a very different attack vector: adversarial examples.

**Tradeoff experiment setup.** Following seminal work by Carlini et al. (2019); Croce & Hein (2020b) and others, a scholarly consensus has emerged regarding the design principles necessary to ensure the rigor of experiments that measure robustness to adversarial examples. We carefully

**Table 4:** Leakage (AttAcc@1) vs. clean/robust accuracies under AutoAttack for various $\epsilon$.

| Data | Arch. | Defense | PPA-AttAcc@1↓ | IF-AttAcc@1↓ | TestAcc↑ | $\epsilon$=0.031↑ | $\epsilon$=0.0025↑ | $\epsilon$=0.0005↑ | $\epsilon$=$10^{-5}$↑ |
|---|---|---|---|---|---|---|---|---|---|
| *Defenses that attempt to increase privacy generically by reducing information dependency/memorization* | | | | | | | | | |
| FaceScrub | RN-152 | NoDef | 0.882 | 0.987 | 0.958 | 0.000 | 0.122 | 0.899 | 0.960 |
| | | MID | 0.391 | 0.391 | 0.950 | 0.000 | 0.006 | 0.770 | 0.928 |
| | | BiDO | 0.780 | 0.937 | 0.928 | 0.000 | 0.121 | 0.847 | 0.923 |
| | | TL-DMI | 0.190 | 0.420 | 0.911 | 0.000 | 0.000 | 0.000 | 0.867 |
| | RN-18 | NoDef | 0.901 | 0.983 | 0.950 | 0.000 | 0.194 | 0.896 | 0.959 |
| | | MID | 0.714 | 0.776 | 0.924 | 0.000 | 0.093 | 0.825 | 0.856 |
| | | BiDO | 0.530 | 0.782 | 0.884 | 0.000 | 0.075 | 0.784 | 0.879 |
| | | TL-DMI | 0.240 | 0.515 | 0.906 | 0.000 | 0.000 | 0.003 | 0.886 |
| CelebA | RN-152 | NoDef | 0.497 | 0.793 | 0.838 | 0.000 | 0.050 | 0.675 | 0.818 |
| | | MID | 0.474 | 0.728 | 0.802 | 0.004 | 0.139 | 0.542 | 0.574 |
| | | BiDO | 0.362 | 0.597 | 0.747 | 0.000 | 0.062 | 0.590 | 0.748 |
| | | TL-DMI | 0.080 | 0.258 | 0.688 | 0.000 | 0.000 | 0.002 | 0.641 |
| | | TL-DMI-5 | 0.281 | 0.661 | 0.814 | 0.000 | 0.000 | 0.119 | 0.799 |
| | RN-18 | NoDef | 0.634 | 0.913 | 0.859 | 0.000 | 0.084 | 0.746 | 0.848 |
| | | MID | 0.386 | 0.550 | 0.737 | 0.003 | 0.168 | 0.452 | 0.464 |
| | | BiDO | 0.146 | 0.368 | 0.625 | 0.000 | 0.007 | 0.394 | 0.609 |
| | | TL-DMI | 0.153 | 0.439 | 0.752 | 0.000 | 0.000 | 0.028 | 0.718 |
| | | TL-DMI-5 | 0.324 | 0.750 | 0.831 | 0.000 | 0.000 | 0.146 | 0.802 |
| *Defenses that suppress attack gradients* | | | | | | | | | |
| FaceScrub | RN-152 | NegLS | 0.160 | 0.147 | 0.906 | 0.000 | 0.048 | 0.769 | 0.905 |
| | | RoLSS | 0.474 | 0.522 | 0.886 | 0.000 | 0.009 | 0.744 | 0.882 |
| | | RoLSS-SSF | 0.343 | 0.373 | 0.869 | 0.000 | 0.013 | 0.721 | 0.857 |
| | | Trap-MID | 0.401 | 0.618 | 0.926 | 0.000 | 0.000 | 0.000 | 0.856 |
| | RN-18 | NegLS | 0.649 | 0.932 | 0.929 | 0.000 | 0.215 | 0.854 | 0.933 |
| | | RoLSS | 0.858 | 0.979 | 0.950 | 0.000 | 0.002 | 0.819 | 0.948 |
| | | RoLSS-SSF | 0.849 | 0.973 | 0.950 | 0.000 | 0.002 | 0.813 | 0.951 |
| | | Trap-MID | 0.735 | 0.918 | 0.928 | 0.000 | 0.000 | 0.530 | 0.914 |
| CelebA | RN-152 | NegLS | 0.340 | 0.554 | 0.805 | 0.000 | 0.013 | 0.559 | 0.771 |
| | | RoLSS | 0.046 | 0.045 | 0.537 | 0.000 | 0.000 | 0.086 | 0.530 |
| | | RoLSS-SSF | 0.039 | 0.044 | 0.428 | 0.000 | 0.000 | 0.042 | 0.406 |
| | | Trap-MID | 0.330 | 0.417 | 0.848 | 0.000 | 0.000 | 0.333 | 0.828 |
| | RN-18 | NegLS | 0.667 | 0.953 | 0.838 | 0.000 | 0.100 | 0.711 | 0.825 |
| | | RoLSS | 0.506 | 0.724 | 0.830 | 0.000 | 0.069 | 0.701 | 0.813 |
| | | RoLSS-SSF | 0.519 | 0.765 | 0.839 | 0.000 | 0.080 | 0.723 | 0.834 |
| | | Trap-MID | 0.442 | 0.670 | 0.830 | 0.000 | 0.044 | 0.685 | 0.827 |

follow these design principles here (see App. F.1 and F.3). We report *robust accuracy*, i.e., worst-case accuracy on adversarial examples across all four attacks in the standard AutoAttack ensemble (Croce & Hein, 2020b): untargeted Auto-PGD; targeted DLR Auto-PGD (Croce & Hein, 2020b); targeted FAB (Croce & Hein, 2020a); and Square (Andriushchenko et al., 2020). We find that all defenses have 0% robust accuracy under AutoAttack with default $\epsilon$=0.031 ($\approx$8/255), rendering comparisons uninformative. Thus we recompute attacks with a range of smaller perturbations: $\epsilon \in \{0.031, 0.0025, 0.0005, 10^{-5}\}$. This evaluates defenses' unintentional reliance on imperceptible 'non-robust features' of different magnitudes. App. F.3 gives details.

**Tradeoff raw results & estimating the privacy-adversarial robustness tradeoff.** Table 4 presents results, and Tables 11, 12, 13, 14 in App. F.4 also report robust accuracies under each of AutoAttack's constituent attacks. Note that for all generic privacy defenses (MID, BiDO, TL-DMI), less privacy leakage (lower AttAcc@1) corresponds to *worse* robust accuracy under AutoAttack (Table 4), particularly with smaller $\epsilon$. To obtain a principled test of this tradeoff, we estimate OLS linear regressions of the following form. This allows us to explicitly test our hypothesis that these defenses' privacy leakage reduction (AttAcc@1 under PPA or IF-GMI) is a linear function of their robust accuracy (or, equivalently, a linear function of their reliance on non-robust features). We include clean test accuracy in the regression, as it has been widely observed that more accurate models leak more (Struppek et al., 2024; Ho et al., 2024; Liu & Chen, 2024):

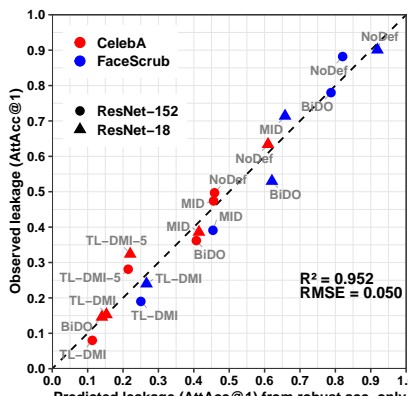

**Figure 2:** Obs. vs. Pred. leakage (PPA), pred. *only* via robust acc. $\beta$ weights.

$$\text{Leakage}_{\text{AttAcc@1},j} = \beta_0 + \beta_1 \text{TestAcc}_j + \beta_2 \text{Acc}_{\epsilon=0.0025,j} + \beta_3 \text{Acc}_{\epsilon=0.0005,j} + \beta_4 \text{Acc}_{\epsilon=10^{-5},j} + e_j \quad (1)$$

**Specification Invariance.** To explore the stability of results, we are careful to also estimate a total of 56 various specifications of this OLS model that explore alternate hypotheses, including alternative Beta regression variants with appropriate distributional assumptions and logit link ($g(x)$) to capture the fact that the response (AttAcc@1 under PPA or IF-GMI) is bounded in $[0, 1]$:

$$\text{Leakage}_{\text{AttAcc@1},j} \sim \text{Beta}(\mu_j \phi, (1 - \mu_j)\phi)$$
$$g(\mu_j) = \beta_0 + \beta_1 \text{TestAcc}_j + \beta_2 \text{Acc}_{\epsilon=0.0025,\,j} + \beta_3 \text{Acc}_{\epsilon=0.0005,\,j} + \beta_4 \text{Acc}_{\epsilon=10^{-5},\,j} \quad (2)$$

**Results.** Fig. 2 plots fit and Table 5 presents a small selection of results (see App. G). We find that (1) leakage (AttAcc@1 under both PPA vs. IF-GMI) is strongly and significantly correlated with robust accuracy at all levels of $\epsilon$ (note 95% CI), with $R^2 \approx 0.93 - 0.95$, and mean abs. error of just $0.042 - 0.05$. Contrary to widespread belief, (2) clean test accuracy is *only weakly or insignificantly correlated* with leakage after controlling

**Table 5:** OLS & Beta leakage (PPA AttAcc@1) regressions $\pm 95\%$ CI. OLS CIs use small-sample $t_{0.975,13}$=2.160 with HC2 SEs; Beta uses Wald ($\pm 1.96$SE) on the logit scale.

|  | **OLS ($\pm$ 95% CI)** | **Beta ($\pm$ 95% CI)** |
|---|---|---|
| Test Acc. | -0.011 $\pm$ 1.033 | -0.192 $\pm$ 3.101 |
| $\epsilon$=0.0025 | **2.299 $\pm$ 1.596** | **12.827 $\pm$ 3.939** |
| $\epsilon$=0.0005 | 0.211 $\pm$ 0.234 | **0.737 $\pm$ 0.637** |
| $\epsilon$=$10^{-5}$ | **0.795 $\pm$ 0.778** | **4.390 $\pm$ 2.312** |
| $R^2$ | 0.934 (adj.) | 0.952 (pseudo) |
| MAE | 0.047 | 0.042 |

for robust accuracy. Thus, at a high level, we can almost perfectly predict a defense's leakage just from knowing its robust accuracy (Fig. 2), and knowing its clean accuracy adds little to no additional information that improves this prediction. Finally, as we hypothesized, this privacy-robustness tradeoff does *not* hold for gradient-suppressing defenses, which (as expected) instead achieve their defense by various other means. Our results hold across all 56 OLS and Beta models, which show they are invariant to specification, PPA and IF-GMI attacks, different distributional assumptions, controls for dataset/architecture, various robust standard errors estimators, etc. App. G gives details.

**The robustness cost of privacy gain.** We can use these linear models to compute the cost of privacy. Per the Beta model, reducing PPA leakage (AttAcc@1) by 1 percentage point (pp) corresponds to a significant *0.31 pp decrease* in AutoAttack robust accuracy at $\epsilon$=0.0025, (95% CI: 0.24–0.45), a significant *5.4 pp decrease* at $\epsilon$=0.0005 (95% CI: 2.9–58.3), and a significant *0.91 pp decrease* at $\epsilon$=$10^{-5}$ (95% CI: 0.60–1.93). These results highlight that privacy-robustness tradeoffs are nonuniform: the largest and smallest-magnitude non-robust features experience small but significant costs, while the 'medium' $\epsilon$=0.0005 magnitude experiences disproportionately large and unstable costs.

> *These results provide evidence that non-robust features were unknowingly* **correlated** *with privacy. Can non-robust features* **cause** *privacy through deliberate algorithmic design?*

## 5 ANTI ADVERSARIAL TRAINING (AT-AT) 🐎

We now show how to exploit this mechanism to *causally induce* superior defense at higher accuracy. To establish this causal relationship, we describe **Anti Adversarial Training** (**AT-AT** 🐎), a novel privacy/MIA defense training regime that balances model accuracy with gradient steps that reverse standard adversarial training. These steps allow us to causally 'treat' the learned feature basis by intentionally shifting it towards non-robust (but still generalizable) features that are visually imperceptible to humans and thus unamenable to visual reconstruction under MIA.

For intuition, suppose that on a given iteration of SGD, we draw an image $x_{\text{Yoda}}$ of class $y$=Yoda.

**Vanilla Training.** Vanilla loss: $\min_\theta \mathbb{E}_{(x,y) \sim \mathcal{D}}[\mathcal{L}(\theta, x, y)]$ just rewards accurate mappings of clean images: $x_{\text{Yoda}} \rightarrow$ Yoda. It is well-known that this approach learns both robust and non-robust features.

**Classic Adversarial Training (AT).** Classic AT (Madry et al., 2017) replaces each clean training example $x$ in the SGD-loop with the untargeted adversarial perturbation $x + \delta$ that is most likely to flip the $x$'s classified label. The AT loss function, $\max_{\delta \in \mathcal{S}} \mathcal{L}(\theta, x + \delta, y)$, then treats features that are robust to this perturbation as the 'signal' we seek to learn. For example, a perturbation $\delta$ might flip Yoda into Luke, i.e., $(x_{\text{Yoda}} + \delta_{\text{Luke}}) \rightarrow y'$=Luke$\neq$Yoda. AT rewards learning $x_{\text{Yoda}}$ features that are *robust* to $\delta$, i.e., rewards $(x_{\text{Yoda}} + \delta_{\text{Luke}}) \rightarrow$ Yoda, and penalizes features in $\delta$ that are *non-robust*.

**Anti Adversarial Training (AT-AT** 🐎**).** At a high level, our goal is the opposite: **AT-AT** uses a loss term that treats the visually meaningful image $x$ as the 'random noise' we seek to ignore,

and the imperceptible-but-generalizable 'non-robust' features exposed in perturbation $\delta$ as the 'signal' we seek to learn. Thus, compared to classic AT, we optimize for the opposite mapping: $(x_{\text{Yoda}} + \delta_{\text{Luke}}) \rightarrow$ **Luke**. More formally, we seek $\min_{\delta \in \mathcal{S}} \mathcal{L}(\theta, x + \delta, \mathbf{y}')$, where $\mathbf{y}' \neq y$. However, as a practical matter, it is infeasible to obtain competitive accuracy for challenging learning tasks like hi-res facial recognition using *only* non-robust features. Thus, we adopt a dual loss function that balances vanilla accuracy (incl. robust features) with non-robust features via user-chosen $\lambda$:

$$\textbf{AT-AT Objective:} \quad \min_\theta \ \mathbb{E}_{(x,y) \sim \mathcal{D}}[\mathcal{L}(\theta, x, y) \ + \ \lambda \cdot \min_{\delta \in \mathcal{S}} \mathcal{L}(\theta, x + \delta, \mathbf{y}')], \text{ where } \mathbf{y}' \neq y \quad (3)$$

This **AT-AT** loss can be optimized similarly to a targeted variant of vanilla AT. Each SGD iteration $i$ draws a training image $x$ and a target class $y'$ each $\mathcal{UAR}$. It then computes a targeted adversarial perturbation $\delta$ (via, e.g., PGD) that causes the current training model $\mathcal{M}_i$ to classify $x$ not as $y$, but as $y'$, e.g., "$(x_{\text{Yoda}} + \delta_{\text{Luke}}) \rightarrow$ Luke." Finally, it sums vanilla loss on $y$ and anti-adversarial loss on $y'$. The latter penalizes robust features in $x$ and rewards non-robust ones exposed by perturbation $\delta$.

**Anti-Adversarial Training (AT-AT)** 🏛

**input** Data $\mathcal{D}$, Model $\mathcal{M}(\theta, \lambda)$
  **for** $(x, y) \in \mathcal{D}$ **do**
    $y' \sim \text{Uniform}(C \setminus y)$
    $x_{\text{adv}} \leftarrow \min_{||x'\text{-}x||_\infty \leq \varepsilon} \mathcal{L}_{\mathcal{M}}(x', y')$ # PGD
    $\mathcal{L} \leftarrow \underbrace{\mathcal{L}_{\mathcal{M}}(x, y)}_{\text{vanilla}} + \lambda \cdot \underbrace{\mathcal{L}_{\mathcal{M}}(x_{\text{adv}}, y')}_{\text{anti-adv.}}$
    $\theta \leftarrow \theta - \alpha \nabla_\theta \mathcal{L}$
  **return** $\mathcal{M}$

**Causal effect of AT-AT loss term on privacy.** We train 10 RN-152's on FaceScrub and randomly assign half as 'control' $\lambda=0$ (equal to Vanilla/NoDef) and half as 'treatment' $\lambda>0$. Beta regression of AttAcc@1 on treatment shows that AT-AT's non-robust term causes a reduction in reconstruction rate (PPA AttAcc@1) from 84% to 6.5% (a 77× reduction in leakage odds, $p<10^{-16}$, $z=38.0$). See App. H.

## 6 EXPERIMENTS: AT-AT VS. GENERIC & GRADIENT DEFENSES

Our goal in this section is to show that **AT-AT** obtains superior reconstruction defense at higher accuracy versus both generic and gradient-suppressing defenses. Fig. 4 on the following page compares **AT-AT** (*red points*) to baselines in terms of accuracy and privacy across all datasets under PPA, IF-GMI, and PPDG inversion attacks via all privacy metrics on ResNet-152.

Across all privacy metrics, data, and attacks, **AT-AT** obtains superior privacy at higher levels of accuracy than the 7 state-of-the-art baselines. In addition to Fig. 4 below, we include full additional results on DenseNet-169 and ResNet-18 in App. I.2. We also provide full tables with confidence intervals in App. I.3. To demonstrate robustness across domains, we also report results on the Stanford Dogs dataset in Table 26.

## 7 CONCLUSION

We present converging correlational and causal evidence that reducing information dependency (including rote memorization) does not cause training data privacy under MIAs, whereas adversarially non-robust features do. This result overturns a dominant assumption in MIA and privacy literature. We describe **AT-AT**, a novel training regime that deliberately learns non-robust features to obtain superior MIA defense. While our scope is limited to hi-res image classification, our results raise timely questions of whether privacy-robustness tradeoffs generalize to paradigms such as LLMs and diffusion, where memorization is similarly thought to drive privacy vulnerabilities.

## 8 REPRODUCIBILITY STATEMENT

Our experiments were designed from the outset to be replicated. Full pushbutton cluster-ready replication codes are available from our lab's GitHub at https://github.com/BreuerLabs/Anti-Adversarial-Training. All MIAs, MIA defense baselines, and adversarial example attacks were computed using their authors' code and identical parameters, except where explicitly noted in our paper (e.g., our additional BiDO parameter-tuned variants, which we report with the exact parameter we used below, alongside where we report the BiDO version that we ran with the authors' original parameters). Further experiment and replication details are provided in App. A.

Our code also provides a suite of tools designed so that researchers can easily apply our experimental setups, evaluations, attacks, and defenses to their own research.

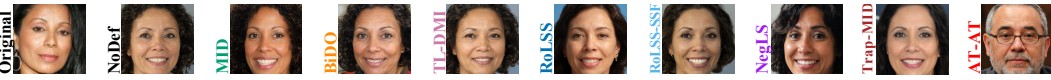

**Figure 3:** PPA reconstructions on ResNet-152. Images selected via max EvalConf. Full results in Fig. 6.

**Figure 4:** AT-AT vs Baselines on RN-152: *Top:* AttAcc@1 & L2 Distance; *Bottom:* AttAcc@5 & EvalConf.

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

# A    DEFERRED DETAILS OF EXPERIMENTS

## A.1    TARGET MODEL BASELINE DEFENSES

We consider two types of defense baselines. Our primary baselines are defenses that attempt to generically increase privacy by reducing information dependency/memorization. In each case, unless otherwise mentioned, we set all parameters identically to those used by their authors, and run defenses using the authors' replication codes:

- **No Defense (NoDef).** We consider vanilla ResNet-152, ResNet-18, and DenseNet-169 baselines, trained without additional privacy defenses.

- **Mutual Information Defense (MID) (Wang et al., 2020).** Inspired by previous information bottleneck research, MID introduces a regularizer that reduces training-data-to-model information dependency by penalizing (an approximation of) mutual information between training examples and the model's intermediate feature representations. We adapt the authors' original implementation to work with ResNet architectures; following previous work (see e.g. Struppek et al. (2024)), we add the information bottleneck layer between the average pooling and final linear layer, and set the bottleneck size to $k = 1024$. Also following previous work, we set the regularization penalty $\beta = 0.01$. On FaceScrub MID models, we use a lower learning rate of 0.0001, as we found it to provide a stronger defense at similar accuracy to the typical learning rate of 0.001. We drop the last batch of the validation set to ensure compatibility with the authors' code.

- **Bilateral Dependency Optimization (BiDO) (Peng et al., 2022).** BiDO introduces two regularizers, the first of which minimizes a kernel-based information dependency measure between inputs and intermediate embeddings (i.e. to penalize data-to-model information dependency), while the second maximizes the same dependency measure between intermediate embeddings and outputs (i.e. to reward accuracy). Following recent work (see e.g. Struppek et al. (2024); Peng et al. (2022)), we use the outputs of the four main ResNet blocks as the intermediate embeddings for the regularized loss. We drop the last batch of the validation set to ensure compatibility with the authors' code. We use the standard BiDO hyperparameters $(\lambda_x, \lambda_y) = (0.05, 0.5)$ both for our main results tables comparing baselines to AT-AT (Tables 23 and 25), as well as in our adversarial robustness experiments (Tables 11, 12, 13, 14). See App. D for our additional analysis of other alternative BiDO hyperparameter values.

- **Transfer Learning Defense against Model Inversion (TL-DMI) (Ho et al., 2024).** When training target models, defenses typically perform full fine-tuning on $\mathcal{D}_{priv}$ of a model pre-trained on some public dataset; TL-DMI instead fine-tunes only later layers of the target model, which is argued to reduce the number of parameters encoding sensitive information within the target model. We use the authors' original implementation with the minimal refactoring necessary to ensure compatibility with our codebase. We additionally remark that the authors of TL-DMI do not claim that the training data encoding that is relevant is necessarily rote-memorized training data, though practitioners widely assume that this is so. However, we note that it is conceivable that this encoded information the authors measure and discuss was unknowingly the very same non-robust features that we find to be operative here.

- **TL-DMI-5.** TL-DMI is typically implemented for ResNets by freezing the first 6 layers of the network. While we find that this parameter choice works well in many cases, we nonetheless find that it fails to obtain competitive test accuracy for on the CelebA dataset in certain cases (particularly when run on lower-capacity architectures like ResNet-18). Therefore, to give TL-DMI the best opportunity to perform well, we also consider an additional variant, TL-DMI-5, which freezes just 5 layers. We report this additional baseline wherever it offers a performance improvement over the recommended 6-layer version.

We note that we exclude the Sparse-Coding Architecture (SCA) defense (Dibbo et al., 2024) in our experiments, as it is not practical to run in the high-resolution scenario that is the focus of this work.

Second, for the sake of comparison, we also consider defenses that attempt to suppress gradients from MIAs, rather than increase generic privacy. Again, as above, we set all parameters identically to those used by their authors, and run defenses using the authors' replication codes:

- **Negative Label Smoothing (NegLS) (Struppek et al., 2024).** The NegLS defense employs negative label smoothing to make the target model consistently overconfident in its predictions. This 'deliberate overconfidence' does not remove information about the training data from the model. Instead, this overconfidence inhibits PPA attacks (and similar gradient-based GAN attacks, including IF-GMI) that rely on a smooth landscape of target model confidence scores to compute the gradient updates they use in their reconstruction steps. We use the authors' original implementation with the minimal refactoring necessary to ensure compatibility with our codebase.

- **Removal of Last Stage Skip-Connection (RoLSS) (Koh et al., 2024).** To impede gradient flow during optimization of PPA and other similar attacks (including IF-GMI), RoLSS proposes to remove the skip connection at the last stage of the target model. We use the skip-customized ResNet architectures provided in the RoLSS replication codes for RoLSS and RoLSS-SSF models. For the ResNet-18 RoLSS model on FaceScrub, we use a lower initial learning rate of 0.0001, as we found it to provide a stronger defense at similar accuracy, compared to the typical learning rate of 0.001.

- **Skip-Connection Scaling Factor (RoLSS-SSF) (Koh et al., 2024).** This variant of RoLSS weakens the weight of the last-layer skip connection instead of removing it completely. We use the standard scaling factor $k = 0.2$ (where $k = 1$ corresponds to an unchanged model, and $k = 0$ is equivalent to non-SSF RoLSS). For the ResNet-18 RoLSS-SSF model on FaceScrub, we use a lower initial learning rate of 0.0001, as we found it to provide a stronger defense at similar accuracy, compared to the typical learning rate of 0.001.

- **Trapdoor-based Defense (Trap-MID) (Liu & Chen, 2024).** The Trap-MID defense implements a trapdoor integrated into the model to predict a specific label when the input is injected with the corresponding trigger. The trapdoor information serves as a *shortcut* for MI attacks, leading them to extract trapdoor triggers rather than private data. We run two sets of hyperparameters suggested by the authors: Trap-MID denotes their tuned hyperparameters for high-resolution experiments ($\alpha = 0.007, \beta = 0.5$), and Trap-MID* denotes the hyperparameters for their lower-resolution experiments ($\alpha = 0.02, \beta = 0.2$), which we add for additional comparison. We use the original authors' code with minimal refactoring for compatibility with our codebase.

## A.2 Datasets

We consider the three datasets that are the focus of recent experiments on high-definition MIAs:

- **FaceScrub (Ng & Winkler, 2014).** FaceScrub is a commonly used facial dataset of 530 celebrity classes (see e.g. Struppek et al. (2024); Koh et al. (2024); Ho et al. (2024); Qiu et al. (2024); Struppek et al. (2022)). For ease of reproducibility, we use a popular Kaggle-hosted version[1] of the dataset and supply automatic download and processing code.

- **CelebA (Liu et al., 2015).** CelebA is a facial celebrity dataset containing 202,599 images of 10,177 celebrities. Following previous work (see e.g. Struppek et al. (2024); Liu & Chen (2024); Qiu et al. (2024); Struppek et al. (2022)), we choose the most common 1000 class identities out of the 10,177 available, for a total of 30,038 images. As is standard, we apply the CelebA HD Cropper[2] to improve the overall image quality.

- **Stanford Dogs (Khosla et al., 2011).** Stanford Dogs is a dog image dataset containing 20,580 images of 120 different species of dogs, commonly used in model inversion (see e.g. Ho et al. (2024); Koh et al. (2024); Struppek et al. (2022); Qiu et al. (2024)). We use the full dataset[3] for our experiments and include automatic download and processing code.

Each dataset was partitioned into 90% training and 10% test. From the training set, 20% was used as a validation set for tracking performance and early stopping. When training target models, we strictly follow the standard MIA data pre-processing steps (see e.g. Struppek et al. (2022; 2024), as follows: we normalize the images with $\mu = \sigma = (0.5, 0.5, 0.5)$, then resize so that the smaller side of the image is 224 pixels. We apply random cropping with patch sizes between 85% and 100%,

---

[1] Available at https://www.kaggle.com/datasets/rajnishe/facescrub-full

[2] Available at https://github.com/LynnHo/HD-CelebA-Cropper

[3] Available at http://vision.stanford.edu/aditya86/ImageNetDogs/

keeping the aspect ratio fixed at 1.0; we then resize the patches to $224 \times 224$ pixels (RandomResized-Cropping). We then applied random color jitter with saturation and hue factors of 0.1 and contrast and brightness factors of 0.2. Half of all samples are also horizontally flipped.

### A.3 Target model training details

Unless otherwise mentioned, we train target models with the following parameters, following previous work (see e.g. Struppek et al. (2024; 2022)): Before training on $\mathcal{D}_{priv}$, we initialize the models with pre-trained ImageNet weights available through Torchvision. We use the Adam optimizer with $\beta = (0.9, 0.999)$, and batch size of 128. Models are trained for a total of 100 epochs, saving the model weights of the epoch with the lowest validation loss. We measure prediction accuracy on the validation split throughout model training, and after training, we measure prediction accuracy on the test split. We use an initial learning rate of 0.001, and we employ a MultiStepLR learning rate scheduler with $\gamma = 0.1$. Full implementation details for target models are contained in our replication codes.

#### A.3.1 Permutation model training details

For the models trained with permuted labels in Sec. 3, we use a lower learning rate of 0.0001, which we found provided more stable training. All other training hyperparameters are identical to those described above. The target models with permuted labels achieve over $99.4\%$ accuracy on their respective altered training sets.

### A.4 Architectures

We follow recent MIA work by considering three architectures:

- **ResNet-152 (He et al., 2015).** As in numerous recent works on high-resolution MIAs (e.g., Struppek et al. (2022); Qiu et al. (2024); Koh et al. (2024); Struppek et al. (2024)), our main focus is ResNet-152 models, which exhibit sufficient depth to act as a good proxy for mainstream production vision models. Also, the adversarial examples literature has shown that models of higher capacity can better articulate robust and non-robust features well (Ilyas et al., 2019; Madry et al., 2017), so the ResNet-152 architecture is a natural choice for testing our hypotheses regarding non-robust features.

- **ResNet-18 (He et al., 2015).** As in recent MIA work (e.g., Struppek et al. (2022); Ho et al. (2024); Qiu et al. (2024)), we consider a ResNet-18 architecture, which permits us to probe how well our results generalize even to lower-capacity networks with more limited ability to capture complex feature bases.

- **DenseNet-169 (Huang et al., 2017).** As in recent high-resolution MIA work (e.g., Struppek et al. (2022); Qiu et al. (2024); Koh et al. (2024); Liu & Chen (2024)), we also consider a higher-capacity DenseNet-169 architecture, which allows us to test how well our results generalize to other deep convolutional architecture families.

### A.5 Reconstruction attacks

We consider three recent attacks that obtain state-of-the-art performance for our high-definition vision setting:

- **Plug-and-Play Attack (PPA) (Struppek et al., 2022).** Through a novel three-stage inversion attack framework (latent vector sampling, optimization, and robust final selection), PPA leverages a pre-trained StyleGAN-2 (Karras et al., 2020) to allow for impressive and efficient attack performance in the high resolution setting. StyleGAN-2 was trained on the FFHQ dataset (Karras et al., 2019), which has no identity overlap with CelebA or Face-Scrub. For the Stanford Dogs experiments, the StyleGAN-2 model used was trained on the AFHQ dogs dataset (Choi et al., 2020). We adapt the original authors' code to ensure compatibility with our codebase. During optimization, samples were optimized for 50 steps for FaceScrub and CelebA target models, and 100 steps for Stanford Dogs. All other hyperparameters are identical to those used in previous work (Struppek et al., 2022; 2024).

**Table 6:** AT-AT hyperparameters for the models reported in the main tables. Columns $\varepsilon$, **Step Size**, and **LR** are $\times\,10^{-4}$. All hyperparameters not mentioned are identical to their corresponding baseline models. 'RN' is ResNet, 'DN' is DenseNet.

| Dataset | Arch. | $\lambda$ | #Steps | $\gamma$ | $\varepsilon$ | Step Size | LR |
|---|---|---|---|---|---|---|---|
| FaceScrub | RN-152 | 0.040 | 3 | 0.4 | 1.5 | 0.5 | 4 |
| | RN-18 | 0.125 | 5 | 0.1 | 5 | 1 | 1 |
| | DN-169 | 0.100 | 3 | 0.4 | 2 | 0.7 | 4 |
| CelebA | RN-152 | 0.020 | 3 | 0.4 | 1 | 0.34 | 5 |
| | RN-18[†b] | 0.040 | 4 | 0.4 | 3 | 0.75 | 10 |
| | RN-18[†a] | 0.010 | 5 | 0.1 | 50 | 10 | 10 |
| | DN-169 | 0.020 | 3 | 0.4 | 3 | 1 | 5 |
| Stanford Dogs | RN-152 | 0.080 | 3 | 0.1 | 5 | 2 | 1 |

*Notes:* [†a] and [†b] are two distinct sets of hyperparameters for our AT-AT ResNet-18 model trained on CelebA, corresponding to the same symbols used in other tables.

- **Intermediate Features enhanced Generative Model Inversion (IF-GMI) (Qiu et al., 2024)** IF-GMI leverages intermediate features of StyleGAN-2 (Karras et al., 2020) to improve the expressiveness and transferability of the generated images, becoming the new SOTA for high-resolution MIAs. We adapt the original authors' code to ensure compatibility with our codebase. All attack hyperparameters are identical to those used by the original authors.

- **Pseudo-Private Data Guided Model Inversion Attack (PPDG) (Peng et al., 2024).** The PPDG authors find that a fixed generative prior, StyleGAN-2 (Karras et al., 2020), leads to a low probability of sampling actual private data during the inversion process due to the distributional gap to the private data distribution. To address this limitation, they increase the density around high-quality pseudo-private data that exhibit characteristics of the private training data, and slightly tune the generator. All attack hyperparameters are identical to those used by the original authors.

PPA and IF-GMI attacks on ResNet-152 and ResNet-18 models trained on FaceScrub and Stanford Dogs target all classes unless otherwise specified; for all models trained on CelebA, all PPDG attacks, and all DenseNet-169 results, we instead target a seeded random subset of 75 of the classes, holding the randomly selected classes constant for each dataset across all baselines. As attack time and compute scale roughly linearly with the number of classes targeted, this approach reduces time and compute by over 90% for our CelebA experiments and over 80% for the relevant FaceScrub experiments, at the cost of a small increase in uncertainty (see the confidence interval margins of error in Tables 27 and 28).

As described in Section 2, we evaluate these attacks using the following standard metrics: Attack Acc@1 (**AttAcc@1**), Attack Acc@5 (**AttAcc@5**), L2-FaceNet Distance (**L2**), and Average Evaluation Confidence (**EvalConf**).

### A.6 GPUS AND CARBON EMISSIONS

We run the majority of our target model training on 1x or 2x NVIDIA RTX A6000 GPUs with 48 GiB of vRAM each. The main attack experiment results were generated primarily through renting virtual machines from Lambda Labs, typically using 1x or 2x NVIDIA A100 GPUs with 80 GiB vRAM, or 1x or 2x NVIDIA H100 GPUs with 80 GiB vRAM. We estimate our total carbon emissions from GPU usage to be roughly 1200 kg of $CO_2$ eq., estimated using the Machine Learning Emissions Calculator (Lacoste et al., 2019).

## B   AT-AT HYPERPARAMETERS AND TRAINING

Our AT-AT scheme, described in Section 5, has multiple tunable hyperparameters; see Table 6 for details. **$\lambda$** governs the relative strength of the anti-AT loss term. For the in-loop generation of adversarial examples, a Projected Gradient Descent (PGD) adversarial attack (see e.g. Madry et al. (2017)) is used; **$\varepsilon$** is the size of the $L_\infty$ ball around the target image permissible for adversarial example generation, **#Steps** is the number of iterations used by the PGD attack, and **Step Size**

**Table 7:** Comparison of AT-AT FaceScrub ResNet-152 models at different $\lambda$ values. All other hyperparameters are the same as reported for ResNet-152 FaceScrub in Table 6. PPA is performed on a seeded random subset of 75 FaceScrub classes. The 4 rightmost columns measure AutoAttack robust accuracy under various $\epsilon$ levels. The $\lambda = 0.04$ model is the same AT-AT model reported in our main results; for this model, the PPA-AttAcc@1 value (marked by *) is on all 530 FaceScrub classes, as opposed to 75.

| Defense | $\lambda$ | TestAcc ↑ | PPA-AttAcc@1 ↓ | $\epsilon$=0.031↑ | $\epsilon$=0.0025↑ | $\epsilon$=0.0005↑ | $\epsilon$=$10^{-5}$↑ |
|---|---|---|---|---|---|---|---|
| | 0.01 | 0.936 | 0.099 | 0 | 0 | 0 | 0.307 |
| | 0.04 | 0.939 | 0.058* | 0 | 0 | 0 | 0.119 |
| AT-AT | 0.06 | 0.931 | 0.049 | 0 | 0 | 0 | 0.077 |
| | 0.10 | 0.921 | 0.064 | 0 | 0 | 0 | 0.146 |
| | 0.15 | 0.899 | 0.071 | 0 | 0 | 0 | 0.036 |

is the $L_\infty$ distance the PGD attack moves in one iteration before clipping, which we set to be $\gtrsim \varepsilon/(\textbf{\#Steps})$. We also tune the learning rate schedule; we report the initial learning rate **LR** (same as $\alpha$ in the Section 5 algorithm) and the decay factor $\gamma$ of the MultiStepLR scheduler. Full implementation and hyperparameter details are available in our replication codes.

In our preliminary experiments, we found that as AT-AT models begin to rely on non-robust features, training often becomes unstable, resulting in occasional downward spikes in model accuracy in the later epochs of training. We also note that AT-AT occasionally fails to converge when data augmentations are not applied. To account for this instability, we perform a variant of early stopping: We save AT-AT models during training at each epoch where they hit a certain pre-set validation accuracy range (e.g. over 90% for FaceScrub models). Notably, all other defenses are also using the corresponding best model checkpoint from the 100 epochs of training, i.e the model with the lowest validation loss. We also note that during AT-AT training, when model accuracy is unstable in later epochs, the reliance of the training model on non-robust features remains reliably high. See Table 7 for an assessment of AT-AT performance at five different values of $\lambda$, holding other hyperparameters constant.

### B.1 Deferred discussion of AT-AT practical runtimes

**AT-AT**'s runtimes are roughly 30% faster than vanilla AT trained via PGD, because AT requires resistance to strong (high-$\varepsilon$) attacks, whereas AT-AT seeks to learn from imperceptibly subtle (low-$\varepsilon$) features, requiring many fewer iterations of PGD. Specifically, a typical research implementation of AT uses a default of 10 steps of PGD, whereas we used just 3 to 5 steps for AT-AT throughout our experiments. These PGD step counts are the practical runtime bottleneck. To be concrete, this translates to runtimes of 3.10 hours (AT) vs. 2.14 hours (AT-AT) for CelebA on ResNet-18 architectures (note that all algorithms train for exactly 100 epochs throughout our experiments). Importantly, like AT, our AT-AT could potentially also benefit from the same well-studied algorithmic and implementation speedups that are widely applied in industry-scale AT implementations (e.g. Wong et al., 2020; Shafahi et al., 2019). As such, AT-AT can be implemented at scale at significantly lower cost than the AT implementations already in widespread industry use.

## C MIAs: notation and related work

The objective of model inversion attacks in image classification models is to generate images semantically similar to the class identities in the training data, causing a leakage of privacy. App. C.1 sets notation used throughout the paper and describes some early MIAs. App. C.2 describes more recent generative MIAs. App. C.3 clarifies the scope of our paper and describes some related inversion attack settings.

### C.1 Notation and early MIAs

Formally, MIAs consider a private training dataset $\mathcal{D}_{priv} = \{x_i, y_i\}_{i=1}^{N}$, where $x_i \in \mathbb{R}^{d_X}$, $y_i \in \{0, 1\}^C$, and each $(x_i, y_i)$ are sampled i.i.d. from a joint distribution $\mathbb{P}_{XY}$. Early MIAs used SGD to directly optimize reconstructions of individual training examples (Fredrikson et al., 2015). However, this approach suffers from various optimization complexities when applied to contemporary vision architectures like deep CNNs. Therefore, recent state-of-the-art attacks simplify

optimization by adopting an 'image prior'—that is, by restricting the search space of training example reconstructions to the image outputs of a domain-relevant GAN, and conducting SGD-based optimization on the GAN's input noise vectors instead of directly on the high-dimensional images' feature space, as first introduced by Zhang et al. (2020b). More specifically, GAN-based MIAs optimize a latent noise vector $\hat{z} \in Z$ such that the resultant GAN-generated image is classified as class $c$ maximized confidence by the target model $T$:

$$\min_{\hat{z}} \mathcal{L}(T, G, D, \hat{z}, c)$$

where $\mathcal{L}$ is a standard loss function (e.g. cross-entropy) to maximize the confidence of the GAN-generated image on class $c$ given pretrained GAN generator $G : Z \to X_{prior}$ and GAN discriminator $D$.

## C.2 RECENT GENERATIVE MIAS

In the following years after the introduction of generative MIAs by Zhang et al. (2020b), low-resolution generative MIAs saw numerous improvements (see e.g. Chen et al. (2021); Wang et al. (2022); Yuan et al. (2023); Nguyen et al. (2023b)). While these attacks show impressive performance in the low-resolution scenario, they have yet to prove effective on higher-resolution data.

MIRROR (An et al., 2022) became the first attack to have success in the high-resolution scenario. PPA (Struppek et al., 2022) made further improvements, proposing a framework that supports the use of a pre-trained StyleGAN model (Karras et al., 2020) unspecific to the target model $T$, which increases robustness to distributional shifts and flexibility in choice of target model while offering impressive performance on high-resolution image data. Most recently, IF-GMI (Qiu et al., 2024) leverages intermediate features of StyleGAN to improve the expressiveness and transferability of the generated images, becoming the new SOTA for high-resolution MIAs.

## C.3 INVERSION ATTACKS IN OTHER SETTINGS

While the mentioned MIAs are all white-box attacks, there has been some exploration of black-box and label-only model inversion attacks (Liu et al., 2024; Han et al., 2023; Nguyen et al., 2023a; An et al., 2022; Yang et al., 2019; Kahla et al., 2022; Zhu et al., 2023; Qiu et al., 2025) and defenses (Zhuang et al., 2025; Zhu et al., 2023). It should be noted that while it has previously been shown that some recent black-box and label-only attacks have outperformed previous white-box attacks (Zhuang et al., 2025; Han et al., 2023; Nguyen et al., 2023a), these performance comparisons had been limited to earlier, low-resolution MIAs (Chen et al., 2021; Zhang et al., 2020b); more recent, high-resolution white-box MIAs (Qiu et al., 2024; Struppek et al., 2022) are likely to outperform black-box and label-only MIAs, which have primarily been evaluated on low-resolution data regimes (Han et al., 2023; Nguyen et al., 2023a). As such, for the purposes of this paper, we focus on the more difficult white-box scenario.

Separately, a related stream of research studies gradient inversion attacks and defenses in the federated learning scenario, where neural networks are collaboratively trained on a server (Fang et al., 2023; Huang et al., 2021; Geiping et al., 2020). While both gradient inversion attacks and modern MIAs share the goal of leaking private training data, the threat models are fundamentally different; gradient inversion attacks often seek per-image or per-batch reconstructions based on gradient information provided by a local user during collaborative training, while modern MIAs instead seek class-representative reconstructions based on queries to an already-trained neural network. As such, we leave the impact of information dependency and adversarial robustness on gradient inversion attacks as an important direction for future research.

## D  HSIC EXTENDED RESULTS

Table 8 is an extended version of Table 1 which includes ResNet-18 models, as well as the extra Trap-MID* baseline. The results are similarly uncorrelated; some defenses, like the ResNet-18 RoLSS-SSF model, have some low HSIC values but perform poorly against a PPA attack, while other defenses, like ResNet-152 NegLS, have similarly high HSIC values to an undefended model, but perform well against a PPA attack. It should be noted that BiDO utilizes an HSIC *estimator* and that in some problems optimized HSIC tests outperform optimizing the statistic (Xu et al., 2025).

**Table 8:** HSIC($X, Z_j$) for different defenses on **FaceScrub**. Models: **ResNet-152** and **ResNet-18**. AttAcc@1 is from a PPA attack on all classes.

| Arch. | Defense | TestAcc ↑ | AttAcc@1 ↓ | HSIC($X, Z_0$) | HSIC($X, Z_1$) | HSIC($X, Z_2$) | HSIC($X, Z_3$) |
|---|---|---|---|---|---|---|---|
| ResNet-152 | NoDef | 0.958 | 0.882 | 71.66 | 72.32 | 72.72 | 72.72 |
| | MID | 0.950 | 0.391 | 71.52 | 72.57 | 72.84 | 61.98 |
| | BiDO | 0.928 | 0.780 | 73.68 | 73.68 | 73.68 | 73.63 |
| | TL-DMI | 0.911 | 0.190 | 71.73 | 72.52 | 72.93 | 72.58 |
| | NegLS | 0.906 | 0.160 | 71.64 | 72.33 | 72.29 | 73.63 |
| | RoLSS | 0.886 | 0.474 | 72.09 | 72.30 | 29.69 | 61.92 |
| | RoLSS-SSF | 0.869 | 0.343 | 71.88 | 72.02 | 32.75 | 64.94 |
| | Trap-MID* | 0.949 | 0.587 | 69.93 | 71.96 | 72.56 | 72.65 |
| | Trap-MID | 0.926 | 0.401 | 69.80 | 71.88 | 72.62 | 69.28 |
| ResNet-18 | NoDef | 0.950 | 0.901 | 63.48 | 49.74 | 37.80 | 72.38 |
| | MID | 0.924 | 0.714 | 61.03 | 48.39 | 30.08 | 65.73 |
| | BiDO | 0.884 | 0.530 | 73.80 | 73.81 | 73.80 | 73.75 |
| | TL-DMI | 0.906 | 0.240 | 61.11 | 50.29 | 36.31 | 72.64 |
| | NegLS | 0.929 | 0.649 | 63.27 | 50.76 | 34.97 | 73.68 |
| | RoLSS | 0.950 | 0.858 | 61.23 | 49.38 | 13.33 | 72.25 |
| | RoLSS-SSF | 0.950 | 0.849 | 61.44 | 49.38 | 13.30 | 72.27 |
| | Trap-MID* | 0.927 | 0.502 | 57.84 | 47.80 | 34.93 | 72.01 |
| | Trap-MID | 0.928 | 0.735 | 62.44 | 47.33 | 34.91 | 67.44 |

App. D.1 revisits BiDO hyperparameter tuning, exploring the possibility of creating a model with low HSIC values with the BiDO scheme.

## D.1 BiDO EXPERIMENTS

Let $X$ and $Y$ be random variables representing the inputs and outputs, respectively, and let $Z_1, \ldots, Z_m$ be intermediate embeddings chosen at $m$ different points from passing $X$ as input to the model (e.g. the outputs of the four major blocks of a ResNet). Motivated by theoretical concepts in statistical dependency (Gretton et al., 2005), the popular BiDO defense Peng et al. (2022) uses the following training loss function:

$$\tilde{\mathcal{L}}(\theta) = \mathcal{L}(\theta) + \lambda_x \sum_{j=1}^{m} \text{HSIC}(X, Z_j) - \lambda_y \sum_{j=1}^{m} \text{HSIC}(Z_j, Y)$$

where $\mathcal{L}$ is a standard cross-entropy loss and $\lambda_x, \lambda_y > 0$ are hyper-parameters. The $\lambda_x$ term is intended to minimize dependency between inputs and intermediate embeddings. As for the $\lambda_y$ term, Peng et al. (2022) argue that solely minimizing HSIC($X, Z_j$) would "...lead to the loss of useful information, so it is necessary to keep the [$\lambda_y$] term to make sure $Z_j$ is informative enough of $Y$ as well as to maintain the discriminative nature of the classifier."

After the surprising discovery that BiDO with $(\lambda_x, \lambda_y) = (0.05, 0.5)$ fails to decrease HSIC($X, Z_j$) (see Table 8), we experimented with other pairs $(\lambda_x, \lambda_y)$ to see if HSIC($X, Z_j$) (referred to hereafter as 'HSIC') could indeed be lowered through the BiDO training scheme. To check whether the $\lambda_y$ term of the objective function was impeding the model's ability to reduce HSIC, we first experimented with setting $\lambda_y = 0$; however, according to the ablation study of Peng et al. (2022), setting $(\lambda_x, \lambda_y) = (0.1, 0)$ completely impedes the model's ability to train. We instead surprisingly find success with smaller values of $\lambda_x$; more specifically, we train ResNet-18 and ResNet-152 BiDO models on FaceScrub with $(\lambda_x, \lambda_y) = (0.001, 0)$ and $(\lambda_x, \lambda_y) = (0.0001, 0)$. We use initial learning rates of 0.001 and 0.0001 for the ResNet-152 and ResNet-18 models, respectively. For the ResNet-18 models, we train for just 75 epochs, as this was sufficient for them to converge; for the ResNet-152 models, we train for 100 epochs. All other training hyperparameters are identical to those described in App. A.3. As a natural next step, we run a PPA attack on our new 'low-HSIC' BiDO models. Due to compute considerations, we attack a seeded random subset of 75 out of the 530 FaceScrub identities; all other attack hyperparameters are identical to those described in App. A.5.

**Results.** As Table 9 shows, across both architectures, BiDO(0.001, 0) and BiDO(0.0001, 0) generally succeed in lowering HSIC($X, Z_j$) for most values of $j$ while losing only 0.1 to 1.8 percentage points in test accuracy. However, these 'low-HSIC' BiDO models fail to provide a meaningful defense against model inversion, with top-1 attack accuracies consistently exceeding 80%. In sum,

Table 9 reinforces the notion that the HSIC dependency measure used in BiDO, while theoretically well-motivated, is orthogonal to model inversion robustness.

**Table 9:** $HSIC(X, Z_j)$ for different BiDO hyperparameter variations, with comparisons to NoDef, MID, and TL-DMI. ResNet-18 and ResNet-152 models trained on FaceScrub. AttAcc@1 is from PPA on all 530 classes, except for the four BiDO models with $\lambda_y = 0$, for which PPA is run on a seeded random subset of 75 classes out of 530. The ResNet-152 BiDO(.001,0) model is also reported in Table 1 as BiDO**.

| Arch. | Defense | TestAcc ↑ | AttAcc@1 ↓ | $HSIC(X, Z_0)$ | $HSIC(X, Z_1)$ | $HSIC(X, Z_2)$ | $HSIC(X, Z_3)$ |
|---|---|---|---|---|---|---|---|
| RN-152 | NoDef | 0.958 | 0.882 | 71.66 | 72.32 | 72.72 | 72.72 |
| | MID | 0.950 | 0.391 | 71.52 | 72.57 | 72.84 | 61.98 |
| | BiDO(.05,.5) | 0.928 | 0.780 | 73.68 | 73.68 | 73.68 | 73.63 |
| | BiDO(.001,0) | 0.940 | 0.815 | 2.159 | 1.256 | 9.310 | 71.81 |
| | BiDO(.0001,0) | 0.948 | 0.849 | 70.62 | 72.15 | 72.71 | 72.68 |
| | TL-DMI | 0.911 | 0.190 | 71.73 | 72.52 | 72.93 | 72.58 |
| RN-18 | NoDef | 0.950 | 0.901 | 63.48 | 49.74 | 37.80 | 72.38 |
| | MID | 0.924 | 0.714 | 61.03 | 48.39 | 30.08 | 65.73 |
| | BiDO(.05,.5) | 0.884 | 0.530 | 73.80 | 73.81 | 73.80 | 73.75 |
| | BiDO(.001,0) | 0.943 | 0.817 | 4.403 | 1.402 | 1.511 | 72.34 |
| | BiDO(.0001,0) | 0.949 | 0.834 | 54.51 | 41.08 | 24.83 | 72.37 |
| | TL-DMI | 0.906 | 0.240 | 61.11 | 50.29 | 36.31 | 72.64 |

# E 97% UNSEEN-PIXEL MODEL EXPERIMENT DETAILS

This section provides further details for our Unseen-Pixels experiment in section 3. Appendix E.1 describes related work in feature selection for neural networks; Appendix E.2 then describes our method for creating a mask that deletes over 97% of pixels from the training data before training the model, while losing only a few percentage points in accuracy. Appendix E.3 describes the relationship between recent theoretical guarantess for data reconstruction attacks and our Unseen-Pixels method; Table 10 presents additional Unseen-Pixels results against an IF-GMI attack.

## E.1 BACKGROUND AND RELATED WORK ON FEATURE SELECTION IN SHALLOW NEURAL NETWORKS

Feature selection within the training of shallow fully-connected neural networks is well-studied, particularly with techniques derived from the popular Lasso regularization (Tibshirani, 1996) on linear models. In particular, LassoNet (Lemhadri et al., 2021) proposes to insert a skip connection directly from the inputs to the output in a feedforward neural network, then imposes a Lasso penalty on the skip connection weights to the loss function during training, such that some of these weights would be zeroed out. The features would then be weighted in the main network corresponding to their weight in the skip connection, which created the desired feature selection. A parallel approach has been to apply a Group Lasso (or $L_{2,1}$ norm) penalty directly to the weights $\boldsymbol{W}$ of the initial fully connected layer (Zhang et al., 2020a; Scardapane et al., 2017; Zhao et al., 2015). Let $\boldsymbol{w}_i$ be the $i$-th column vector of $\boldsymbol{W}$; the Group Lasso penalty is then $\phi(\boldsymbol{W}) = \sum ||\boldsymbol{w}_i||_2$. Adding this penalty to the loss function then allows for the zeroing out of unhelpful groups of weights in this initial matrix; each of these groups of weights correspond with an input feature. Building on this idea, the "GroupLasso+AdaptiveGroupLasso" (GL+AGL) technique (Dinh & Ho, 2021) first calculates a Group Lasso-regularized version of the weights $\boldsymbol{W}$, then uses this as a "rough data-dependent estimate to shrink groups of parameters with different regularization strengths", creating a more aggressive and consistent feature selector (Dinh & Ho, 2020). However, all of the above approaches

**Table 10:** Unseen-Pixels trained on FaceScrub obtains very limited defense against reconstruction by IF-GMI.

| Arch. | Defense | %Pix-deleted | TestAcc ↑ | AttAcc@1 ↓ | AttAcc@5 ↓ | L2-Face ↑ | EvalConf ↓ |
|---|---|---|---|---|---|---|---|
| RN-152 | NoDef | 0 | 0.958 | 0.987 | 0.998 | 0.700 | 0.984 |
| | NoDef-EarlyStop | 0 | 0.917 | 0.886 | 0.981 | 0.772 | 0.875 |
| | TL-DMI | 0 | 0.911 | 0.420 | 0.694 | 0.946 | 0.401 |
| | **Unseen-Pixels** | **97.8** | 0.910 | 0.860 | 0.966 | 0.770 | 0.848 |
| RN-18 | NoDef | 0 | 0.950 | 0.983 | 0.996 | 0.740 | 0.979 |
| | NoDef-EarlyStop | 0 | 0.890 | 0.897 | 0.980 | 0.757 | 0.881 |
| | TL-DMI | 0 | 0.906 | 0.515 | 0.760 | 0.935 | 0.492 |
| | **Unseen-Pixels** | **97.2** | 0.892 | 0.797 | 0.935 | 0.821 | 0.780 |

rely on the existence of a set of weights in the neural network that correspond to each feature; in the deep CNNs that are the focus of our paper, such weights do not exist, and so these approaches cannot be readily applied.

As an alternative approach, Li et al. (2016) propose prepending sparse one-to-one layer $D$, of the same dimensions as a member of the training data, to the model; the forward pass is done through elementwise multiplication. During training, an Elastic-Net (Zou & Hastie, 2005) penalty is applied to the weights $D$; features are removed when their corresponding weight in $D$ vanishes. Other similar approaches include *CancelOut* (Borisov et al., 2019) and *Feature-Aware Drop Layer* (Jiménez-Navarro et al., 2024). While the above approaches are all compatible with convolutional networks, none of their empirical analyses explores the hi-res image classification scenario with deep convolutional networks that is the subject of our paper.

### E.2 A GENERAL METHOD FOR FEATURE SELECTION IN DEEP CNNS

To achieve our goal of creating a mask that selects only a small subset of pixels from high-resolution training data images without compromising on model utility, we use a combination of many of the above approaches. We prepend a one-to-one *mask layer* to the front of the CNN and perform a variant of GL+AGL regularization (Dinh & Ho, 2021) on this layer. More specifically, let the mask layer weights be a matrix $D$ of same shape as the input images (in our experimental setting, shape $3 \times 224 \times 224$). Then, our new model $T_{mask}$ is simply the normal model $T$ after an elementwise multiplication with $D$: $T_{mask}(X) = T(X \odot D)$.

All of our datasets are RGB images, and as such, each weight of $D$ corresponds to *one RGB channel* of a pixel, as opposed to a whole pixel. Let $D_{ij}^c$ represent the channel for color $c$ for pixel at position $(i, j)$ in the image. For visual ease, we seek to select *pixels*, not channels of pixels; to do so, we will apply a variant of GL+AGL on the group of three channels for each pixel. In our experimental settings, this creates groups $\langle D_{ij}^R, D_{ij}^G, D_{ij}^B \rangle$ for each pair $(i, j)$ where $0 \leq i, j \leq 223$, for a total of $224 \times 224 = 50176$ groups. Let $D^{Norm}$ be a $224 \times 224$ matrix where $D_{ij}^{Norm} = ||\langle D_{ij}^R, D_{ij}^G, D_{ij}^B \rangle||_2$.

To create the mask we use to delete training data pixels prior to training our model, we proceed in two training stages: (1) Ridge base estimator creation and (2) GL+AGL feature selection. In Stage 1, to generate the base estimator needed for GL+AGL, we apply a Ridge regression to the grouped weights of $D$:

$$\tilde{\mathcal{L}}(\theta) = \mathcal{L}(\theta) + \lambda_1 \cdot ||D^{Norm}||_2.$$

We train in Stage 1 for 100 epochs and save the resulting mask layer $\hat{D}$ as a base estimate for Stage 2, discarding the rest of the trained model. For Stage 2, we apply the penalty formula given in Dinh & Ho (2020) with our ridge base estimator:

$$\tilde{\mathcal{L}}(\theta) = \mathcal{L}(\theta) + \lambda_2 \cdot \mathcal{M}(D),$$

where

$$\mathcal{M}(D) = \sum_{i}^{224} \sum_{j}^{224} \frac{D_{ij}^{Norm}}{(\hat{D}_{ij}^{Norm})^\rho}.$$

During Stage 2, after each gradient update, we apply a threshold $t$ to $D$ such that if $D_{ij}^{Norm} \leq t$, then $\langle D_{ij}^R, D_{ij}^G, D_{ij}^B \rangle$ is set to $\mathbf{0}$ within $D$, dropping the pixel at position $(i, j)$ from model training. We begin with $t = $ 1e-6, and after epoch 10, we set $t = 0.0025$.

We train in Stage 2 for 100 epochs and save the resulting trained mask $D^*$, discarding the rest of the trained model. $D^*$ is then altered such that any nonzero element is set to 1, so that elementwise multiplication by $D^*$ has no additional effect besides dropping the pixels it has learned to mask. This completes the creation of the mask $D^*$.

Finally, the true target model is trained with the mask $D^*$ applied. The result of this 'true' training run is the 'Unseen-Pixels' model reported in Table 3. $D^*$ is applied as a pre-augmentation data transform, ensuring that no data augmentations allow otherwise-deleted pixels to influence model training. This training run is done with identical hyperparameters and data augmentations to those described for target models in App. A.

In Figure 1, the mask applied is $D^*$ for our ResNet-152 Unseen-Pixels model. The reconstructions in the third column are the result of the same PPA attack on this ResNet-152 Unseen-Pixels model reported in Table 3.

$\lambda_1$ controls the relative strength of the ridge penalty in Stage 1, and $\lambda_2$ controls the relative strength of the AGL penalty in Stage 2. $\rho$ regulates the influence of the base estimator during Stage 2. For both ResNet-18 and ResNet-152, we set $\lambda_1 = 0.01$, $\lambda_2 = 0.1$, and $\rho = 1$. During Stages 1 and 2, we apply only brightness, saturation, and hue augmentations; random resized cropping, contrast augmentations, and random horizontal flipping are removed since their inclusion would allow otherwise-deleted pixels to influence mask training. All unmentioned training hyperparameters during Stage 1 and Stage 2 are identical to those described in App. A; full materials for reproduction of results are given in our replication codes.

This method of feature selection is primarily intended as a proof-of-concept to test the effectiveness of MIAs on a model trained on data with large amounts of hidden pixels. The creation of the mask $D^*$ involved many design choices with how to incorporate ridge vs. lasso penalties, whether to apply Group Lasso on its own or GL+AGL, and so on; we do not claim that our approach is the optimal one within this family of design choices, or that our hyperparameter choices for $\lambda_1$, $\lambda_2$, $t$, and $\rho$ are optimal. We leave this analysis as an interesting direction for future work in feature selection for deep CNNs.

Nevertheless, our approach succeeds in the creation of masks that allows us to delete over 97% of the training pixels in each image, such that they are never seen during training, while losing only a few percentage points in accuracy. The fact that this unseen data is still devastatingly reconstructed by MIAs calls into question the intuitive role of training-data-to-model dependencies in MIAs.

### E.3 Unseen-Pixels and recent theoretical privacy results

In this section, we describe recent theoretical guarantees for data reconstruction attacks that lower-bound the variance of unbiased estimators of training data, and we describe why no such estimator exists for the hidden pixels of our Unseen-Pixels models, suggesting that guarantees based on unbiased reconstruction estimators are overly optimistic for the MIA setting.

The recent works of Hannun et al. (2021) and Chaudhuri et al. (2024) seek to provide theoretical guarantees on training data privacy by framing the privacy leakage issue as a statistical parameter estimation problem, as follows: Let $\mathbf{z} = [x_1^\top, y_1, \ldots, x_N^\top, y_N] \in \mathbb{R}^{N \cdot (d_X + 1)}$ be formed by concatenating all of the examples in the training dataset $\mathcal{D}_{priv}$. Given a randomized learning algorithm $\mathcal{A}(\mathbf{z})$ that outputs a model in hypothesis space $\mathcal{H}$ after training on data $\mathcal{D}_{priv}$, we consider the probability distribution of possible models $h$ (or set of model outputs $h(x)$) to be parameterized by the dataset vector $\mathbf{z}$. The adversary then seeks to create an estimator $\hat{\mathbf{z}}$ based on the observation $h$ (or observations $h(x)$); a successful estimator would leak information about the dataset vector $\mathbf{z}$, creating a breach of privacy. To create theoretical privacy guarantees, both papers provide methods to lower-bound the variance of any unbiased estimator $\hat{\mathbf{z}}$; Hannun et al. (2021) introduce the concept using Fisher Information Loss and the Cramér-Rao bound, while Chaudhuri et al. (2024) use Hammersley-Chapman-Robbins bounds, which are tractable to compute for deeper convolutional networks.

In this statistical framing, the hidden pixels of our Unseen-Pixels models have no effect on the model or model outputs, and so cannot be inferred by an unbiased estimator. This is the strongest version of this privacy guarantee from the above papers; instead of a lower bound on variance, it is simply the case that there is *no* unbiased estimator of the hidden pixels (or, equivalently, unbiased reconstruction estimators of the hidden pixels have unbounded variance). However, as Fig. 1 shows, in spite of this strong privacy guarantee on over 97% of pixels in the training data images, MIAs are still able to cause significant privacy leakage.

This result suggests that privacy guarantees related to unbiased estimators $\hat{\mathbf{z}}$ may not provide adequate privacy protection in the context of MIAs, perhaps unless they can be applied to *all* pixels of the input images. We hypothesize this is because of the *strong prior* that contemporary MIAs use in the form of a pre-trained facial GAN. As mentioned in Chaudhuri et al. (2024), the requirement that $\hat{\mathbf{z}}$ is unbiased is "tantamount to forbidding the use of extra, outside information such as a Bayesian prior", and that unbiasedness is a "reasonable yet significant restriction". Our experiment under-

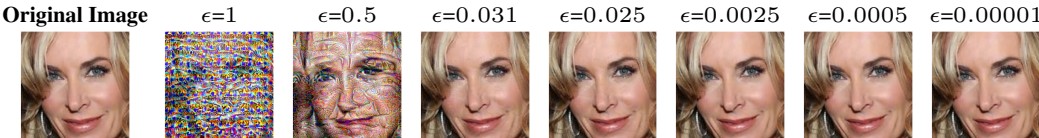

**Figure 5:** Adversarial examples (PGD attack) for the different perturbation magnitudes $\epsilon$.

scores the significance of this restriction, since the GAN prior of PPA is powerful enough to create faithful representations of faces of which less than 3% of the pixels have been seen by the target model.

We emphasize that our discussion is limited to the usefulness of privacy guarantees given by unbiased reconstruction estimators in the MIA scenario. The output of an MIA is not intended to be a pixel-by-pixel reconstruction of each individual image, but instead a class-representative image that captures the likeness of a certain private identity. It is likely that guarantees on unbiased reconstruction estimators have more success against other forms of data privacy attacks that do indeed seek exact reconstructions of individual data points, such as training data extraction from generative models (Carlini et al., 2023).

On a separate note, biased estimators are also discussed in similar work (Hannun et al., 2021; Guo et al., 2022); strong guarantees on biased estimators may be more applicable in the MIA experimental setting, as they can potentially account for the presence of strong prior information (Chaudhuri et al., 2024). We consider further exploration of the practical usefulness of privacy guarantees on biased reconstruction estimators to be an interesting area for future research.

Regardless, our results caution against the idea that a strong guarantee against unbiased reconstruction estimators yields universal confidentiality protections.

# F ADVERSARIAL ROBUSTNESS: EVALUATION DETAILS AND CONNECTIONS TO PRIVACY

## F.1 ADVERSARIAL EXAMPLES: ADDITIONAL BACKGROUND

Adversarial examples (Szegedy et al., 2014) are small noise perturbations that when added to an image in the dataset, changes the prediction of the model. The noise is restricted to be in a small range $||\delta||_p < \epsilon$, where $\delta$ is the perturbation, $p$ is the norm, and $\epsilon$ is the limit of the size of perturbations. Since the perturbations are limited to very small changes, they are mostly imperceptible for humans and don't carry semantic meaning. Figure 5 shows adversarial examples generated from a Project Gradient Descent (PGD) attack (Madry et al., 2017) at various $L_\infty$ perturbation magnitudes $\epsilon$.

The existence of adversarial examples was first discovered in Szegedy et al. (2014). Algorithms to generate them (Croce & Hein, 2020b) and defend against them (Madry et al., 2017; Shafahi et al., 2019; Wong et al., 2020) have since been heavily studied.

## F.2 ADVERSARIAL EXAMPLES AND PRIVACY: RELATED WORK

Our work is inspired by many previous studies on adversarial robustness (or non-robustness) and its connection to privacy. Tursynbek et al. (2021) and Boenisch et al. (2021) show how different aspects of differentially private training (Abadi et al., 2016) can make models vulnerable to adversarial attacks, while Zhang & Bu (2023) provide further analysis showing that differentially private models can indeed also be adversarially robust. Song et al. (2019) show empirically that adversarially robust deep learning models are often vulnerable to membership inference attacks, exploring

related factors such as model capacity and robustness generalization. Hayes (2020) explores this relationship further, describing the effect of the size of the adversarial perturbation used, as well as the size of the training set. Khaled et al. (2022) show that model extraction attacks are often more effective on adversarially trained models. We also consider our work related to the recent stream of evidence that more interpretable models are inherently less privacy-preserving (Naretto et al., 2022; Shokri et al., 2021).

Recent works also focus on the connection of adversarial examples to model inversion specifically. Mejia et al. (2019) show that adversarially trained models allow for the success of previously-intractable low-resolution model inversion attacks, arguing that the more semantically meaningful feature space that AT creates (Engstrom et al., 2019; Madry et al., 2017) enhances the performance of MIAs. Separately, a recent study on black-box MIAs (Zhou et al., 2024) incorporates adversarial examples into training data in order to increase the diversity of semantically meaningful information available to the adversary.

Wen et al. (2021) proposes to defend against black-box MIAs by adding carefully-chosen adversarial noise to the output of the target model, offering impressive performance. While this defense does not apply in the white-box scenario, as adversaries would have access to the output before the adversarial noise is applied, we nonetheless consider this work complementary to our results involving the effectiveness of **AT-AT**.

Our work is particularly inspired by the ideas presented in Mejia et al. (2019); we expand upon this work by 1) considering newer, high-resolution MIAs; 2) showing that recent generic MIA defenses obtain their performance through an unintended reliance on non-robust features; and 3) introducing a novel algorithm, **AT-AT**, that allows for a direct tradeoff between model inversion defense and adversarial robustness.

### F.3 ADVERSARIAL ROBUSTNESS EVALUATION: DEFERRED EXPERIMENTAL DETAILS

As explained by Carlini et al. (2019), adversarial robustness (and, therefore, adversarial vulnerability) is often improperly quantified. To mitigate this issue, Carlini et al. (2019) provide a checklist of actions to take to ensure proper evaluation of adversarial robustness, including performing adaptive attacks, applying a diverse set of attacks, and verifying that attacks have converged. In response to this advice, the AutoAttack ensemble (Croce & Hein, 2020b) has emerged as a popular benchmark to measure adversarial robustness due to its fixed hyperparameters, automatic convergence checks, and diversity of attacks. The four primary adversarial attacks in the ensemble are 1) untargeted Auto-PGD with cross-entropy loss; 2) targeted Auto-PGD with Difference of Logits Ratio (DLR) loss (APGD-T); 3) targeted Fast Adaptive Boundary (FAB-T) (Croce & Hein, 2020a); and 4) Square Attack (Andriushchenko et al., 2020). The two Auto-PGD variants are original to the AutoAttack paper (Croce & Hein, 2020b). We run AutoAttack with varying $L_\infty$ $\epsilon$-ball constraints on 1024 examples of the validation set for each of the reported target models.

### F.4 DEFERRED FULL RESULTS FROM AUTOATTACK AND ADVERSARIAL ROBUSTNESS EXPERIMENTS

Tables 11, 12, 13, 14 present the full results of the AutoAttack adversarial robustness experiments for $\epsilon \in \{0.031, 0.0025, 0.0005, 10^{-5}\}$, respectively. Specifically, for each of these values of $\epsilon$, the respective table reports robust accuracy of each defense on each dataset and architecture both (1) overall (i.e. worst-case), and (2) for each of standard AutoAttack's four constituent attacks: untargeted Auto-PGD; targeted DLR Auto-PGD (Croce & Hein, 2020b); targeted FAB (Croce & Hein, 2020a); and Square (Andriushchenko et al., 2020).

**Table 11:** AutoAttack robust accuracies with $\epsilon$=0.031. Attack is on 8 batches (1024 validation set examples).

| Data | Arch. | Defense | TestAcc ↑ | Overall ↑ | APGD-CE ↑ | APGD-T ↑ | FAB-T ↑ | Square ↑ |
|---|---|---|---|---|---|---|---|---|
| *Defenses that attempt to increase privacy generically by reducing information dependency/memorization* | | | | | | | | |
| FaceScrub | RN-152 | NoDef | 0.958 | 0 | 0 | 0 | 0 | 0.002 |
| | | MID | 0.950 | 0 | 0.009 | 0.009 | 0.169 | 0.888 |
| | | BiDO | 0.928 | 0 | 0 | 0 | 0 | 0.003 |
| | | TL-DMI | 0.911 | 0 | 0 | 0 | 0 | 0.006 |
| | | AT-AT | 0.939 | 0 | 0 | 0 | 0 | 0.038 |
| | RN-18 | NoDef | 0.950 | 0 | 0 | 0 | 0 | 0 |
| | | MID | 0.924 | 0 | 0.022 | 0.030 | 0.349 | 0.880 |
| | | BiDO | 0.884 | 0 | 0 | 0 | 0 | 0.004 |
| | | TL-DMI | 0.906 | 0 | 0 | 0 | 0 | 0.002 |
| | | AT-AT | 0.921 | 0 | 0 | 0 | 0 | 0.008 |
| CelebA | RN-152 | NoDef | 0.838 | 0 | 0 | 0 | 0 | 0.003 |
| | | MID | 0.802 | 0.004 | 0.069 | 0.086 | 0.306 | 0.705 |
| | | BiDO | 0.747 | 0 | 0 | 0 | 0 | 0.001 |
| | | TL-DMI | 0.688 | 0 | 0 | 0 | 0 | 0.001 |
| | | TL-DMI-5 | 0.814 | 0 | 0 | 0 | 0 | 0.005 |
| | | AT-AT | 0.819 | 0 | 0 | 0 | 0 | 0.021 |
| | RN-18 | NoDef | 0.859 | 0 | 0 | 0 | 0 | 0.001 |
| | | MID | 0.737 | 0.003 | 0.099 | 0.120 | 0.391 | 0.624 |
| | | BiDO | 0.625 | 0 | 0 | 0 | 0 | 0.003 |
| | | TL-DMI | 0.752 | 0 | 0 | 0 | 0 | 0.001 |
| | | TL-DMI-5 | 0.831 | 0 | 0 | 0 | 0 | 0.003 |
| | | AT-AT[†a] | 0.825 | 0 | 0 | 0 | 0 | 0.004 |
| | | AT-AT[†b] | 0.787 | 0 | 0 | 0 | 0 | 0.002 |
| *Defenses that suppress attack gradients* | | | | | | | | |
| FaceScrub | RN-152 | RoLSS-SSF | 0.869 | 0 | 0 | 0 | 0 | 0.005 |
| | | RoLSS | 0.886 | 0 | 0 | 0 | 0 | 0.005 |
| | | NegLS | 0.906 | 0 | 0.782 | 0 | 0 | 0.004 |
| | | Trap-MID | 0.927 | 0 | 0 | 0 | 0.023 | 0.001 |
| | | Trap-MID* | 0.948 | 0 | 0 | 0 | 0.023 | 0.023 |
| | RN-18 | RoLSS-SSF | 0.950 | 0 | 0 | 0 | 0 | 0.003 |
| | | RoLSS | 0.950 | 0 | 0 | 0 | 0 | 0.003 |
| | | NegLS | 0.929 | 0 | 0.815 | 0 | 0 | 0.002 |
| | | Trap-MID | 0.928 | 0 | 0 | 0 | 0.009 | 0.004 |
| | | Trap-MID* | 0.927 | 0 | 0 | 0 | 0.004 | 0.003 |
| CelebA | RN-152 | RoLSS-SSF | 0.428 | 0 | 0 | 0 | 0 | 0 |
| | | RoLSS | 0.537 | 0 | 0 | 0 | 0.001 | 0 |
| | | NegLS | 0.805 | 0 | 0.153 | 0 | 0 | 0.003 |
| | | Trap-MID | 0.848 | 0 | 0 | 0 | 0.068 | 0.005 |
| | | Trap-MID* | 0.806 | 0 | 0 | 0 | 0.023 | 0.002 |
| | RN-18 | RoLSS-SSF | 0.839 | 0 | 0 | 0 | 0 | 0.002 |
| | | RoLSS | 0.830 | 0 | 0 | 0 | 0 | 0.001 |
| | | NegLS | 0.838 | 0 | 0.006 | 0 | 0 | 0 |
| | | Trap-MID | 0.830 | 0 | 0 | 0 | 0.020 | 0.004 |
| | | Trap-MID* | 0.811 | 0 | 0 | 0 | 0.014 | 0.004 |

*Notes:* [†a] and [†b] denote two sets of hyperparameters for AT-AT (see Table 6).

**Table 12:** AutoAttack robust accuracies with $\epsilon$=0.0025. Attack is on 8 batches (1024 validation set examples).

| Data | Arch. | Defense | TestAcc ↑ | Overall ↑ | APGD-CE ↑ | APGD-T ↑ | FAB-T ↑ | Square ↑ |
|---|---|---|---|---|---|---|---|---|
| *Defenses that attempt to increase privacy generically by reducing information dependency/memorization* | | | | | | | | |
| FaceScrub | RN-152 | NoDef | 0.958 | 0.122 | 0.15 | 0.123 | 0.143 | 0.894 |
| | | MID | 0.950 | 0.006 | 0.047 | 0.079 | 0.223 | 0.942 |
| | | BiDO | 0.928 | 0.121 | 0.147 | 0.121 | 0.133 | 0.844 |
| | | TL-DMI | 0.911 | 0 | 0 | 0 | 0 | 0.334 |
| | | AT-AT | 0.939 | 0 | 0 | 0 | 0 | 0.101 |
| | RN-18 | NoDef | 0.950 | 0.194 | 0.231 | 0.194 | 0.215 | 0.883 |
| | | MID | 0.924 | 0.093 | 0.127 | 0.44 | 0.612 | 0.905 |
| | | BiDO | 0.884 | 0.075 | 0.089 | 0.075 | 0.092 | 0.779 |
| | | TL-DMI | 0.906 | 0 | 0 | 0 | 0 | 0.478 |
| | | AT-AT | 0.921 | 0 | 0 | 0 | 0 | 0.074 |
| CelebA | RN-152 | NoDef | 0.838 | 0.050 | 0.066 | 0.050 | 0.067 | 0.699 |
| | | MID | 0.802 | 0.139 | 0.222 | 0.485 | 0.571 | 0.726 |
| | | BiDO | 0.747 | 0.062 | 0.070 | 0.062 | 0.072 | 0.600 |
| | | TL-DMI | 0.688 | 0 | 0 | 0 | 0 | 0.224 |
| | | TL-DMI-5 | 0.814 | 0 | 0 | 0 | 0 | 0.541 |
| | | AT-AT | 0.819 | 0 | 0 | 0 | 0 | 0.045 |
| | RN-18 | NoDef | 0.859 | 0.084 | 0.102 | 0.084 | 0.099 | 0.754 |
| | | MID | 0.737 | 0.168 | 0.266 | 0.423 | 0.551 | 0.650 |
| | | BiDO | 0.625 | 0.007 | 0.010 | 0.007 | 0.011 | 0.445 |
| | | TL-DMI | 0.752 | 0 | 0 | 0 | 0 | 0.317 |
| | | TL-DMI-5 | 0.831 | 0 | 0 | 0 | 0 | 0.517 |
| | | AT-AT[†a] | 0.825 | 0 | 0 | 0 | 0 | 0.368 |
| | | AT-AT[†b] | 0.787 | 0 | 0 | 0 | 0 | 0.191 |
| *Defenses that suppress attack gradients* | | | | | | | | |
| FaceScrub | RN-152 | RoLSS-SSF | 0.869 | 0.013 | 0.021 | 0.014 | 0.029 | 0.727 |
| | | RoLSS | 0.886 | 0.009 | 0.019 | 0.009 | 0.021 | 0.767 |
| | | NegLS | 0.906 | 0.048 | 0.784 | 0.048 | 0.065 | 0.774 |
| | | Trap-MID | 0.927 | 0 | 0 | 0 | 0.023 | 0.096 |
| | | Trap-MID* | 0.948 | 0 | 0 | 0 | 0.023 | 0.037 |
| | RN-18 | RoLSS-SSF | 0.950 | 0 | 0.002 | 0 | 0.013 | 0.882 |
| | | RoLSS | 0.950 | 0.002 | 0.004 | 0.002 | 0.009 | 0.880 |
| | | NegLS | 0.929 | 0.215 | 0.814 | 0.215 | 0.238 | 0.838 |
| | | Trap-MID | 0.928 | 0 | 0 | 0 | 0.010 | 0.295 |
| | | Trap-MID* | 0.927 | 0 | 0 | 0 | 0.005 | 0.018 |
| CelebA | RN-152 | RoLSS-SSF | 0.428 | 0 | 0 | 0 | 0 | 0.185 |
| | | RoLSS | 0.537 | 0 | 0 | 0 | 0.001 | 0.234 |
| | | NegLS | 0.805 | 0.013 | 0.184 | 0.013 | 0.019 | 0.624 |
| | | Trap-MID | 0.848 | 0 | 0 | 0 | 0.068 | 0.498 |
| | | Trap-MID* | 0.806 | 0 | 0 | 0 | 0.023 | 0.039 |
| | RN-18 | RoLSS-SSF | 0.839 | 0.08 | 0.094 | 0.08 | 0.098 | 0.729 |
| | | RoLSS | 0.830 | 0.069 | 0.078 | 0.070 | 0.083 | 0.709 |
| | | NegLS | 0.838 | 0.100 | 0.147 | 0.100 | 0.114 | 0.724 |
| | | Trap-MID | 0.830 | 0.044 | 0.144 | 0.044 | 0.079 | 0.690 |
| | | Trap-MID* | 0.811 | 0 | 0 | 0 | 0.015 | 0.069 |

*Notes:* [†a] and [†b] denote two sets of hyperparameters for AT-AT (see Table 6).

**Table 13:** AutoAttack robust accuracies with $\epsilon=0.0005$. Attack is on 8 batches (1024 validation set examples).

| Data | Arch. | Defense | TestAcc ↑ | Overall ↑ | APGD-CE ↑ | APGD-T ↑ | FAB-T ↑ | Square ↑ |
|---|---|---|---|---|---|---|---|---|
| *Defenses that attempt to increase privacy generically by reducing information dependency/memorization* | | | | | | | | |
| FaceScrub | RN-152 | NoDef | 0.958 | 0.899 | 0.900 | 0.899 | 0.900 | 0.955 |
| | | MID | 0.950 | 0.770 | 0.793 | 0.849 | 0.904 | 0.941 |
| | | BiDO | 0.928 | 0.847 | 0.848 | 0.847 | 0.849 | 0.91 |
| | | TL-DMI | 0.911 | 0 | 0 | 0 | 0 | 0.801 |
| | | AT-AT | 0.939 | 0 | 0 | 0 | 0 | 0.516 |
| | RN-18 | NoDef | 0.950 | 0.896 | 0.896 | 0.896 | 0.896 | 0.946 |
| | | MID | 0.924 | 0.825 | 0.871 | 0.900 | 0.898 | 0.904 |
| | | BiDO | 0.884 | 0.784 | 0.785 | 0.784 | 0.786 | 0.859 |
| | | TL-DMI | 0.906 | 0.003 | 0.004 | 0.003 | 0.005 | 0.843 |
| | | AT-AT | 0.921 | 0 | 0 | 0 | 0 | 0.696 |
| CelebA | RN-152 | NoDef | 0.838 | 0.675 | 0.677 | 0.675 | 0.675 | 0.800 |
| | | MID | 0.802 | 0.542 | 0.704 | 0.726 | 0.717 | 0.728 |
| | | BiDO | 0.747 | 0.590 | 0.590 | 0.590 | 0.590 | 0.717 |
| | | TL-DMI | 0.688 | 0.002 | 0.002 | 0.002 | 0.004 | 0.559 |
| | | TL-DMI-5 | 0.814 | 0.119 | 0.130 | 0.119 | 0.132 | 0.765 |
| | | AT-AT | 0.819 | 0 | 0 | 0 | 0 | 0.104 |
| | RN-18 | NoDef | 0.859 | 0.746 | 0.748 | 0.746 | 0.747 | 0.835 |
| | | MID | 0.737 | 0.452 | 0.599 | 0.636 | 0.639 | 0.646 |
| | | BiDO | 0.625 | 0.394 | 0.398 | 0.394 | 0.396 | 0.581 |
| | | TL-DMI | 0.752 | 0.028 | 0.037 | 0.028 | 0.041 | 0.656 |
| | | TL-DMI-5 | 0.831 | 0.146 | 0.154 | 0.146 | 0.165 | 0.763 |
| | | AT-AT[†a] | 0.825 | 0 | 0 | 0 | 0 | 0.721 |
| | | AT-AT[†b] | 0.787 | 0 | 0 | 0 | 0 | 0.617 |
| *Defenses that suppress attack gradients* | | | | | | | | |
| FaceScrub | RN-152 | RoLSS-SSF | 0.869 | 0.721 | 0.721 | 0.721 | 0.723 | 0.843 |
| | | RoLSS | 0.886 | 0.744 | 0.747 | 0.744 | 0.750 | 0.868 |
| | | NegLS | 0.906 | 0.769 | 0.828 | 0.769 | 0.769 | 0.882 |
| | | Trap-MID | 0.927 | 0 | 0 | 0 | 0.023 | 0.790 |
| | | Trap-MID* | 0.948 | 0 | 0 | 0 | 0.023 | 0.890 |
| | RN-18 | RoLSS-SSF | 0.950 | 0.813 | 0.816 | 0.813 | 0.820 | 0.941 |
| | | RoLSS | 0.950 | 0.819 | 0.822 | 0.819 | 0.826 | 0.938 |
| | | NegLS | 0.929 | 0.854 | 0.869 | 0.854 | 0.854 | 0.916 |
| | | Trap-MID | 0.928 | 0.530 | 0.565 | 0.548 | 0.585 | 0.852 |
| | | Trap-MID* | 0.927 | 0 | 0 | 0 | 0.004 | 0.833 |
| CelebA | RN-152 | RoLSS-SSF | 0.428 | 0.042 | 0.048 | 0.042 | 0.054 | 0.394 |
| | | RoLSS | 0.537 | 0.086 | 0.094 | 0.087 | 0.102 | 0.494 |
| | | NegLS | 0.805 | 0.559 | 0.578 | 0.559 | 0.562 | 0.749 |
| | | Trap-MID | 0.848 | 0.333 | 0.412 | 0.338 | 0.409 | 0.787 |
| | | Trap-MID* | 0.806 | 0 | 0 | 0 | 0.023 | 0.601 |
| | RN-18 | RoLSS-SSF | 0.839 | 0.723 | 0.728 | 0.723 | 0.725 | 0.819 |
| | | RoLSS | 0.830 | 0.701 | 0.704 | 0.701 | 0.702 | 0.796 |
| | | NegLS | 0.838 | 0.711 | 0.711 | 0.711 | 0.712 | 0.803 |
| | | Trap-MID | 0.830 | 0.685 | 0.732 | 0.685 | 0.690 | 0.812 |
| | | Trap-MID* | 0.811 | 0.001 | 0.001 | 0.001 | 0.016 | 0.724 |

*Notes:* [†a] and [†b] denote two sets of hyperparameters for AT-AT (see Table 6).

**Table 14:** AutoAttack robust accuracies with $\epsilon=10^{-5}$. Attack is on 8 batches (1024 validation set examples).

| Data | Arch. | Defense | TestAcc ↑ | Overall ↑ | APGD-CE ↑ | APGD-T ↑ | FAB-T ↑ | Square ↑ |
|---|---|---|---|---|---|---|---|---|
| *Defenses that attempt to increase privacy generically by reducing information dependency/memorization* | | | | | | | | |
| FaceScrub | RN-152 | NoDef | 0.958 | 0.960 | 0.960 | 0.960 | 0.960 | 0.960 |
| | | MID | 0.950 | 0.928 | 0.944 | 0.940 | 0.945 | 0.941 |
| | | BiDO | 0.928 | 0.923 | 0.923 | 0.923 | 0.923 | 0.923 |
| | | TL-DMI | 0.911 | 0.867 | 0.867 | 0.867 | 0.867 | 0.902 |
| | | AT-AT | 0.939 | 0.119 | 0.127 | 0.120 | 0.157 | 0.897 |
| | RN-18 | NoDef | 0.950 | 0.959 | 0.959 | 0.959 | 0.959 | 0.959 |
| | | MID | 0.924 | 0.856 | 0.900 | 0.905 | 0.898 | 0.914 |
| | | BiDO | 0.884 | 0.879 | 0.880 | 0.879 | 0.879 | 0.881 |
| | | TL-DMI | 0.906 | 0.886 | 0.887 | 0.886 | 0.886 | 0.899 |
| | | AT-AT | 0.921 | 0.692 | 0.702 | 0.692 | 0.711 | 0.911 |
| CelebA | RN-152 | NoDef | 0.838 | 0.818 | 0.818 | 0.818 | 0.819 | 0.820 |
| | | MID | 0.802 | 0.574 | 0.729 | 0.723 | 0.720 | 0.733 |
| | | BiDO | 0.747 | 0.748 | 0.749 | 0.748 | 0.748 | 0.751 |
| | | TL-DMI | 0.688 | 0.641 | 0.644 | 0.641 | 0.643 | 0.668 |
| | | TL-DMI-5 | 0.814 | 0.799 | 0.801 | 0.799 | 0.801 | 0.811 |
| | | AT-AT | 0.819 | 0.155 | 0.178 | 0.157 | 0.180 | 0.722 |
| | RN-18 | NoDef | 0.859 | 0.848 | 0.848 | 0.848 | 0.848 | 0.849 |
| | | MID | 0.737 | 0.464 | 0.644 | 0.648 | 0.651 | 0.646 |
| | | BiDO | 0.625 | 0.609 | 0.609 | 0.609 | 0.609 | 0.611 |
| | | TL-DMI | 0.752 | 0.718 | 0.718 | 0.718 | 0.719 | 0.734 |
| | | TL-DMI-5 | 0.831 | 0.802 | 0.802 | 0.802 | 0.803 | 0.816 |
| | | AT-AT[†a] | 0.825 | 0.756 | 0.758 | 0.756 | 0.757 | 0.802 |
| | | AT-AT[†b] | 0.787 | 0.645 | 0.661 | 0.645 | 0.652 | 0.786 |
| *Defenses that suppress attack gradients* | | | | | | | | |
| FaceScrub | RN-152 | RoLSS-SSF | 0.869 | 0.857 | 0.857 | 0.857 | 0.857 | 0.860 |
| | | RoLSS | 0.886 | 0.882 | 0.882 | 0.882 | 0.882 | 0.883 |
| | | NegLS | 0.906 | 0.905 | 0.905 | 0.905 | 0.905 | 0.911 |
| | | Trap-MID | 0.927 | 0.856 | 0.859 | 0.856 | 0.861 | 0.911 |
| | | Trap-MID | 0.948 | 0.943 | 0.943 | 0.943 | 0.943 | 0.948 |
| | RN-18 | RoLSS-SSF | 0.950 | 0.951 | 0.952 | 0.951 | 0.952 | 0.954 |
| | | RoLSS | 0.950 | 0.948 | 0.948 | 0.948 | 0.948 | 0.951 |
| | | NegLS | 0.929 | 0.933 | 0.933 | 0.933 | 0.933 | 0.934 |
| | | Trap-MID | 0.928 | 0.914 | 0.914 | 0.914 | 0.914 | 0.916 |
| | | Trap-MID* | 0.927 | 0.921 | 0.921 | 0.921 | 0.921 | 0.934 |
| CelebA | RN-152 | RoLSS-SSF | 0.428 | 0.406 | 0.408 | 0.406 | 0.408 | 0.414 |
| | | RoLSS | 0.537 | 0.530 | 0.530 | 0.530 | 0.530 | 0.538 |
| | | NegLS | 0.805 | 0.771 | 0.771 | 0.771 | 0.771 | 0.776 |
| | | Trap-MID | 0.848 | 0.828 | 0.830 | 0.828 | 0.828 | 0.835 |
| | | Trap-MID* | 0.806 | 0.760 | 0.760 | 0.760 | 0.760 | 0.800 |
| | RN-18 | RoLSS-SSF | 0.839 | 0.834 | 0.834 | 0.834 | 0.834 | 0.834 |
| | | RoLSS | 0.830 | 0.813 | 0.813 | 0.813 | 0.814 | 0.815 |
| | | NegLS | 0.838 | 0.825 | 0.825 | 0.825 | 0.825 | 0.825 |
| | | Trap-MID | 0.830 | 0.827 | 0.828 | 0.827 | 0.828 | 0.829 |
| | | Trap-MID* | 0.811 | 0.790 | 0.790 | 0.790 | 0.791 | 0.809 |

*Notes:* [†a] and [†b] denote two sets of hyperparameters for AT-AT (see Table 6).

## G  DEFERRED PRIVACY-ROBUSTNESS REGRESSION ANALYSIS DETAILS AND RESULTS

Section 4 investigates whether the privacy associated with mainstream generic information dependency-based defenses (MID, BiDO, TL-DMI) is actually correlated with, or explained by, a tradeoff with adversarial robustness. Concretely, we ask:

1. Can we perfectly predict MIA leakage (AttAcc@1) just from a defense's robust accuracy under AutoAttack?

2. For each percentage point reduction in leakage (AttAcc@1 under PPA or IF-GMI), what is the associated cost in terms of robust accuracy under AutoAttack at different $\varepsilon$, which capture non-robust features of different magnitudes (See Fig. 5).

We accomplish this by presenting the first systematic tests of the robustness of MIA defenses to a very different attack vector: adversarial examples. Tables 11, 12, 13, 14 in App. F.4 report robust accuracies under each of AutoAttack's constituent attacks. Specifically, we are interested in the standard AutoAttack robust accuracy, which is the worst-case robust accuracy across the various attacks in the AutoAttack ensemble, at various magnitudes $\varepsilon$. This is reported in Table 4.

To obtain a principled test of whether these raw AutoAttack and privacy leakage results reflect a relationship between leakage and robustness under AutoAttack, we consider the following simple *main regression specifications*. These standard regression models test the degree to which privacy leakage (our response variable, measured by AttAcc@1 $\in [0, 1]$ under PPA or under IF-GMI) is a linear function of a defense's *robust accuracy* under AutoAttack at several perturbation magnitudes, above and beyond its *clean* test accuracy (control variable). Intuitively, these models ask: *holding clean accuracy fixed, do defenses that are less robust to small adversarial perturbations (i.e., that rely more on non-robust features) exhibit less leakage under MIA?*

Specifically, the following OLS model estimates the magnitude and significance of this privacy-robustness relationship:

$$\text{Leakage}_{\text{AttAcc@1},j} = \beta_0 + \beta_1 \text{TestAcc}_j + \beta_2 \text{Acc}_{\epsilon=0.0025,j} + \beta_3 \text{Acc}_{\epsilon=0.0005,j} + \beta_4 \text{Acc}_{\epsilon=10^{-5},j} + e_j \quad (4)$$

We are careful to also estimate Beta regression variants with appropriate distributional assumptions and logit link ($g(x)$) to capture the fact that the response (AttAcc@1 under PPA or IF-GMI) is bounded in $[0, 1]$:

$$\text{Leakage}_{\text{AttAcc@1},j} \sim \text{Beta}(\mu_j \phi, (1 - \mu_j)\phi)$$
$$g(\mu_j) = \beta_0 + \beta_1 \text{TestAcc}_j + \beta_2 \text{Acc}_{\epsilon=0.0025,j} + \beta_3 \text{Acc}_{\epsilon=0.0005,j} + \beta_4 \text{Acc}_{\epsilon=10^{-5},j} \quad (5)$$

Because we estimate these models using a relatively small number of observations, we are also careful to report, for OLS, the appropriate small sample-robust HC2 standard errors. We are also careful to compute confidence intervals with the conservative small sample $t_{0.975,13}=2.160$ (with the HC2 SEs) and report *adjusted* $R^2$. Beta regressions already accommodate appropriate assumptions, so we report CI with the standard Wald ($\pm 1.96$SE) on the logit scale.

To evaluate goodness-of-fit, we report *degree-of-freedom adjusted* $R^2$ (for OLS) or the appropriately estimated pseudo-$R^2$ (for Beta regressions). For all specifications, we also report Mean Absolute Error (MAE) on the response scale.

Figure 2 visualizes the fitted values.

**Interpretation (units and signs).**  All accuracies and leakages are in $[0, 1]$ during estimation; for readability we discuss effects in *percentage points* (pp). A positive regression $\beta_2, \beta_3$, and/or $\beta_4$ means that, *holding clean accuracy fixed*, higher robust accuracy at that respective $\epsilon$ is associated with *higher* leakage (worse privacy). Equivalently, privacy gains from generic defenses coincide with losses in robustness to small perturbations (i.e., increased reliance on non-robust features). The coefficient $\beta_1$ captures any residual association between clean accuracy and leakage after conditioning on robustness.

*Defenses, scope, and limitations.* In keeping with our hypothesis that *generic* defenses work, at least in part, by shifting their models' feature bases towards non-robust features, we estimate regressions

this set of defenses (MID, BiDO, TL-DMI, and TL-DMI-5 where applicable). Note that TL-DMI-5 is a variant of TL-DMI with 5 layers frozen instead of the standard 6 layers, which we include alongside the standard TL-DMI to give this defense the best opportunity to perform well on lower-capacity ResNet-18 networks that have fewer layers.

We then reestimate all regressions on the dataset of gradient-suppressing defenses (NegLS, RoLSS, RoLSS-SSF), which directly target PPA/IF-GMI gradients, as we do not hypothesize that these defenses work via non-robust features.

We emphasize that these regressions provide correlational evidence, and *do not by themselves prove causality*. Our separate randomized AT-AT experiment (Section 5) provides causal evidence by treating the model's reliance on non-robust features in a randomized controlled experiment and observing the induced change in leakage.

### G.1 DEFERRED OLS AND BETA REGRESSION RESULTS.

We find that leakage (AttAcc@1 under both PPA and IF-GMI) is strongly and significantly correlated with robust accuracy at all levels of $\epsilon$ (note $p$ values), with $R^2 \approx 0.93 - 0.95$, and mean abs. error of just $0.042 - 0.05$. Contrary to widespread belief, (2) clean test accuracy is *only weakly or insignificantly correlated* with leakage after controlling for robust accuracy. To examine the robustness of these results to different regression specifications, PPA versus IF-GMI attack leakage, different distributional assumptions, controls for dataset/architecture, various robust standard errors estimators, etc., we show that these results hold across 56 alternate regression specifications.

Specifically, Table 15 presents full deferred OLS and Beta regressions for NoDef and generic defenses (MID, BiDO, TL-DMI), which we hypothesize unintentionally obtain their privacy from non-robust features.

Table 18 recomputes these regressions on generic privacy defenses using IF-GMI AttAcc@1 to measure privacy leakage (response variable) instead of PPA. A similar strong relationship exists between robust accuracy and leakage using the IF-GMI AttAcc@1 as a leakage measure.

Table 16 recomputes these regressions using PPA AttAcc@1, but on gradient-suppressing defenses (NegLS, RoLSS, RoLSS-SSF), which we do not hypothesize to work via non-robust features. As expected, the regression results do not show the same strong, significant relationship between leakage and non-robust features. Also note that the model fit metrics (e.g. $R^2$) are far lower for these gradient-suppressing defenses.

Table 19 shows that the lack of clear, significant relationship between leakage and non-robust features *for gradient-suppressing defenses* (NegLS, RoLSS, RoLSS-SSF) persists when we measure leakage via IF-GMI AttAcc@1 instead of PPA AttAcc@1.

Finally, Tables 17 and 20 present multiple additional 'robustness test' regressions, showing that the strong and highly statistically significant relationship between leakage and non-robust features is not merely an artifact of the regression specification. Specifically, Tables 17 and 20, *left side* show that controlling for dataset interactions still yields a significantly worse fitting regression model compared to the non-robust features regressions described above, regardless of whether leakage is measured via PPA attacks (Table 17) or IF-GMI attacks (Table 20). Tables 17 and 20, *right side* show that the same applies when we instead control for architecture interactions. Note that the goodness-of-fit metrics in both cases are inferior to the goodness-of-fit metrics for non-robust regressions.

**Using coefficients to estimate the *cost of privacy*.** Per the Beta model, we can compute that reducing PPA leakage (AttAcc@1) by 1 percentage point (pp) corresponds to a significant *0.31 pp decrease* in AutoAttack robust accuracy at $\epsilon$=0.0025, (95% CI: 0.24–0.45), a significant *5.4 pp decrease* at $\epsilon$=0.0005 (95% CI: 2.9–58.3), and a significant *0.91 pp decrease* at $\epsilon$=$10^{-5}$ (95% CI: 0.60–1.93). To derive these 'costs' of privacy, note that in Beta regression with logit link (Eq. 2), $\text{logit}(\mu_j) = X_j\beta$, coefficients $\beta_\epsilon$ live on the link scale. The response-scale marginal effect of robust accuracy at $\epsilon$ on expected leakage is $\frac{\partial \mathbb{E}[\text{Leakage}]}{\partial \text{Acc}_\epsilon} = \beta_\epsilon \, \mu(1-\mu)$. Evaluated at the sample mean

leakage $\bar{\mu}$ from the regression sample, the *robustness cost of a 1 pp privacy gain* is the inverse slope

$$\gamma_\epsilon \;\equiv\; \left.\frac{\partial\,\mathrm{Acc}_\epsilon}{\partial\,\mathrm{Leakage}}\right|_{\mu=\bar{\mu}} \;=\; \frac{1}{\beta_\epsilon\,\bar{\mu}(1-\bar{\mu})} \quad \text{(pp of robust accuracy per 1 pp change in leakage).}$$

*Confidence intervals.* Using the delta method, if we treat $\bar{\mu}$ as fixed, $\mathrm{SE}(\hat{\gamma}_\epsilon) \approx \left|\frac{\partial\gamma}{\partial\beta_\epsilon}\right| \mathrm{SE}(\hat{\beta}_\epsilon) = \frac{1}{|\hat{\beta}_\epsilon|^2\,\bar{\mu}(1-\bar{\mu})}\,\mathrm{SE}(\hat{\beta}_\epsilon)$, so a 95% CI is $\hat{\gamma}_\epsilon \pm 1.96\,\mathrm{SE}(\hat{\gamma}_\epsilon)$. More generally, allowing for uncertainty in the parameter $\mu$:

$$\mathrm{Var}(\hat{\gamma}_\epsilon) \;\approx\; \underbrace{\left(\frac{\partial\gamma}{\partial\beta_\epsilon}\right)^2 \mathrm{Var}(\hat{\beta}_\epsilon)}_{\text{link coef.}} \;+\; \underbrace{\left(\frac{\partial\gamma}{\partial\mu}\right)^2 \mathrm{Var}(\hat{\mu})}_{\text{mean leakage}} \;+\; 2\frac{\partial\gamma}{\partial\beta_\epsilon}\frac{\partial\gamma}{\partial\mu}\,\mathrm{Cov}(\hat{\beta}_\epsilon,\hat{\mu}),$$

with $\frac{\partial\gamma}{\partial\beta_\epsilon} = -\frac{1}{\beta_\epsilon^2\,\mu(1-\mu)}$ and $\frac{\partial\gamma}{\partial\mu} = -\frac{1-2\mu}{\beta_\epsilon\,\mu^2(1-\mu)^2}$. In practice we report CIs using the fixed-$\bar{\mu}$ version (slightly narrower); including $\mathrm{Var}(\hat{\mu})$ yields virtually identical conclusions.

**Table 15:** Linear (OLS) & Beta regressions quantifying the privacy–robustness tradeoff for **generic information dependency-based privacy defenses** under various specifications. Leakage (response) is **PPA AttAcc@1**. SE's in parentheses; OLS MacKinnon & White (HC2) small-sample robust SE's in brackets. Test accuracy is clean; $\epsilon$'s are robust accuracies under the AutoAttack ensemble at the indicated noise levels. Note that Beta coefficients are on the logit scale.

| | All Predictors | | All $\epsilon$ | | Clean Acc. | | $\epsilon$=0.0025 | | $\epsilon$=0.0005 | | $\epsilon$=$10^{-5}$ | |
|---|---|---|---|---|---|---|---|---|---|---|---|---|
| | OLS | Beta | OLS | Beta | OLS | Beta | OLS | Beta | OLS | Beta | OLS | Beta |
| *Predictors* | | | | | | | | | | | | |
| Test Acc. | -0.011 | -0.192 | | | 1.759** | 7.495*** | | | | | | |
| | (0.390) | (1.582) | | | (0.477) | (2.041) | | | | | | |
| | [0.478] | | | | [0.458] | | | | | | | |
| $\epsilon$=0.0025 | 2.299*** | 12.827*** | 2.291*** | 12.685*** | | | 3.049*** | 13.689*** | | | | |
| | (0.452) | (2.010) | (0.363) | (1.672) | | | (0.604) | (2.694) | | | | |
| | [0.739] | | [0.428] | | | | [0.768] | | | | | |
| $\epsilon$=0.0005 | 0.211* | 0.737* | 0.212* | 0.748* | | | | | 0.639*** | 2.915*** | | |
| | (0.080) | (0.325) | (0.074) | (0.313) | | | | | (0.090) | (0.442) | | |
| | [0.108] | | [0.083] | | | | | | [0.084] | | | |
| $\epsilon$=$10^{-5}$ | 0.795* | 4.390*** | 0.788*** | 4.263*** | | | | | | | 0.969** | 4.561** |
| | (0.285) | (1.180) | (0.130) | (0.597) | | | | | | | (0.378) | (1.524) |
| | [0.360] | | [0.113] | | | | | | | | [0.474] | |
| *Fit statistics* | | | | | | | | | | | | |
| $R^2$ | 0.934 | 0.952 | 0.939 | 0.952 | 0.425 | 0.485 | 0.590 | 0.613 | 0.743 | 0.733 | 0.247 | 0.304 |
| Mean abs. err. | 0.047 | 0.042 | 0.047 | 0.042 | 0.156 | 0.151 | 0.126 | 0.124 | 0.099 | 0.095 | 0.186 | 0.181 |

*Note: $R^2$ are Adjusted (OLS); Pseudo (Beta).* $\qquad$ *p<0.1; **p<0.05; ***p<0.01

**Table 16:** Linear (OLS) & Beta regressions quantifying privacy–robustness tradeoff for **attack gradient suppressor defenses (NegLS, RoLSS, RoLSS-SSF)** under various specifications. Leakage (response) is **PPA AttAcc@1**. SE's in parentheses; OLS MacKinnon & White (HC2) small-sample robust SE's in brackets. Test accuracy is clean; $\epsilon$'s are robust accuracies under AutoAttack at the indicated noise levels. Note that Beta coefficients are on the logit scale.

| | All Predictors | | All $\epsilon$ | | Clean Acc. | | $\epsilon$=0.0025 | | $\epsilon$=0.0005 | | $\epsilon$=$10^{-5}$ | |
|---|---|---|---|---|---|---|---|---|---|---|---|---|
| | OLS | Beta | OLS | Beta | OLS | Beta | OLS | Beta | OLS | Beta | OLS | Beta |
| *Predictors* | | | | | | | | | | | | |
| Test Acc. | -2.016 | -0.053 | | | 1.296** | 6.434*** | | | | | | |
| | (8.500) | (27.413) | | | (0.353) | (1.721) | | | | | | |
| | [3.780] | | | | [0.282] | | | | | | | |
| $\epsilon$=0.0025 | -0.167 | -0.912 | -0.026 | -0.908 | | | 1.264 | 4.883 | | | | |
| | (1.419) | (4.615) | (1.211) | (4.189) | | | (1.327) | (4.833) | | | | |
| | [0.785] | | [0.646] | | | | [0.906] | | | | | |
| $\epsilon$=0.0005 | 0.757 | 2.868 | 0.547 | 2.862 | | | | | 0.780** | 3.805*** | | |
| | (1.808) | (6.437) | (1.481) | (5.743) | | | | | (0.208) | (1.025) | | |
| | [0.537] | | [0.691] | | | | | | [0.125] | | | |
| $\epsilon$=$10^{-5}$ | 2.018 | 1.759 | 0.387 | 1.717 | | | | | | | 1.258** | 6.171*** |
| | (7.307) | (23.662) | (2.326) | (8.951) | | | | | | | (0.340) | (1.649) |
| | [4.067] | | [1.337] | | | | | | | | [0.282] | |
| *Fit statistics* | | | | | | | | | | | | |
| $R^2$ | 0.355 | 0.727 | 0.431 | 0.727 | 0.532 | 0.718 | -0.008 | 0.080 | 0.543 | 0.722 | 0.535 | 0.713 |
| Mean abs. err. | 0.120 | 0.124 | 0.123 | 0.124 | 0.137 | 0.131 | 0.194 | 0.195 | 0.122 | 0.118 | 0.135 | 0.129 |

*Note: $R^2$ are Adj. (OLS); Pseudo (Beta).* $\qquad$ *p<0.05; **p<0.01; ***p<0.001

**Table 17:** Robustness checks: Leakage regressions with Clean Accuracy $\times$ Dataset or Clean Accuracy $\times$ Architecture interaction as an alternate explanation for the leakage of generic information dependency-based defenses. Leakage (response) is **PPA AttAcc@1**. Left: Dataset interaction; Right: architecture interaction (ResNet-152 vs. ResNet-18). SE's in parentheses; OLS MacKinnon & White (HC2) small-sample robust SE's in brackets.

| | OLS | Beta | | OLS | Beta |
|---|---|---|---|---|---|
| (Intercept) | $-1.055$ | $-6.997^{***}$ | (Intercept) | $-1.021$ | $-6.447^{**}$ |
| | (0.664) | (2.714) | | (0.650) | (2.527) |
| | [0.412] | | | [0.617] | |
| Test Acc. | $1.805^{*}$ | $8.188^{**}$ | Test Acc. | $1.719^{**}$ | $7.320^{**}$ |
| | (0.860) | (3.469) | | (0.762) | (2.944) |
| | [0.538] | | | [0.785] | |
| DatasetFaceScrub | $-3.983$ | $-18.816^{*}$ | ArchRN-18 | $-0.051$ | $-0.203$ |
| | (2.688) | (10.302) | | (0.866) | (3.395) |
| | [3.746] | | | [0.775] | |
| Test Acc.:DatasetFaceScrub | $4.257$ | $20.047^{*}$ | Test Acc.:ArchRN-18 | $0.112$ | $0.443$ |
| | (2.940) | (11.282) | | (1.024) | (3.989) |
| | [4.042] | | | [0.988] | |
| $R^2$ | 0.442 | 0.574 | $R^2$ | 0.353 | 0.493 |
| Mean abs. err. | 0.150 | 0.146 | Mean abs. err. | 0.155 | 0.152 |

*Note: $R^2$ are Adj. (OLS); Pseudo (Beta).*   $^{*}$**p<0.1**; $^{**}$**p<0.05**; $^{***}$**p<0.01**

**Table 18:** Linear (OLS) & Beta regressions quantifying the privacy–robustness tradeoff for **generic information dependency-based privacy defenses** under various specifications. Leakage (response) is **IF-GMI AttAcc@1**. SE's in parentheses; OLS MacKinnon & White (HC2) small-sample robust SE's in brackets. Test accuracy is clean; $\epsilon$'s are robust accuracies under the AutoAttack ensemble at the indicated noise levels. Note that Beta coefficients are on the logit scale.

| | All Predictors | | All $\epsilon$ | | Clean Acc. | | $\epsilon=0.0025$ | | $\epsilon=0.0005$ | | $\epsilon=10^{-5}$ | |
|---|---|---|---|---|---|---|---|---|---|---|---|---|
| | OLS | Beta | OLS | Beta | OLS | Beta | OLS | Beta | OLS | Beta | OLS | Beta |
| *Predictors* | | | | | | | | | | | | |
| Test Acc. | -0.496 | $-7.272^{**}$ | | | $1.343^{**}$ | $7.146^{***}$ | | | | | | |
| | (0.827) | (3.149) | | | (0.471) | (2.102) | | | | | | |
| | [1.116] | | | | [0.466] | | | | | | | |
| $\epsilon=0.0025$ | $2.471^{**}$ | $20.719^{***}$ | $2.153^{**}$ | $13.175^{***}$ | | | $2.330^{***}$ | $13.590^{***}$ | | | | |
| | (0.960) | (4.438) | (0.781) | (3.688) | | | (0.642) | (3.129) | | | | |
| | [1.766] | | [1.049] | | | | [0.748] | | | | | |
| $\epsilon=0.0005$ | 0.030 | -0.686 | 0.056 | -0.043 | | | | | $0.458^{***}$ | $2.249^{***}$ | | |
| | (0.169) | (0.622) | (0.160) | (0.663) | | | | | (0.117) | (0.555) | | |
| | [0.257] | | [0.204] | | | | | | [0.108] | | | |
| $\epsilon=10^{-5}$ | $1.072^{*}$ | $10.162^{***}$ | $0.752^{**}$ | $4.688^{***}$ | | | | | | | $0.787^{**}$ | $4.146^{***}$ |
| | (0.605) | (2.598) | (0.279) | (1.268) | | | | | | | (0.349) | (1.507) |
| | [0.840] | | [0.221] | | | | | | | | [0.389] | |
| *Fit statistics* | | | | | | | | | | | | |
| $R^2$ | 0.628 | 0.837 | 0.645 | 0.801 | 0.295 | 0.361 | 0.418 | 0.486 | 0.460 | 0.488 | 0.194 | 0.301 |
| Mean abs. err. | 0.096 | 0.082 | 0.094 | 0.094 | 0.143 | 0.138 | 0.138 | 0.129 | 0.123 | 0.124 | 0.167 | 0.166 |

*Note: $R^2$ are Adjusted (OLS); Pseudo (Beta).*   $^{*}$**p<0.1**; $^{**}$**p<0.05**; $^{***}$**p<0.01**

**Table 19:** Linear (OLS) & Beta regressions quantifying privacy–robustness tradeoff for **attack gradient suppressing privacy defenses (NegLS, RoLSS, RoLSS-SSF)** under various specifications. Leakage (response) is **IF-GMI AttAcc@1**. SE's in parentheses; OLS MacKinnon & White (HC2) small-sample robust SE's in brackets. Test accuracy is clean; $\epsilon$'s are robust accuracies under AutoAttack at the indicated noise levels. Note that Beta coefficients are on the logit scale.

| | All Predictors | | All $\epsilon$ | | Clean Acc. | | $\epsilon$=0.0025 | | $\epsilon$=0.0005 | | $\epsilon$=$10^{-5}$ | |
|---|---|---|---|---|---|---|---|---|---|---|---|---|
| | *OLS* | *Beta* | *OLS* | *Beta* | *OLS* | *Beta* | *OLS* | *Beta* | *OLS* | *Beta* | *OLS* | *Beta* |
| ***Predictors*** | | | | | | | | | | | | |
| Test Acc. | 3.326 | 6.828 | | | 1.577** | 6.567** | | | | | | |
| | (11.153) | (37.227) | | | (0.490) | (2.107) | | | | | | |
| | [4.541] | | | | [0.314] | | | | | | | |
| $\epsilon$=0.0025 | 1.102 | 2.484 | 0.870 | 2.015 | | | 2.513 | 6.428 | | | | |
| | (1.861) | (6.437) | (1.591) | (5.876) | | | (1.606) | (5.551) | | | | |
| | [0.945] | | [0.742] | | | | [1.186] | | | | | |
| $\epsilon$=0.0005 | 0.826 | 2.483 | 1.172 | 3.233 | | | | | 0.975** | 4.027** | | |
| | (2.373) | (8.159) | (1.948) | (7.188) | | | | | (0.281) | (1.254) | | |
| | [0.617] | | [0.850] | | | | | | [0.146] | | | |
| $\epsilon$=$10^{-5}$ | -3.149 | -4.382 | -0.460 | 1.049 | | | | | | | 1.518** | 6.327** |
| | (9.587) | (32.113) | (3.059) | (11.264) | | | | | | | (0.477) | (2.040) |
| | [5.097] | | [1.579] | | | | | | | | [0.314] | |
| ***Fit statistics*** | | | | | | | | | | | | |
| $R^2$ | 0.336 | 0.595 | 0.412 | 0.593 | 0.459 | 0.572 | 0.116 | 0.137 | 0.501 | 0.589 | 0.453 | 0.569 |
| Mean abs. err. | 0.157 | 0.180 | 0.160 | 0.183 | 0.196 | 0.201 | 0.249 | 0.264 | 0.177 | 0.188 | 0.197 | 0.201 |

*Note: $R^2$ are Adj. (OLS); Pseudo (Beta).*    *$^{*}$p<0.05; $^{**}$p<0.01; $^{***}$p<0.001*

**Table 20:** Robustness checks: Leakage regressions with Clean Accuracy × Dataset or Clean Accuracy × Architecture interaction as an alternate explanation for the leakage of generic information dependency-based defenses. Leakage (response) is **IF-GMI AttAcc@1**. Left: Dataset interaction; Right: architecture interaction (ResNet-152 vs. ResNet-18). SE's in parentheses; OLS MacKinnon & White (HC2) small-sample robust SE's in brackets.

| | *OLS* | *Beta* |
|---|---|---|
| (Intercept) | $-1.333^{*}$ | $-7.277^{**}$ |
| | (0.635) | (2.722) |
| | [0.509] | |
| Test Acc. | $2.520^{**}$ | $9.987^{***}$ |
| | (0.822) | (3.541) |
| | [0.634] | |
| DatasetFaceScrub | $-0.631$ | $-13.955$ |
| | (2.570) | (11.315) |
| | [3.777] | |
| Test Acc.:DatasetFaceScrub | 0.382 | 14.268 |
| | (2.810) | (12.406) |
| | [4.109] | |
| $R^2$ | 0.471 | 0.474 |
| Mean abs. err. | 0.127 | 0.126 |

| | *OLS* | *Beta* |
|---|---|---|
| (Intercept) | $-0.322$ | $-5.336^{**}$ |
| | (0.634) | (2.604) |
| | [0.842] | |
| Test Acc. | 1.136 | $7.187^{**}$ |
| | (0.743) | (3.093) |
| | [1.028] | |
| ArchRN-18 | $-0.300$ | 0.158 |
| | (0.844) | (3.436) |
| | [0.883] | |
| Test Acc.:ArchRN-18 | 0.428 | 0.026 |
| | (0.998) | (4.107) |
| | [1.088] | |
| $R^2$ | 0.363 | 0.369 |
| Mean abs. err. | 0.144 | 0.138 |

*Note: $R^2$ are Adj. (OLS); Pseudo (Beta).*    *$^{*}$p<0.1; $^{**}$p<0.05; $^{***}$p<0.01*

# H    DEFERRED DETAILS OF RANDOMIZED CONTROLLED EXPERIMENT: AT-AT CAUSES PRIVACY

**Causal effect of AT-AT non-robust feature reward term on privacy.** By design, note that **AT-AT** with $\lambda = 0$ simplifies to vanilla training i.e. NoDef. Therefore, to estimate the causal effect of including **AT-AT**'s non-robust feature reward term (i.e. $\lambda > 0$) on privacy, we train 10 RN-152's on FaceScrub and randomly assign half as 'control' $\lambda$=0 (equal to Vanilla/NoDef) and half as 'treatment' $\lambda$>0. For treatment units, we use the default $\lambda = 0.04$ used throughout our experiments (See App 6).

**Causal OLS (ATE on the probability scale).** Let $Y_j \equiv \text{Leakage}_j$ (AttAcc@1 for model $j$) and $D_j \equiv \mathbb{1}\{\lambda_j > 0\}$ indicate assignment to nonzero $\lambda$, i.e., treatment with the **AT-AT** non-robust reward in the loss function. With complete randomization, $D_j \perp (Y_j(0), Y_j(1))$ and the ATE is $\tau = \mathbb{E}[Y_j(1) - Y_j(0)]$. We estimate

$$Y_j = \alpha + \tau D_j + u_j, \qquad \mathbb{E}[u_j \mid D_j] = 0, \tag{6}$$

and report HC2-robust SEs with small-sample $t$ critical values.

**Causal Beta regression (randomized AT-AT vs. control).** As before, we note that the response (AttAcc@1) is bounded in [0,1], thus we also estimate the causal effect via the appropriate beta regression. Let $Y_j \equiv \text{Leakage}_j$ (AttAcc@1 for model $j$) and $D_j \equiv \mathbb{1}\{\lambda_j > 0\}$ indicate assignment to AT-AT (treatment). Under complete randomization, $D_j \perp (Y_j(0), Y_j(1))$.

$$Y_j \sim \text{Beta}(\mu_j \phi, \, (1 - \mu_j)\phi),$$
$$\text{logit}(\mu_j) = \alpha \, + \, \tau \, D_j. \tag{7}$$

*Odds-ratio interpretation.* The treatment effect on the log-odds scale is $\tau$; the odds ratio is $\exp(\tau)$. A "77× reduction in odds" corresponds to $\exp(\tau) = 1/77$, i.e. $\tau = -\ln 77$.

*Percentage-point effect (ATE on response scale).* Let $\mu_0 \equiv \text{logit}^{-1}(\alpha)$ and $\mu_1 \equiv \text{logit}^{-1}(\alpha + \tau)$. The average treatment effect on the response scale is

$$\Delta \equiv \mathbb{E}[Y_j \mid D_j = 1] - \mathbb{E}[Y_j \mid D_j = 0] \; = \; \mu_1 - \mu_0 \quad \text{(in proportion units; multiply by 100 for pp).}$$

*Inference.* We estimate (7) by MLE and report Wald $z$-tests and 95% CIs. For $\tau$, a 95% CI for the odds ratio is $\exp\big(\hat{\tau} \pm 1.96 \, \text{SE}(\hat{\tau})\big)$. For $\Delta$, use the delta method with gradient $\big(\partial\Delta/\partial\alpha, \partial\Delta/\partial\tau\big) = \big(\mu_1(1 - \mu_1) - \mu_0(1 - \mu_0), \, \mu_1(1 - \mu_1)\big)$:

$$\text{Var}(\hat{\Delta}) \approx g^\top \text{Var}\begin{pmatrix}\hat{\alpha} \\ \hat{\tau}\end{pmatrix} g, \quad g = \begin{pmatrix}\mu_1(1 - \mu_1) - \mu_0(1 - \mu_0) \\ \mu_1(1 - \mu_1)\end{pmatrix},$$

and a 95% CI is $\hat{\Delta} \pm 1.96 \, \text{SE}(\hat{\Delta})$.

**Causal inference results.** Table 21 reports causal inference results. In particular, beta regression of PPA AttAcc@1 on treatment shows that AT-AT's non-robust term causes a $77\times$ reduction in leakage odds ($z{=}38.0$, $p{<}10^{-16}$, 6.5% reconstruction rate under **AT-AT** vs. 84% for control).

**Table 21:** Causal regressions of leakage (AttAcc@1) on AT-AT assignment. OLS 95% CIs use HC2 SEs with $t_{0.975,8} = 2.306$; Beta uses Wald ($\pm 1.96$SE) on the logit scale. Baseline is **NoDef**; the coefficient on **AT-AT (Treatment)** is the effect relative to NoDef. Odds ratio (AT-AT vs NoDef) from Beta: 0.0130 (95% CI: 0.0104–0.0162).

|  | **OLS ($\pm$ 95% CI)** | **Beta ($\pm$ 95% CI)** |
|---|---|---|
| Intercept (NoDef) | $0.842 \pm 0.029$ | $1.677 \pm 0.126$ |
| AT-AT (Treatment) | **-0.778 $\pm$ 0.029** | **-4.344 $\pm$ 0.224** |
| $R^2$ | 0.9976 (adj.) | 0.995 (pseudo) |

We can similarly identify the causal effect of **AT-AT** on L2 reconstruction distance. Specifically, we recompute the OLS regression replacing the response with our L2 reconstruction distance evaluation metric (note that there is no need for beta regression for L2 response). We find that treating with non-robust features via **AT-AT**'s reward increases the L2 reconstruction distance by $0.430 \pm 0.017$:

**Table 22:** OLS regression of reconstruction distance ($L_2$) on AT-AT assignment. 95% CIs use HC2 SEs with $t_{0.975,8} = 2.306$. Baseline is **NoDef**; the coefficient on **AT-AT (Treatment)** is the effect relative to NoDef.

|  | **OLS ($\pm$ 95% CI)** |
|---|---|
| Intercept (NoDef) | $0.786 \pm 0.012$ |
| AT-AT (Treatment) | **0.430 $\pm$ 0.017** |
| $R^2$ | 0.9973 (adj.) |

# I  MIA DEFENSE EXTENDED RESULTS

Table 23 presents full results and privacy metrics for AT-AT and all baselines run on FaceScrub and CelebA using the standard ResNet-152 architecture. These results are an extended tabular version of Fig. 4.

## I.1 QUALITATIVE COMPARISONS

Fig. 6 shows examples of original images side-by-side with their respective PPA attack reconstructions under each defense. To avoid cherry-picking images to include in this comparison, we display the reconstructions for which the evaluation model is maximally confident that the reconstruction class matches the original class.

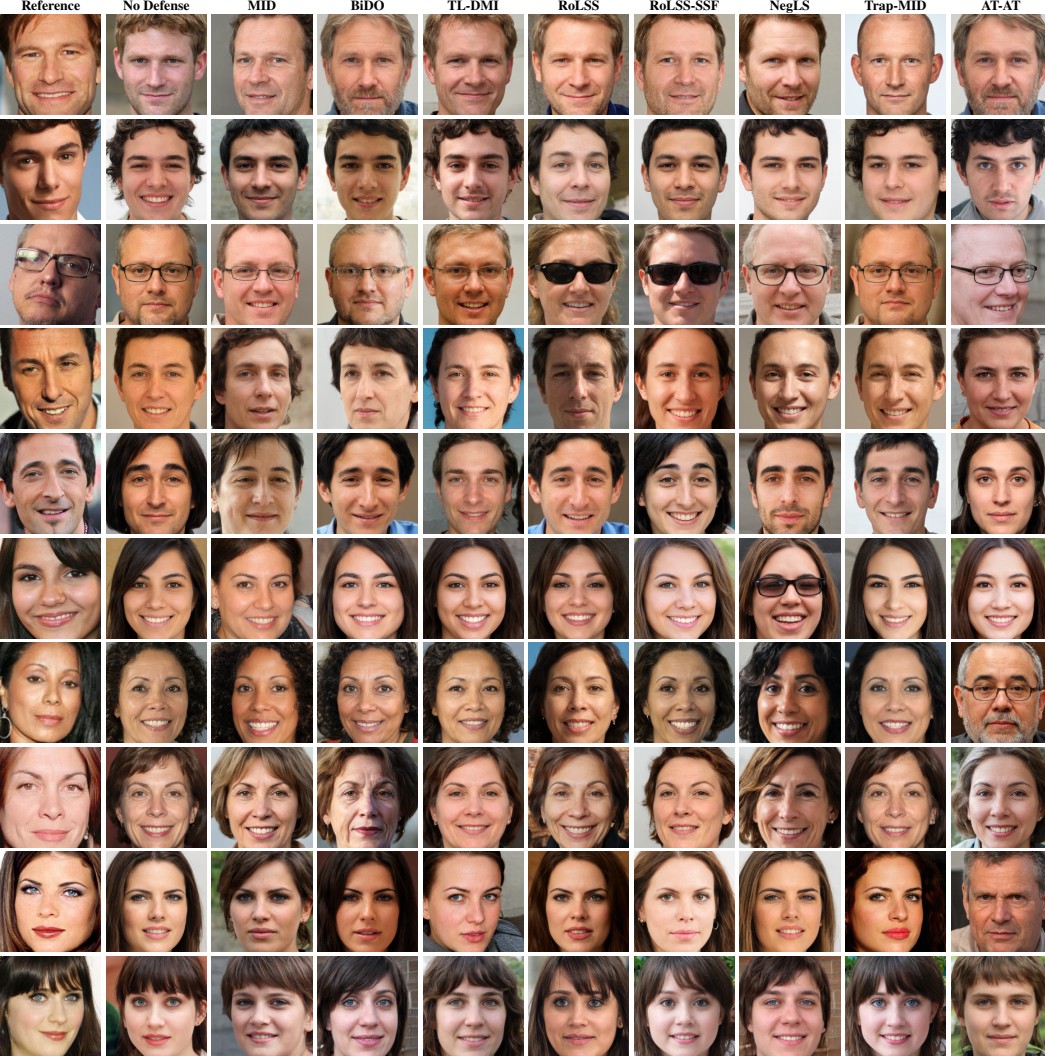

**Figure 6:** Image reconstructions generated from PPA attacks for each defense applied on a ResNet-152 trained on FaceScrub. To avoid cherry-picking, each image is selected by maximum confidence on the evaluation model (worst case).

## I.2 OTHER ARCHITECTURES AND DATASETS

Table 24 presents full results and privacy metrics for AT-AT and all baselines run on FaceScrub and CelebA using the DenseNet-169 architecture. Table 25 presents the results of the same experiments rerun using the lower-capacity ResNet-18 architecture.

Table 26a presents PPA attack results and privacy metrics for AT-AT and all baselines using the standard ResNet-152 architecture on the Stanford Dogs dataset. Table 26b presents the results of the same experiments for the lower-capacity ResNet-18 architecture.

### I.3 TABLES INCLUDING CONFIDENCE INTERVALS

We note that the main results tables, Tables 23 and 25, omitted confidence intervals for readability. We therefore include these confidence intervals separately below. Specifically, Tables 27 and 28 are expanded versions of the main MIA results (Tables 23 and 25) that add the deferred per-class or per-image confidence intervals. More specifically, for the AttAcc@1/AttAcc@5 metrics, we run a per-image 95% Wilson confidence interval with the proportion $p =$AttAcc@1/AttAcc@5, and the sample size $n$ equal to the total # of images on which attack accuracy is calculated (for FaceScrub, $n = 26500$, and for CelebA, $n = 3750$). For the L2 distance metric, as it is a mean over attack target classes, we run a 95% per-class $t$-interval on the list of L2 metric values for each attack target class (of length 530 for FaceScrub, and length 75 for CelebA).

## J REQUIRED LLM DISCLOSURE

LLMs were used in the creation of this paper to (1) read for typos, and (2) to suggest LaTeX formatting fixes (table spacing, algorithm watermark, etc.). LLMs were not used to write the paper, code the experiments, or for research ideation.

**Table 23:** Full privacy experiments results: AT-AT vs Baselines on standard **ResNet-152** architectures on CelebA and FaceScrub data under PPA, IF-GMI, and PPDG attacks.

| Data | Attack | Defense | TestAcc ↑ | AttAcc@1 ↓ | AttAcc@5 ↓ | L2 ↑ | EvalConf ↓ |
|---|---|---|---|---|---|---|---|
| *Defenses that attempt to increase privacy generically* | | | | | | | |
| FaceScrub | PPA | NoDef | 0.958 | 0.882 | 0.980 | 0.768 | 0.862 |
| | | MID | 0.950 | 0.391 | 0.699 | 0.950 | 0.364 |
| | | BiDO | 0.928 | 0.780 | 0.949 | 0.820 | 0.756 |
| | | TL-DMI | 0.911 | 0.190 | 0.435 | 1.057 | 0.178 |
| | | **AT-AT** | **0.939** | **0.058** | **0.190** | **1.224** | **0.055** |
| | IF-GMI | NoDef | 0.958 | 0.987 | 0.998 | 0.700 | 0.984 |
| | | MID | 0.950 | 0.391 | 0.668 | 1.002 | 0.365 |
| | | BiDO | 0.928 | 0.937 | 0.988 | 0.792 | 0.927 |
| | | TL-DMI | 0.911 | 0.420 | 0.694 | 0.946 | 0.401 |
| | | **AT-AT** | **0.939** | **0.084** | **0.238** | **1.216** | **0.080** |
| | PPDG | NoDef | 0.958 | 0.902 | 0.979 | 0.805 | 0.890 |
| | | MID | 0.950 | 0.343 | 0.583 | 1.056 | 0.315 |
| | | BiDO | 0.928 | 0.799 | 0.937 | 0.885 | 0.777 |
| | | TL-DMI | 0.911 | 0.142 | 0.338 | 1.125 | 0.132 |
| | | **AT-AT** | **0.939** | **0.022** | **0.084** | **1.353** | **0.021** |
| CelebA | PPA | NoDef | 0.838 | 0.497 | 0.753 | 0.843 | 0.473 |
| | | MID | 0.802 | 0.474 | 0.725 | 0.821 | 0.450 |
| | | BiDO | 0.747 | 0.362 | 0.629 | 0.923 | 0.340 |
| | | TL-DMI | 0.688 | 0.080 | 0.221 | 1.103 | 0.075 |
| | | TL-DMI-5 | 0.814 | 0.281 | 0.550 | 0.929 | 0.265 |
| | | **AT-AT** | **0.819** | **0.049** | **0.138** | **1.245** | **0.044** |
| | IF-GMI | NoDef | 0.838 | 0.793 | 0.940 | 0.770 | 0.773 |
| | | MID | 0.802 | 0.728 | 0.864 | 0.836 | 0.709 |
| | | BiDO | 0.747 | 0.597 | 0.834 | 0.867 | 0.575 |
| | | TL-DMI | 0.688 | 0.258 | 0.513 | 0.964 | 0.243 |
| | | TL-DMI-5 | 0.814 | 0.661 | 0.883 | 0.805 | 0.641 |
| | | **AT-AT** | **0.819** | **0.188** | **0.377** | **1.140** | **0.181** |
| | PPDG | NoDef | 0.838 | 0.559 | 0.796 | 0.863 | 0.536 |
| | | MID | 0.802 | 0.491 | 0.688 | 0.928 | 0.461 |
| | | BiDO | 0.747 | 0.389 | 0.647 | 0.956 | 0.368 |
| | | TL-DMI | 0.688 | 0.079 | 0.206 | 1.157 | 0.076 |
| | | TL-DMI-5 | 0.814 | 0.319 | 0.585 | 0.959 | 0.303 |
| | | **AT-AT** | **0.819** | **0.023** | **0.071** | **1.430** | **0.021** |
| *Defenses that suppress attack gradients* | | | | | | | |
| FaceScrub | PPA | NegLS | 0.906 | 0.160 | 0.365 | 1.151 | 0.152 |
| | | RoLSS | 0.886 | 0.474 | 0.760 | 0.858 | 0.450 |
| | | RoLSS-SSF | 0.869 | 0.343 | 0.635 | 0.918 | 0.322 |
| | | Trap-MID* | 0.949 | 0.587 | 0.830 | 0.920 | 0.556 |
| | | Trap-MID | 0.926 | 0.401 | 0.676 | 0.980 | 0.376 |
| | IF-GMI | NegLS | 0.906 | 0.147 | 0.352 | 1.199 | 0.142 |
| | | RoLSS | 0.886 | 0.522 | 0.785 | 0.884 | 0.499 |
| | | RoLSS-SSF | 0.869 | 0.373 | 0.657 | 0.956 | 0.354 |
| | | Trap-MID* | 0.949 | 0.714 | 0.792 | 0.939 | 0.705 |
| | | Trap-MID | 0.926 | 0.618 | 0.738 | 1.004 | 0.603 |
| | PPDG | NegLS | 0.906 | 0.105 | 0.245 | 1.247 | 0.101 |
| | | RoLSS | 0.886 | 0.456 | 0.742 | 0.910 | 0.432 |
| | | RoLSS-SSF | 0.869 | 0.294 | 0.558 | 0.999 | 0.272 |
| | | Trap-MID* | 0.949 | 0.634 | 0.813 | 0.958 | 0.616 |
| | | Trap-MID | 0.926 | 0.425 | 0.665 | 1.039 | 0.408 |
| CelebA | PPA | NegLS | 0.805 | 0.340 | 0.623 | 0.908 | 0.324 |
| | | RoLSS | 0.537 | 0.046 | 0.145 | 1.147 | 0.043 |
| | | RoLSS-SSF | 0.428 | 0.039 | 0.118 | 1.183 | 0.035 |
| | | Trap-MID* | 0.806 | 0.092 | 0.193 | 1.358 | 0.088 |
| | | Trap-MID | 0.847 | 0.330 | 0.569 | 0.944 | 0.309 |
| | IF-GMI | NegLS | 0.805 | 0.554 | 0.811 | 0.885 | 0.532 |
| | | RoLSS | 0.537 | 0.045 | 0.145 | 1.172 | 0.042 |
| | | RoLSS-SSF | 0.428 | 0.044 | 0.131 | 1.218 | 0.040 |
| | | Trap-MID* | 0.806 | 0.168 | 0.223 | 1.437 | 0.163 |
| | | Trap-MID | 0.847 | 0.417 | 0.601 | 1.004 | 0.403 |
| | PPDG | NegLS | 0.805 | 0.331 | 0.580 | 0.955 | 0.314 |
| | | RoLSS | 0.537 | 0.038 | 0.110 | 1.266 | 0.035 |
| | | RoLSS-SSF | 0.428 | 0.029 | 0.093 | 1.295 | 0.027 |
| | | Trap-MID* | 0.806 | 0.074 | 0.148 | 1.455 | 0.072 |
| | | Trap-MID | 0.847 | 0.331 | 0.556 | 1.035 | 0.315 |

**Table 24:** Full privacy experiments results: AT-AT vs Baselines on standard **DenseNet-169** architectures on CelebA and FaceScrub data under PPA, IF-GMI, and PPDG attacks.

| Data | Attack | Defense | TestAcc↑ | AttAcc@1↓ | AttAcc@5↓ | L2↑ | EvalConf↓ |
|------|--------|---------|----------|-----------|-----------|-----|-----------|
| *Defenses that attempt to increase privacy generically* | | | | | | | |
| FaceScrub | PPA | NoDef | 0.947 | 0.890 | 0.980 | 0.763 | 0.868 |
| | | MID | 0.949 | 0.595 | 0.835 | 0.869 | 0.561 |
| | | BiDO | 0.921 | 0.693 | 0.891 | 0.831 | 0.667 |
| | | TL-DMI | 0.921 | 0.294 | 0.575 | 0.996 | 0.278 |
| | | **AT-AT** | **0.933** | **0.058** | **0.190** | **1.194** | **0.057** |
| | IF-GMI | NoDef | 0.947 | 0.983 | 0.998 | 0.722 | 0.979 |
| | | MID | 0.949 | 0.621 | 0.808 | 0.951 | 0.587 |
| | | BiDO | 0.921 | 0.872 | 0.967 | 0.826 | 0.855 |
| | | TL-DMI | 0.921 | 0.627 | 0.845 | 0.885 | 0.606 |
| | | **AT-AT** | **0.933** | **0.124** | **0.303** | **1.157** | **0.116** |
| | PPDG | NoDef | 0.947 | 0.915 | 0.984 | 0.787 | 0.903 |
| | | MID | 0.949 | 0.537 | 0.749 | 0.970 | 0.510 |
| | | BiDO | 0.921 | 0.747 | 0.915 | 0.856 | 0.724 |
| | | TL-DMI | 0.921 | 0.329 | 0.586 | 1.017 | 0.311 |
| | | **AT-AT** | **0.933** | **0.037** | **0.118** | **1.305** | **0.035** |
| CelebA | PPA | NoDef | 0.856 | 0.503 | 0.788 | 0.800 | 0.480 |
| | | MID | 0.730 | 0.419 | 0.675 | 0.836 | 0.394 |
| | | BiDO | 0.611 | 0.132 | 0.313 | 1.021 | 0.123 |
| | | TL-DMI | 0.788 | 0.151 | 0.361 | 1.000 | 0.145 |
| | | **AT-AT** | **0.836** | **0.086** | **0.205** | **1.177** | **0.078** |
| | IF-GMI | NoDef | 0.856 | 0.820 | 0.943 | 0.755 | 0.804 |
| | | MID | 0.730 | 0.577 | 0.731 | 0.882 | 0.557 |
| | | BiDO | 0.611 | 0.199 | 0.409 | 1.005 | 0.192 |
| | | TL-DMI | 0.788 | 0.474 | 0.735 | 0.861 | 0.453 |
| | | **AT-AT** | **0.836** | **0.389** | **0.599** | **0.990** | **0.369** |
| | PPDG | NoDef | 0.856 | 0.594 | 0.826 | 0.819 | 0.570 |
| | | MID | 0.730 | 0.402 | 0.616 | 0.952 | 0.380 |
| | | BiDO | 0.611 | 0.142 | 0.311 | 1.062 | 0.135 |
| | | TL-DMI | 0.788 | 0.145 | 0.333 | 1.068 | 0.137 |
| | | **AT-AT** | **0.836** | **0.041** | **0.101** | **1.359** | **0.039** |
| *Defenses that suppress attack gradients* | | | | | | | |
| FaceScrub | PPA | NegLS | 0.924 | 0.208 | 0.450 | 1.107 | 0.202 |
| | | RoLSS | 0.732 | 0.098 | 0.267 | 1.116 | 0.093 |
| | | RoLSS-SSF | 0.886 | 0.462 | 0.761 | 0.882 | 0.438 |
| | | Trap-MID | 0.920 | 0.529 | 0.787 | 0.893 | 0.500 |
| | | Trap-MID* | 0.946 | 0.708 | 0.912 | 0.833 | 0.680 |
| | IF-GMI | NegLS | 0.924 | 0.448 | 0.714 | 1.047 | 0.433 |
| | | RoLSS | 0.732 | 0.096 | 0.253 | 1.105 | 0.088 |
| | | RoLSS-SSF | 0.886 | 0.551 | 0.823 | 0.877 | 0.531 |
| | | Trap-MID | 0.920 | 0.707 | 0.837 | 0.902 | 0.689 |
| | | Trap-MID* | 0.946 | 0.903 | 0.939 | 0.778 | 0.894 |
| | PPDG | NegLS | 0.924 | 0.239 | 0.478 | 1.145 | 0.230 |
| | | RoLSS | 0.732 | 0.069 | 0.197 | 1.160 | 0.065 |
| | | RoLSS-SSF | 0.886 | 0.486 | 0.746 | 0.908 | 0.465 |
| | | Trap-MID | 0.920 | 0.575 | 0.784 | 0.928 | 0.552 |
| | | Trap-MID* | 0.946 | 0.783 | 0.917 | 0.836 | 0.764 |
| CelebA | PPA | NegLS | 0.820 | 0.528 | 0.800 | 0.781 | 0.507 |
| | | RoLSS | 0.275 | 0.034 | 0.126 | 1.149 | 0.034 |
| | | RoLSS-SSF | 0.557 | 0.103 | 0.285 | 1.007 | 0.100 |
| | | Trap-MID | 0.681 | 0.195 | 0.404 | 0.976 | 0.185 |
| | | Trap-MID* | 0.711 | 0.198 | 0.443 | 0.957 | 0.188 |
| | IF-GMI | NegLS | 0.820 | 0.819 | 0.938 | 0.719 | 0.800 |
| | | RoLSS | 0.275 | 0.038 | 0.121 | 1.161 | 0.035 |
| | | RoLSS-SSF | 0.557 | 0.111 | 0.323 | 1.005 | 0.108 |
| | | Trap-MID | 0.681 | 0.264 | 0.461 | 1.044 | 0.250 |
| | | Trap-MID* | 0.711 | 0.412 | 0.650 | 0.942 | 0.393 |
| | PPDG | NegLS | 0.820 | 0.587 | 0.821 | 0.793 | 0.563 |
| | | RoLSS | 0.275 | 0.027 | 0.102 | 1.193 | 0.025 |
| | | RoLSS-SSF | 0.557 | 0.091 | 0.259 | 1.040 | 0.087 |
| | | Trap-MID | 0.681 | 0.164 | 0.358 | 1.079 | 0.157 |
| | | Trap-MID* | 0.711 | 0.224 | 0.453 | 1.032 | 0.211 |

**Table 25:** Additional privacy experiments results: Lower-capacity **ResNet-18** architectures on CelebA and FaceScrub data under PPA and IF-GMI attacks.

| Data | Attack | Defense | TestAcc ↑ | AttAcc@1 ↓ | AttAcc@5 ↓ | L2 ↑ | EvalConf ↓ |
|---|---|---|---|---|---|---|---|
| *Defenses that attempt to increase privacy generically* | | | | | | | |
| FaceScrub | PPA | NoDef | 0.950 | 0.901 | 0.986 | 0.761 | 0.881 |
| | | MID | 0.924 | 0.714 | 0.903 | 0.815 | 0.688 |
| | | BiDO | 0.884 | 0.530 | 0.811 | 0.915 | 0.504 |
| | | TL-DMI | 0.906 | 0.240 | 0.505 | 1.020 | 0.225 |
| | | **AT-AT** | **0.921** | **0.110** | **0.297** | **1.143** | **0.104** |
| | IF-GMI | NoDef | 0.950 | 0.983 | 0.996 | 0.740 | 0.979 |
| | | MID | 0.924 | 0.776 | 0.840 | 0.899 | 0.764 |
| | | BiDO | 0.884 | 0.782 | 0.925 | 0.887 | 0.764 |
| | | TL-DMI | 0.906 | 0.515 | 0.760 | 0.935 | 0.492 |
| | | **AT-AT** | **0.921** | **0.283** | **0.520** | **1.082** | **0.268** |
| CelebA | PPA | NoDef | 0.859 | 0.634 | 0.859 | 0.774 | 0.608 |
| | | MID | 0.737 | 0.386 | 0.671 | 0.838 | 0.362 |
| | | BiDO | 0.625 | 0.146 | 0.359 | 1.009 | 0.136 |
| | | TL-DMI | 0.752 | 0.153 | 0.339 | 1.013 | 0.142 |
| | | TL-DMI-5 | 0.831 | 0.324 | 0.602 | 0.886 | 0.302 |
| | | **AT-AT[†a]** | **0.825** | **0.296** | **0.561** | **0.945** | **0.279** |
| | | **AT-AT[†b]** | **0.787** | **0.155** | **0.350** | **1.029** | **0.144** |
| | IF-GMI | NoDef | 0.859 | 0.913 | 0.970 | 0.708 | 0.903 |
| | | MID | 0.737 | 0.550 | 0.740 | 0.889 | 0.529 |
| | | BiDO | 0.625 | 0.368 | 0.629 | 0.908 | 0.347 |
| | | TL-DMI | 0.752 | 0.439 | 0.710 | 0.872 | 0.421 |
| | | TL-DMI-5 | 0.831 | 0.750 | 0.922 | 0.751 | 0.728 |
| | | **AT-AT[†a]** | **0.825** | **0.783** | **0.912** | **0.799** | **0.764** |
| | | **AT-AT[†b]** | **0.787** | **0.617** | **0.806** | **0.856** | **0.599** |
| *Defenses that suppress attack gradients* | | | | | | | |
| FaceScrub | PPA | NegLS | 0.929 | 0.649 | 0.888 | 0.876 | 0.624 |
| | | RoLSS | 0.950 | 0.858 | 0.973 | 0.746 | 0.837 |
| | | RoLSS-SSF | 0.950 | 0.849 | 0.973 | 0.751 | 0.828 |
| | | Trap-MID* | 0.927 | 0.502 | 0.752 | 0.949 | 0.475 |
| | | Trap-MID | 0.928 | 0.735 | 0.930 | 0.831 | 0.704 |
| | IF-GMI | NegLS | 0.929 | 0.932 | 0.989 | 0.824 | 0.923 |
| | | RoLSS | 0.950 | 0.979 | 0.996 | 0.683 | 0.975 |
| | | RoLSS-SSF | 0.950 | 0.973 | 0.992 | 0.699 | 0.969 |
| | | Trap-MID* | 0.927 | 0.687 | 0.770 | 0.960 | 0.673 |
| | | Trap-MID | 0.928 | 0.918 | 0.970 | 0.790 | 0.910 |
| CelebA | PPA | NegLS | 0.838 | 0.667 | 0.887 | 0.730 | 0.640 |
| | | RoLSS | 0.830 | 0.506 | 0.769 | 0.809 | 0.474 |
| | | RoLSS-SSF | 0.839 | 0.519 | 0.764 | 0.810 | 0.490 |
| | | Trap-MID* | 0.811 | 0.019 | 0.031 | 1.571 | 0.018 |
| | | Trap-MID | 0.830 | 0.443 | 0.700 | 0.854 | 0.423 |
| | IF-GMI | NegLS | 0.838 | 0.953 | 0.993 | 0.649 | 0.944 |
| | | RoLSS | 0.830 | 0.724 | 0.904 | 0.778 | 0.705 |
| | | RoLSS-SSF | 0.839 | 0.765 | 0.909 | 0.775 | 0.742 |
| | | Trap-MID* | 0.811 | 0.088 | 0.134 | 1.482 | 0.084 |
| | | Trap-MID | 0.830 | 0.670 | 0.797 | 0.911 | 0.647 |

*Notes:* [†a] and [†b] denote two sets of hyperparameters for AT-AT (see Table 6).

**Table 26:** Additional privacy experiments results: PPA attack on Stanford Dogs data using the pre-trained *dog* StyleGAN-2.

**(a)** ResNet-152

| Data | Attack | Defense | TestAcc ↑ | AttAcc@1 ↓ | AttAcc@5 ↓ | L2 ↑ | EvalConf ↓ |
|---|---|---|---|---|---|---|---|
| | | *Defenses that attempt to increase privacy generically* | | | | | |
| Stanford Dogs | PPA | NoDef | 0.867 | 0.820 | 0.957 | 106.6 | 0.708 |
| | | MID | 0.820 | 0.484 | 0.759 | 241.3 | 0.381 |
| | | BiDO | 0.817 | 0.717 | 0.916 | 139.8 | 0.727 |
| | | TL-DMI | 0.852 | 0.754 | 0.930 | 127.3 | 0.605 |
| | | **AT-AT** | 0.814 | **0.386** | **0.632** | **275.8** | **0.235** |
| | | *Defenses that suppress attack gradients* | | | | | |
| | PPA | NegLS | 0.816 | 0.625 | 0.858 | 181.7 | 0.532 |
| | | RoLSS | 0.684 | 0.551 | 0.820 | 198.7 | 0.712 |
| | | RoLSS-SSF | 0.673 | 0.481 | 0.789 | 212.5 | 0.415 |

**(b)** ResNet-18

| Data | Attack | Defense | TestAcc ↑ | AttAcc@1 ↓ | AttAcc@5 ↓ | L2 ↑ | EvalConf ↓ |
|---|---|---|---|---|---|---|---|
| | | *Defenses that attempt to increase privacy generically* | | | | | |
| Stanford Dogs | PPA | NoDef | 0.776 | 0.823 | 0.949 | 105.7 | 0.759 |
| | | MID | 0.675 | 0.680 | 0.850 | 167.0 | 0.615 |
| | | BiDO | 0.660 | 0.626 | 0.845 | 176.1 | 0.563 |
| | | TL-DMI | 0.777 | 0.774 | 0.933 | 118.1 | 0.709 |
| | | **AT-AT** | 0.711 | **0.468** | **0.724** | **238.7** | **0.415** |
| | | *Defenses that suppress attack gradients* | | | | | |
| | PPA | NegLS | 0.778 | 0.828 | 0.953 | 113.2 | 0.745 |
| | | RoLSS | 0.745 | 0.793 | 0.939 | 113.8 | 0.730 |
| | | RoLSS-SSF | 0.753 | 0.786 | 0.935 | 119.2 | 0.721 |

**Table 27:** Full privacy experiments results: AT-AT vs Baselines on **ResNet-152** on CelebA and FaceScrub under PPA and IF-GMI. Entries show mean $\pm$ 95% CI half-width for non-confidence metrics.

| Data | Attack | Defense | TestAcc↑ | AttAcc@1↓ | AttAcc@5↓ | L2↑ | EvalConf↓ |
|------|--------|---------|----------|-----------|-----------|-----|-----------|
| *Defenses that attempt to increase privacy generically* | | | | | | | |
| FaceScrub | PPA | NoDef | 0.958 | $0.882 \pm 0.004$ | $0.980 \pm 0.002$ | $0.768 \pm 0.010$ | 0.862 |
| | | MID | 0.950 | $0.391 \pm 0.006$ | $0.699 \pm 0.006$ | $0.950 \pm 0.010$ | 0.364 |
| | | BiDO | 0.928 | $0.780 \pm 0.005$ | $0.949 \pm 0.003$ | $0.820 \pm 0.010$ | 0.756 |
| | | TL-DMI | 0.911 | $0.190 \pm 0.005$ | $0.435 \pm 0.006$ | $1.057 \pm 0.011$ | 0.178 |
| | | **AT-AT** | 0.939 | $0.058 \pm 0.003$ | $0.190 \pm 0.005$ | $1.225 \pm 0.013$ | 0.055 |
| | IF-GMI | NoDef | 0.958 | $0.987 \pm 0.001$ | $0.998 \pm 0.001$ | $0.700 \pm 0.009$ | 0.984 |
| | | MID | 0.950 | $0.391 \pm 0.006$ | $0.668 \pm 0.006$ | $1.002 \pm 0.010$ | 0.365 |
| | | BiDO | 0.928 | $0.937 \pm 0.003$ | $0.988 \pm 0.001$ | $0.792 \pm 0.009$ | 0.927 |
| | | TL-DMI | 0.911 | $0.420 \pm 0.006$ | $0.694 \pm 0.006$ | $0.946 \pm 0.010$ | 0.401 |
| | | **AT-AT** | 0.939 | $0.084 \pm 0.003$ | $0.238 \pm 0.005$ | $1.216 \pm 0.013$ | 0.080 |
| CelebA | PPA | NoDef | 0.838 | $0.497 \pm 0.016$ | $0.753 \pm 0.014$ | $0.843 \pm 0.035$ | 0.473 |
| | | MID | 0.802 | $0.474 \pm 0.016$ | $0.725 \pm 0.014$ | $0.821 \pm 0.035$ | 0.450 |
| | | BiDO | 0.747 | $0.362 \pm 0.015$ | $0.629 \pm 0.015$ | $0.923 \pm 0.037$ | 0.340 |
| | | TL-DMI | 0.688 | $0.080 \pm 0.009$ | $0.221 \pm 0.013$ | $1.103 \pm 0.037$ | 0.075 |
| | | TL-DMI-5 | 0.814 | $0.281 \pm 0.014$ | $0.550 \pm 0.016$ | $0.929 \pm 0.035$ | 0.265 |
| | | **AT-AT** | 0.819 | $0.049 \pm 0.007$ | $0.138 \pm 0.011$ | $1.245 \pm 0.047$ | 0.044 |
| | IF-GMI | NoDef | 0.838 | $0.793 \pm 0.013$ | $0.940 \pm 0.008$ | $0.770 \pm 0.034$ | 0.773 |
| | | MID | 0.802 | $0.728 \pm 0.014$ | $0.864 \pm 0.011$ | $0.836 \pm 0.037$ | 0.709 |
| | | BiDO | 0.747 | $0.597 \pm 0.016$ | $0.834 \pm 0.012$ | $0.867 \pm 0.036$ | 0.575 |
| | | TL-DMI | 0.688 | $0.258 \pm 0.014$ | $0.513 \pm 0.016$ | $0.964 \pm 0.037$ | 0.243 |
| | | TL-DMI-5 | 0.814 | $0.661 \pm 0.015$ | $0.883 \pm 0.010$ | $0.805 \pm 0.033$ | 0.641 |
| | | **AT-AT** | 0.819 | $0.188 \pm 0.013$ | $0.377 \pm 0.016$ | $1.140 \pm 0.051$ | 0.181 |
| *Defenses that suppress attack gradients* | | | | | | | |
| FaceScrub | PPA | NegLS | 0.906 | $0.160 \pm 0.004$ | $0.365 \pm 0.006$ | $1.151 \pm 0.016$ | 0.152 |
| | | RoLSS | 0.886 | $0.474 \pm 0.006$ | $0.760 \pm 0.005$ | $0.858 \pm 0.011$ | 0.450 |
| | | RoLSS-SSF | 0.869 | $0.343 \pm 0.006$ | $0.635 \pm 0.006$ | $0.918 \pm 0.012$ | 0.322 |
| | | Trap-MID | 0.926 | $0.401 \pm 0.016$ | $0.676 \pm 0.015$ | $0.980 \pm 0.039$ | 0.376 |
| | | Trap-MID* | 0.949 | $0.587 \pm 0.016$ | $0.830 \pm 0.012$ | $0.920 \pm 0.030$ | 0.556 |
| | IF-GMI | NegLS | 0.906 | $0.147 \pm 0.004$ | $0.352 \pm 0.006$ | $1.199 \pm 0.014$ | 0.142 |
| | | RoLSS | 0.886 | $0.522 \pm 0.006$ | $0.785 \pm 0.005$ | $0.884 \pm 0.011$ | 0.499 |
| | | RoLSS-SSF | 0.869 | $0.373 \pm 0.006$ | $0.657 \pm 0.006$ | $0.956 \pm 0.011$ | 0.354 |
| | | Trap-MID | 0.926 | $0.618 \pm 0.016$ | $0.738 \pm 0.014$ | $1.004 \pm 0.046$ | 0.603 |
| | | Trap-MID* | 0.949 | $0.713 \pm 0.014$ | $0.792 \pm 0.013$ | $0.939 \pm 0.039$ | 0.705 |
| CelebA | PPA | NegLS | 0.805 | $0.340 \pm 0.015$ | $0.623 \pm 0.016$ | $0.908 \pm 0.037$ | 0.324 |
| | | RoLSS | 0.537 | $0.046 \pm 0.007$ | $0.145 \pm 0.011$ | $1.147 \pm 0.044$ | 0.043 |
| | | RoLSS-SSF | 0.428 | $0.039 \pm 0.006$ | $0.118 \pm 0.010$ | $1.183 \pm 0.045$ | 0.035 |
| | | Trap-MID | 0.847 | $0.330 \pm 0.015$ | $0.569 \pm 0.016$ | $0.944 \pm 0.051$ | 0.309 |
| | | Trap-MID* | 0.806 | $0.092 \pm 0.009$ | $0.193 \pm 0.013$ | $1.358 \pm 0.066$ | 0.088 |
| | IF-GMI | NegLS | 0.805 | $0.554 \pm 0.016$ | $0.811 \pm 0.013$ | $0.885 \pm 0.035$ | 0.532 |
| | | RoLSS | 0.537 | $0.045 \pm 0.007$ | $0.145 \pm 0.011$ | $1.172 \pm 0.053$ | 0.042 |
| | | RoLSS-SSF | 0.428 | $0.044 \pm 0.007$ | $0.131 \pm 0.011$ | $1.218 \pm 0.056$ | 0.040 |
| | | Trap-MID | 0.847 | $0.417 \pm 0.016$ | $0.600 \pm 0.016$ | $1.004 \pm 0.052$ | 0.403 |
| | | Trap-MID* | 0.806 | $0.168 \pm 0.012$ | $0.223 \pm 0.013$ | $1.437 \pm 0.058$ | 0.163 |

**Table 28:** Full privacy experiments results: AT-AT vs Baselines on **ResNet-18** on CelebA and FaceScrub under PPA and IF-GMI. Entries show mean $\pm$ 95% CI half-width for non-confidence metrics.

| Data | Attack | Defense | TestAcc ↑ | AttAcc@1 ↓ | AttAcc@5 ↓ | L2 ↑ | EvalConf ↓ |
|---|---|---|---|---|---|---|---|
| *Defenses that attempt to increase privacy generically* | | | | | | | |
| FaceScrub | PPA | NoDef | 0.950 | $0.901 \pm 0.004$ | $0.986 \pm 0.002$ | $0.761 \pm 0.010$ | 0.881 |
| | | MID | 0.924 | $0.714 \pm 0.006$ | $0.903 \pm 0.004$ | $0.815 \pm 0.012$ | 0.688 |
| | | BiDO | 0.884 | $0.530 \pm 0.006$ | $0.811 \pm 0.005$ | $0.915 \pm 0.011$ | 0.504 |
| | | TL-DMI | 0.906 | $0.240 \pm 0.005$ | $0.505 \pm 0.006$ | $1.020 \pm 0.011$ | 0.225 |
| | | AT-AT | 0.921 | $0.110 \pm 0.004$ | $0.297 \pm 0.006$ | $1.143 \pm 0.013$ | 0.104 |
| | IF-GMI | NoDef | 0.950 | $0.983 \pm 0.002$ | $0.996 \pm 0.001$ | $0.740 \pm 0.010$ | 0.979 |
| | | MID | 0.924 | $0.776 \pm 0.005$ | $0.840 \pm 0.004$ | $0.899 \pm 0.013$ | 0.764 |
| | | BiDO | 0.884 | $0.782 \pm 0.005$ | $0.925 \pm 0.003$ | $0.887 \pm 0.011$ | 0.764 |
| | | TL-DMI | 0.906 | $0.515 \pm 0.006$ | $0.760 \pm 0.006$ | $0.935 \pm 0.011$ | 0.492 |
| | | AT-AT | 0.921 | $0.283 \pm 0.006$ | $0.520 \pm 0.006$ | $1.082 \pm 0.014$ | 0.268 |
| CelebA | PPA | NoDef | 0.859 | $0.634 \pm 0.016$ | $0.859 \pm 0.011$ | $0.774 \pm 0.033$ | 0.608 |
| | | MID | 0.737 | $0.386 \pm 0.016$ | $0.671 \pm 0.015$ | $0.838 \pm 0.036$ | 0.362 |
| | | BiDO | 0.625 | $0.146 \pm 0.011$ | $0.359 \pm 0.016$ | $1.009 \pm 0.037$ | 0.136 |
| | | TL-DMI | 0.752 | $0.153 \pm 0.012$ | $0.339 \pm 0.016$ | $1.013 \pm 0.037$ | 0.142 |
| | | TL-DMI-5 | 0.831 | $0.324 \pm 0.015$ | $0.602 \pm 0.016$ | $0.886 \pm 0.035$ | 0.302 |
| | | AT-AT[†a] | 0.825 | $0.296 \pm 0.015$ | $0.561 \pm 0.016$ | $0.945 \pm 0.038$ | 0.279 |
| | | AT-AT[†b] | 0.787 | $0.156 \pm 0.012$ | $0.350 \pm 0.016$ | $1.029 \pm 0.039$ | 0.145 |
| | IF-GMI | NoDef | 0.859 | $0.913 \pm 0.009$ | $0.970 \pm 0.006$ | $0.708 \pm 0.031$ | 0.903 |
| | | MID | 0.737 | $0.550 \pm 0.016$ | $0.740 \pm 0.014$ | $0.889 \pm 0.045$ | 0.529 |
| | | BiDO | 0.625 | $0.368 \pm 0.016$ | $0.629 \pm 0.016$ | $0.908 \pm 0.033$ | 0.347 |
| | | TL-DMI | 0.752 | $0.439 \pm 0.016$ | $0.710 \pm 0.015$ | $0.872 \pm 0.036$ | 0.421 |
| | | TL-DMI-5 | 0.831 | $0.750 \pm 0.014$ | $0.922 \pm 0.009$ | $0.751 \pm 0.029$ | 0.728 |
| | | AT-AT[†a] | 0.825 | $0.783 \pm 0.014$ | $0.912 \pm 0.009$ | $0.799 \pm 0.037$ | 0.764 |
| | | AT-AT[†b] | 0.787 | $0.617 \pm 0.016$ | $0.806 \pm 0.013$ | $0.856 \pm 0.040$ | 0.599 |
| *Defenses that suppress attack gradients* | | | | | | | |
| FaceScrub | PPA | NegLS | 0.929 | $0.649 \pm 0.006$ | $0.888 \pm 0.004$ | $0.876 \pm 0.013$ | 0.624 |
| | | RoLSS | 0.950 | $0.858 \pm 0.005$ | $0.973 \pm 0.002$ | $0.746 \pm 0.010$ | 0.837 |
| | | RoLSS-SSF | 0.950 | $0.849 \pm 0.005$ | $0.973 \pm 0.002$ | $0.751 \pm 0.010$ | 0.828 |
| | | Trap-MID | 0.928 | $0.735 \pm 0.014$ | $0.930 \pm 0.008$ | $0.831 \pm 0.027$ | 0.704 |
| | | Trap-MID* | 0.927 | $0.502 \pm 0.016$ | $0.752 \pm 0.014$ | $0.949 \pm 0.035$ | 0.475 |
| | IF-GMI | NegLS | 0.929 | $0.932 \pm 0.004$ | $0.989 \pm 0.001$ | $0.824 \pm 0.014$ | 0.923 |
| | | RoLSS | 0.950 | $0.979 \pm 0.002$ | $0.996 \pm 0.001$ | $0.683 \pm 0.009$ | 0.975 |
| | | RoLSS-SSF | 0.950 | $0.973 \pm 0.002$ | $0.992 \pm 0.001$ | $0.699 \pm 0.010$ | 0.969 |
| | | Trap-MID | 0.928 | $0.918 \pm 0.009$ | $0.970 \pm 0.005$ | $0.790 \pm 0.025$ | 0.910 |
| | | Trap-MID* | 0.927 | $0.687 \pm 0.015$ | $0.770 \pm 0.013$ | $0.960 \pm 0.043$ | 0.673 |
| CelebA | PPA | NegLS | 0.838 | $0.667 \pm 0.015$ | $0.887 \pm 0.011$ | $0.730 \pm 0.030$ | 0.640 |
| | | RoLSS | 0.830 | $0.506 \pm 0.016$ | $0.769 \pm 0.014$ | $0.809 \pm 0.034$ | 0.474 |
| | | RoLSS-SSF | 0.839 | $0.519 \pm 0.016$ | $0.764 \pm 0.014$ | $0.810 \pm 0.033$ | 0.490 |
| | | Trap-MID | 0.830 | $0.443 \pm 0.016$ | $0.700 \pm 0.015$ | $0.854 \pm 0.050$ | 0.423 |
| | | Trap-MID* | 0.811 | $0.019 \pm 0.004$ | $0.032 \pm 0.006$ | $1.571 \pm 0.040$ | 0.018 |
| | IF-GMI | NegLS | 0.838 | $0.953 \pm 0.007$ | $0.993 \pm 0.003$ | $0.649 \pm 0.031$ | 0.944 |
| | | RoLSS | 0.830 | $0.724 \pm 0.014$ | $0.904 \pm 0.010$ | $0.778 \pm 0.037$ | 0.705 |
| | | RoLSS-SSF | 0.839 | $0.765 \pm 0.014$ | $0.909 \pm 0.009$ | $0.775 \pm 0.034$ | 0.742 |
| | | Trap-MID | 0.830 | $0.669 \pm 0.015$ | $0.797 \pm 0.013$ | $0.911 \pm 0.062$ | 0.647 |
| | | Trap-MID* | 0.810 | $0.088 \pm 0.009$ | $0.134 \pm 0.011$ | $1.482 \pm 0.047$ | 0.084 |

*Notes:* [†a] and [†b] denote two AT-AT hyperparameter settings (see Table 6).

