# OpenReview forum: "Reducing information dependency does not cause training data privacy. Adversarially non-robust features do."
_ICLR.cc/2026/Conference — ICLR 2026 Poster_

### Official Review · Reviewer_1QzK · 2025-10-16

**Soundness:** 2
**Presentation:** 3
**Contribution:** 4
**Rating:** 6
**Confidence:** 4

**Summary:**

This paper challenges the prevailing view that information dependency causes privacy leakage in machine learning models. It compellingly argues that vulnerability to reconstruction attacks stems from the model's reliance on "robust," human-interpretable features. Based on this, it introduces Anti-Adversarial Training (AT-AT), a novel method that intentionally trains the model to use "non-robust," human-imperceptible features to obscure visually sensitive information from attackers.


I recommend a Weak Accept.
- The paper's excellent and fundamental contribution in identifying that robust features are the root cause of privacy leakage. However, I find its proposed solution, which sacrifices model robustness and interpretability for privacy, to be a questionable direction that I do not endorse.

**Strengths:**

- Fundamental and Novel Contribution: The paper's primary strength is its rigorous and convincing refutation of the widely-held "information dependency → leakage" hypothesis. The identification of the connection between robust features and privacy vulnerability is a fundamental, paradigm-shifting insight that could redirect future research in this area.

- Strong Empirical Evidence: The three initial experiments are exceptionally well-designed and counter-intuitive, providing strong evidence to dismantle the status quo. The findings—that effective defenses don't reduce information dependency metrics, that rote-memorizing models are private, and that data from massively pixel-deleted images can still be reconstructed—are surprising and impactful.


- Revealing a New Trade-off: The discovery and formalization of the "privacy-robustness trade-off" is a significant conceptual contribution in itself. It provides a new and valuable lens for understanding the inherent tensions in designing secure and private machine learning models.

**Weaknesses:**

- Undermines Model Interpretability and Trustworthiness: A core goal of building reliable AI is for models to learn features that are robust and align with human perception, making them more interpretable. The AT-AT framework intentionally subverts this principle by compelling the model to rely on non-robust, human-imperceptible features for classification. While this may protect visual privacy, it achieves it by making the model's decision-making process opaque and semantically meaningless to humans. This is a significant step backward for explainable and trustworthy AI.

- Creates Intentionally Fragile Models: The direct consequence of relying on non-robust features is an extreme vulnerability to adversarial attacks. The authors acknowledge this as a trade-off, but I argue that deliberately engineering models to be fragile is not an acceptable solution. For years, the community has worked to make models more robust; this approach reverses that progress, trading one major security flaw (privacy leakage) for another (vulnerability to adversarial manipulation). This does not seem like a net gain for model security.

While the paper's contribution in identifying the role of robust features in privacy leakage is excellent, I do not endorse the proposed solution of relying on non-robust features as a viable or "correct" way to build private models. The solution sacrifices too much in terms of robustness and interpretability.

**Questions:**

- The AT-AT framework appears to encourage models to rely on non-robust, human-imperceptible features for classification. Could the authors clarify how this design aligns with the broader goal of developing interpretable and trustworthy AI systems?

- Have the authors considered integrating interpretability-preserving constraints or perceptual alignment mechanisms to mitigate the opacity introduced by adversarial feature reliance?

- Since the framework intentionally leverages non-robust features, how do the authors ensure that the resulting models are not excessively fragile to adversarial perturbations or distributional shifts?

- In the broader security context, how do the authors justify deliberately reducing model robustness as a viable long-term strategy for privacy protection?


While the paper's contribution in identifying the role of robust features in privacy leakage is excellent, I do not endorse the proposed solution of relying on non-robust features as a viable or "correct" way to build private models. The solution sacrifices too much in terms of robustness and interpretability.

---

> ### Author Response · Authors · 2025-11-21
>
> **We sincerely thank you for recognizing our work's _"fundamental, paradigm-shifting insight"_ and _"excellent and compelling experimental evidence."_ We understand that your hesitation and reason for a score of 6 instead of "full accept" arises from the question of whether AT-AT's use of non-robust features conflicts with Trustworthy ML goals.**
>
> We agree that robustness and interpretability are essential. **Our paper directly advances the Trustworthy ML agenda** you highlight. The paper's explicit message is _not_ that practitioners should use fragile models, but that:
>
> **Understanding and exposing hidden robustness losses is essential to building trustworthy systems and avoiding unintentional failure modes.**
>
> Without our findings, practitioners would continue deploying generic privacy defenses (MID, BiDO, TL-DMI) that _unknowingly_ introduce severe vulnerabilities to adversarial examples. This is precisely the kind of hidden failure that Trustworthy ML seeks to prevent.
>
> Our paper addresses this by making the hidden tradeoff experimentally testable, quantifying it, giving practitioners tunable control, and showing how robustness is an essential design axis for future privacy research. This directly advances the Trustworthy ML goals motivating your concerns.
>
> **Below, we show where the manuscript already explicitly states these points, using your concerns as the organizing framework. We strongly feel that this is a high impact paper. We would appreciate it if you would consider improving your score based on the clarifications below.**

---

> > ### Author Response · Authors · 2025-11-21
> >
> > - **1\. The paper explicitly states that AT-AT is _not_ proposing that we start using fragile or untrustworthy models.**
> >
> >
> >   - **Instead, AT-AT's**  **core contribution is that it gives us an instrument that allows us to formally and causally test whether non-robust features (unexpectedly) cause privacy (by explicitly controlling the amount of emphasis on non-robustness). The result that emerges from that test, that non-robust features cause privacy, is our core result and most impactful finding.**
> >
> >   - Your review states: "I do not endorse the proposed solution of relying on non-robust features…"
> >
> >  -  But the paper emphasizes that AT-AT is **not** an argument that we should _begin_ using non-robust features to address privacy. Instead, it is a scientific instrument that diagnoses an unknown tradeoff that already exists, and that all previous generic defenses exploited _unintentionally_.
> >
> >  -  Specifically, the paper's *Main Contribution* section states that "MIA defenses _unintentionally_ make a tradeoff: by shifting towards non-robust features, they strip semantically meaningful information from the model, but open significant new vulnerabilities to adversarial examples" (lines 121-126).
> >
> >  -  The tradeoff is _inherent_, not introduced by AT-AT. Our *Main Contribution* section emphasizes:
> >
> >       - "_…our results suggest an inherent privacy vulnerability associated with robust representations_…" (lines 155-7).
> >
> >  -  This is a scientific finding. The best we can hope for is the ability to measure, understand, and control this tradeoff, which is what we obtain with AT-AT.

---

> > > ### Author Response · Authors · 2025-11-21
> > >
> > > - **2\. AT-AT does not blindly sacrifice robustness. Instead, it reveals the hidden tradeoff and makes it controllable and tunable via the parameter** λ.
> > >
> > >   - We appreciate that your main concern was that AT-AT makes models fragile/brittle/non-robust. The paper explicitly clarifies that, in order to test whether fragility _causes_ privacy, we require a tool to manipulate fragility (e.g. the causal "treatment"). AT-AT provides this tool. In other words, unlike previous defenses, AT-AT makes the privacy-robustness tradeoff **controllable**, not accidental, so we can expose it to experimental testing.
> > >
> > >  -  Our Main Contribution section carefully emphasizes:
> > >
> > >     - "As with other generic defenses, AT-AT's performance comes at the cost of increased vulnerability to adversarial examples. However, in our case, this vulnerability cost is a tunable parameter, offering a new design axis for private models" (lines 141-143).
> > >
> > >  This line directly addresses your concern: AT-AT provides **control**, not total sacrifice.
> > >
> > >  -  **Why this advances Trustworthy ML:**
> > > Prior defenses forced practitioners into accidental robustness failures. In contrast, AT-AT exposes the frontier so they can intentionally choose (1)  via λ the appropriate privacy-robustness balance for their specific application, and also (2) via ε the size of adversarial perturbations they care about for their application. These are direct Trustworthy ML benefits.

---

> > > > ### Author Response · Authors · 2025-11-21
> > > >
> > > > - **3\. The paper also advances Trustworthy ML by explicitly quantifying the "robustness cost of privacy gain," making the tradeoff measurable and controllable.**
> > > >
> > > >   - Because we fully share your motivation to build Trustworthy ML, we devote substantial space to **measuring and reporting** the robustness cost of privacy gain for all baselines and for AT-AT.
> > > >
> > > >  -  **The manuscript explicitly states:** "_We can compute the adversarial robustness cost of MIA privacy gain… reducing MIA leakage by 1 pp corresponds to a statistically significant_ **_0.31-5.4 pp_** _decrease in robust accuracy (depending on ϵ)._"  (Main Contribution lines **121-126**, with full results at 380-389, and detailed discussion in Appendix G.1).
> > > >
> > > >  -  This is the **first rigorous quantification** of this tradeoff in the literature, We do **\*not\*** promote fragility; we provide the _first rigorous quantification_ of how robustness is affected, enabling a deliberate choice for trustworthy deployment.
> > > >
> > > > -   We also provide extended robustness results across 4 attacks × 4 ε values (App. F.4, from line 1472) for all baselines and AT-AT (run with a privacy-maximizing λ, which naturally yields worse robust accuracy per Fig. 2 (line 304) and the "robustness cost of privacy gain" quoted above).
> > > >
> > > >   This gives practitioners complete visibility into the frontier and eliminates hidden robustness failures, again directly advancing Trustworthy ML.

---

> ### Author Response · Authors · 2025-11-21
>
> -  **Final thoughts:**
>   **We appreciate your thoughtful review and hope the citations above clarify that the manuscript already addresses your concerns explicitly, both at the level of the Main Contributions sections and with extended results tables.**
>
>
>     **We deeply share your emphasis on Trustworthy ML. Indeed, the motivation of the paper is to reveal hidden robustness failures so that the community can avoid them in future privacy defenses.**
>
>
>     **We hope this resolves the misunderstanding and enables you to increase your recommendation and score to reflect the strength and Trustworthy-ML relevance of the contribution.**

---

> > ### Comment · Reviewer_1QzK · 2025-11-24
> >
> > Your response has addressed most of my concerns. I will increase my rating. Best of luck to you.

---

> > > ### Author Response · Authors · 2025-11-24
> > >
> > > Thank you -- we are grateful for your constructive and detailed review and for your thoughtful and sincere engagement in the discussion.

---

### Official Review · Reviewer_K1VG · 2025-10-31

**Soundness:** 3
**Presentation:** 2
**Contribution:** 3
**Rating:** 8
**Confidence:** 3

**Summary:**

This paper addresses the challenge of defending against model inversion attacks (MIA). It investigates the relationship between information dependency and privacy, yielding three key findings:
Models that effectively defend against MIA (measured by Attack Acc@1) do not necessarily exhibit reduced information dependency (quantified by HSIC).
Models that memorize training data can still remain robust to MIA (evaluated via L2-Face).
Even models trained on images with 97% of pixels masked can still be reconstructed by MIA (as indicated by Attack Acc@1, @5, and L2-Face).
To interpret these counterintuitive results, the authors introduce the concept of non-robust features—imperceptible perturbations that alter predicted labels. By analyzing robust accuracies under varying perturbation magnitudes, they demonstrate a clear linear relationship between robust accuracy and attack success rate.
Building on this insight, the paper proposes a new training objective, AT-AT, which learns non-robust features to obtain MIA defense.
The paper offers interesting new perspectives, is generally well written (even though this could be further improved, as detailed below), and the experimental evaluation shows some promise (even though some details are unclear).

**Strengths:**

This work challenges the conventional assumption that reducing information dependency between inputs and latent representations inherently improves privacy. Moreover, it identifies a meaningful correlation between privacy leakage and robust accuracy, opening a new perspective and research direction at the intersection of privacy and robustness.
The experimental design is sound, comprehensive, and statistically validated, lending credibility to the reported observations.
The paper compares against a sufficient range of baseline methods, including those focusing on information dependency as well as those emphasizing gradient-based denfense strategies.
The visualizations and plots are clear, well-structured, and intuitive, effectively supporting the main arguments.

**Weaknesses:**

Lack of causal evidence: Section 4 establishes a correlation between privacy and non-robust features, which is well supported by experimental results. However, no causal relationship is demonstrated. Consequently, the claim that “recent MIA defenses obtain their privacy improvements by unintentionally shifting models toward such features” remains insufficiently substantiated.
Missing hyperparameter details: In the proposed AT-AT algorithm, the hyperparameter ε (epsilon)—which controls perturbation magnitude—is neither defined nor discussed. The absence of this detail limits reproducibility and interpretability of the results.
Limited dependency metrics: The paper relies solely on HSIC to measure information dependency. Other established measures, such as CLUB, could have been included to provide a more comprehensive and cross-validated analysis.
Restricted model scope: The experiments are conducted exclusively on ResNet-based architectures, with no evaluation on Transformer-based models. This limits generalizability of the findings to such architectures.

Minor comments:
The structure and organization of the paper could be improved. For example, robust accuracy is frequently referenced in Section 4 and the Introduction, but is not clearly defined, and its relationship to non-robust features is insufficiently discussed. Moreover, there is substantial redundancy between the Introduction and the Experiments section.
Terminological consistency should be improved. The terms information dependency and memorization are used somewhat interchangeably, leading to ambiguity. Choosing a single consistent terminology and providing a clearer definition would enhance readability and conceptual clarity.

Ref: Cheng, Pengyu, et al. “Club: A contrastive log-ratio upper bound of mutual information.” International conference on machine learning. PMLR, 2020.

**Questions:**

First experiment:
Why does training with HSIC regularization result in a higher HSIC value? According to the BiDO paper, HSIC was originally introduced to reduce information dependency, yet this work reports the opposite effect. It would be helpful to clarify this discrepancy. Interestingly, the BiDO paper did not explicitly report post-training HSIC values for comparison.

Second experiment:
How exactly are the labels permuted? If labels are permuted per person (on the images), this would merely rename the classes and should not alter the learned representations. I assume the permutation is performed per image, but this should be explicitly stated.
Additionally, an overfitted model typically learns a direct mapping from input to label rather than meaningful features, effectively reducing the dependency between images and latent representations while increasing the dependency between latent representations and labels. For example, the model can memorize the label of a specific input by memorizing several pixel values. Reporting HSIC values between images and latent spaces for both the permuted and non-permuted models would provide valuable insight.

Third experiment:
In this setting, the available information in the image space is extremely limited, meaning the latent space could potentially encode nearly all pixel-level details. This might explain the model’s vulnerability to inversion attacks. Measuring the HSIC between images and latent spaces in this scenario could help validate this hypothesis.
Furthermore, what datasets are used to train the PPA generative model? If the generative model is trained on the same dataset as the classifier, reconstructing images from only 3% visible pixels may be significantly easier, which would influence the interpretation of the results.
Line 410 and Equation (3):
Why is the objective formulated as a maximization rather than a minimization? In the AT-AT algorithm, ( x + \delta ) corresponds to ( x’ ). Given that the model should predict ( y’ ) for input ( x’ ), it seems that a minimization objective would be more appropriate. Please clarify.

---

> ### Author Response · Authors · 2025-11-21
>
> **We sincerely thank you for your extremely thoughtful and positive assessment, and for highlighting that our paper _"offers interesting new perspectives,"_ _"open\[s\] a new perspective and research direction at the intersection of privacy and robustness"_  with "_experimental design \[that\] is sound, comprehensive, and statistically validated, lending credibility to the reported observations."_ We are also grateful for your positive comments about the clarity of our visualizations, and for suggesting specific places where we can clarify our contribution.**
>
> **Below we address all concerns. We also expand our empirical scope by adding a new set of results with an additional attack and baseline defense suggested by Reviewer *2dtt*:**
> - ***Link to new results plots (Crosshairs denote additions during the rebuttal): <https://imgur.com/a/hgzJTuH>***

---

> ### Author Response · Authors · 2025-11-21
>
> **1. Re: Causality vs. correlation (clarification). Thank you for the opportunity to clarify. Our argument that we provide causal evidence proceeds in four steps:**
>
> - ***(i)***\. Table 1 and the additional experiments in Appendix D.1 show that HSIC and privacy leakage are statistically orthogonal across all existing generic privacy defenses. HSIC explains no significant variance in AttAcc@1, and manipulating BiDO to reduce HSIC (App. D.1) does not increase privacy. Under standard causal inference logic, when a proposed causal driver exhibits no association even under controlled variation, this constitutes _negative causal evidence_ against that variable being a causal mechanism. Thus, our results rule out HSIC reductions as a meaningful causal pathway to privacy.
>
> - ***(ii)***\. We empirically show that existing defenses systematically favor non-robust features. Specifically, our robust accuracy under AutoAttack experiments (Section 4) show that existing defenses heavily favor non-robust features; and
>
> - ***(iii)***\. Non-robust features are strongly _correlated_ with privacy, such that we can predict privacy leakage almost perfectly (R^2 = 0.95) just from knowing the non-robust accuracy. Our paper is careful to state both up front and also in this section that the goal in **Section 4** is correlational, and we make no causal claims here, though it paves the way for the causal inference experiments described in the subsequent sections. Specifically, we begin **Section 4**  (paper lines 300-302) by stating "We show that the defense performance obtained by the class of defenses widely believed to work via information dependency reduction is highly _correlated_ with vulnerability to adversarial examples and non-robust feature bases." We carefully re-conclude the section by re-emphasizing this in bold/italic font (line 387-389) that these regression results **_"provide evidence that non-robust features were unknowingly_ correlated _with privacy." This motivates us to ask (line 389): Can non-robust features_ cause _privacy through deliberate algorithmic design?_**
>
> - ***(iv)***\. AT-AT enables a direct causal test by intervening on the feature basis during training. Section 5 (_'Causal effect of AT-AT loss term on privacy,'_ line 425) explicitly formulates this as a randomized controlled experiment, where the treatment is rewarding non-robust features in the loss. Holding all other factors fixed, this intervention **causally** reduces PPA AttAcc@1 from 84% to 6.5% (a 77× reduction in leakage odds, z = 38.0, p < 10⁻¹⁶). This identifies non-robust feature emphasis as a causal mechanism driving privacy improvements.
>
>
> **So in sum, the argument is:**
>
> - **(i) Reducing HSIC doesn't cause privacy as widely believed (shown via negative causal evidence);**
>
> - **(ii) Existing defenses heavily favor non-robust features;**
>
> - **(iii) Non-robust features are highly correlated with privacy;**
>
> - **(iv) Increasing non-robust features _causes_ privacy to increase (tested via a formal randomized controlled experiment).**
>
>  Note that we provide a **full formalization of the causal inference framework in Appendix H**, where we are careful to re-specify the correlation vs. causation experiments (see paper 1658-61) and also report our causal identification strategy, causal estimation, and causal inference, as well as full results tables in that appendix.

---

> ### Author Response · Authors · 2025-11-21
>
> **2\. Re: Are there missing ε hyperparameter details, and does this limit reproducibility of the results?**
> - All ε values (i.e. $ℓ_∞$ perturbation radii) were reported in Table 6, line 1020, on p. 20 in the Appendix Section B.: "AT-AT Hyperparameters and Training." We'll add a $\backslash$ref{...} to this section in the paper body of the camera-ready manuscript.  We hope and expect that this work will be widely replicated, so we also provided extensive high-quality pushbutton replication codes and complete replication details.

---

> > ### Author Response · Authors · 2025-11-21
> >
> > **3\. Re: Dependency metric (HSIC) vs. alternatives**
> > - We focused on HSIC because it is the specific metric chosen by a SOTA generic privacy defense (BiDO) in its loss function, the main metric discussed in the literature, and also because it can be computed for a generic model (unlike, e.g., MID's mutual information approximation formula, which is also discussed in the MIA literature).
> > -  By focusing on HSIC, we were able to *measure the hypothesized drivers of privacy on their own terms* and reveal the surprising result that **even HSIC-penalizing defenses do not reduce HSIC**. We agree that incorporating CLUB or related tractable MI proxies is a valuable direction for future work, particularly because our results show that HSIC, despite being the dominant choice in existing defenses, does not align with privacy. We hope and expect that our paper will spark much discussion and exploration of these topics.

---

> > > ### Author Response · Authors · 2025-11-21
> > >
> > > **4\. Re: Is the term "robust accuracy" undefined?**
> > > - Robust accuracy is defined in lines 309-314 of our paper right before we introduce Table 4 of robust accuracy results: "We report _robust accuracy_, i.e., worst-case accuracy on adversarial examples across all four attacks in the standard AutoAttack ensemble (Croce & Hein, 2020b)" Appendix F (lines 1404 - 1592) also provide full background, literature review, and additional results on robust accuracy. To improve clarity, we'll also add the definition to the introduction where we first refer to this term in the camera-ready manuscript.

---

> > > > ### Author Response · Authors · 2025-11-21
> > > >
> > > > **5.  Experiment-specific clarifications**
> > > >
> > > > - **Re: Why doesn't HSIC regularization decrease HSIC in BiDO?**
> > > >   - We discuss the reasons for this surprising result in Appendix D.1 (paper lines 1193 - 1207). As you note, the authors of BiDO do not report HSIC. In short, BiDO in practice fails to decreases HISC between data and model parameters because the BiDO loss function adds 2 terms to the loss function: HSIC penalty between data and model parameters (denoted by $λ_x)$, and HSIC _reward_ between model parameters and label (denoted by $λ_y$). In practice, under all default parameters used by the BiDO authors, the second term dominates, rendering the first of these terms to be negligible. This is why, in our appendix HSIC experiments (App. D.1), we are able to use BiDO to _reduce_ HSIC between data and model parameters---to do so, we zero out the second term such that the first term is not dominated (for how this relates to the ablation study of the original BiDO paper, see App. D.1). Interestingly, this works to significantly lower HSIC from 73.68 to 2.16, but reducing HSIC here still fails to improve privacy (in fact, the privacy becomes _worse_). Again, we conclude that intervening to reduce HSIC does not cause privacy.
> > > >
> > > > - **Re: Permuted labels: how are they applied?**
> > > >   - In line with your intuition, we permute labels **per image**, not per person, thus destroying the correlation between *X* and *y*.
> > > >
> > > > - **Re: Can a permutation model become accurate by memorizing just a few pixels of each image?**
> > > >   - No. Importantly, as with all our trained models, we use extensive data augmentation as is standard in the MIA literature (See data augmentation in App A.2, and permutation model training details A.3.1). Augmentations include: _Random resized cropping_, and _Random horizontal flipping_ as the geometric augmentations, and _Brightness, Contrast, Saturation,_ and _Hue,_ as the photometric augmentations. Together, these augmentations mean that the permutation model cannot become accurate just by memorizing several pixel values that map to each label, as such pixels change under augmentation.
> > > >
> > > > - **Re: minimization sign in objective**
> > > >   - Thank you very much for catching this 'max' $\rightarrow$ 'min' typo. It is a leftover from an early previous formulation that used untargeted PGD. Untargeted PGD has a maximization objective instead of a minimization one. We'll fix this in the camera-ready manuscript.
> > > >
> > > > - **Re: Do attacks use GANs trained on the same datasets we use in experiments?**
> > > >   - The pre-trained generative model for PPA and IF-GMI is StyleGAN2 which is trained on Flickr-Faces-HQ Dataset (FFHQ). That FFHQ dataset has no overlap of identities with the datasets in our experiments. FFHQ also has a significant distributional shift compared to the face datasets we use in our experiments, so it is all the more impressive (and alarming) that we can obtain devastating reconstructions of >97% censored images for our datasets.
> > > >
> > > > **We sincerely thank you again for your very constructive and thorough review.**

---

> > > > > ### Comment · Reviewer_K1VG · 2025-11-28
> > > > >
> > > > > Thank you for the nice and detailed clarification of my concerns. In particular, the most important concern regarding causality is nicely explained, and I hope this can be used to improve the final version of this paper.
> > > > > I will update my review scores accordingly.
> > > > > I believe this paper makes interesting contributions.

---

### Official Review · Reviewer_2dtt · 2025-11-01

**Soundness:** 3
**Presentation:** 2
**Contribution:** 3
**Rating:** 4
**Confidence:** 5

**Summary:**

This paper challenges the prevailing paradigm that "reducing information dependency mitigates training data leakage via Model Inversion Attacks (MIAs)" through three counterintuitive experiments. To explain these findings, the authors identify a novel privacy-adversarial robustness tradeoff. Based on these findings, they propose Anti Adversarial Training (AT-AT), a defense that proactively learns non-robust features by reversing standard adversarial training. Experimental results have validated the effectiveness of the proposed method.

**Strengths:**

- Good performance: the proposed method outperforms baselines in most metrics
- Generalizability: results hold across datasets, architectures, and attacks, demonstrating the tradeoff’s universality and AT-AT’s adaptability.
- Practical Relevance: AT-AT’s tunable λ addresses a key limitation of SOTA defenses (fixed privacy-robustness tradeoffs) and works for deployed models without modifications on the training/generation process.

**Weaknesses:**

- This work only focuses on the white-box attacks. It is recommended that the author conducts more evaluation on the black-box attacks and label-only attacks to further validate the correlation between adversarial robustness and privacy.
- The compared baselines are not sufficient. Attacks like PLGMI [1] and PPDG [2] and defenses like Trap-MID [3] are expected to compare.
- If a model is fine-tuned post-deployment (shifting its non-robust features), the author does not evaluate whether AT-AT’s privacy guarantees persist or require retraining.


[1] Yuan X, Chen K, Zhang J, et al. Pseudo label-guided model inversion attack via conditional generative adversarial network[C]//Proceedings of the AAAI Conference on Artificial Intelligence. 2023, 37(3): 3349-3357.

[2] Peng X, Han B, Liu F, et al. Pseudo-private data guided model inversion attacks[J]. Advances in Neural Information Processing Systems, 2024, 37: 33338-33375.

[3] Liu Z T, Chen S T. Trap-MID: Trapdoor-based Defense against Model Inversion Attacks[J]. Advances in Neural Information Processing Systems, 2024, 37: 88486-88526.

**Questions:**

Refer to Weaknesses.

---

> ### Author Response · Authors · 2025-11-21
>
> **We thank you for highlighting the paper's "good performance," "generalizability" across datasets/architectures/attacks, and "practical relevance."**
>
> We also thank you for suggesting the **additional experiments - we include them below**, and we agree that they provide strong evidence of AT-AT's contribution.
>
> **We strongly feel that this is a high-impact paper. We believe your remaining reservations and reasons for score 4 instead of "full accept" can be addressed by clarifying two key aspects below.**
>
> **We ask that you consider improving your score accordingly.**
>
>
>
> -  **Reviewer's Suggested Additional Experiments:**
>
>    ***Re: the Reviewer's comment "The compared baselines are not sufficient.  Attacks like PLGMI \[1\] and PPDG \[2\] and defenses like Trap-MID \[3\] are expected to compare.*"**
>
>     - **Thank you -- we expanded our experiments for the rebuttal:**
>       - **PPDG:**  We add this recent attack, and we are grateful for suggesting this way to highlight our contribution, as AT-AT continues to continues to outperform all baselines under PPDG;
>
>       - **Trap-MID:** we add this recent defense, though it is not a generic privacy defense (which is our main focus), as it attempts to mislead the specific style of MIA (similar to RoLSS) rather than obtain privacy in general, yielding well-known performance stochasticity. Nevertheless, we are grateful for this suggestion: it is **particularly compelling** that these new experiments now demonstrate that AT-AT, a generic privacy defense, outperforms the new Trap-MID defense that is deliberately designed to thwart the attacks we consider.
>
>       - **PLGMI:** PLGMI requires repeated, from-scratch training of a separate custom GAN for each baseline, which is computationally infeasible in our hi-res setting, so we are not aware of published hi-res research that uses it.
>
>     - **LINK to new results plots (Crosshairs denote additions during the rebuttal): <https://imgur.com/a/hgzJTuH>**
>
> ***We address the other aspects of the review in the comment below.***

---

> ### Author Response · Authors · 2025-11-21
>
> **Reviewer's two key clarifications:**
>
> **_1\. Re: the Reviewer's comment "This work only focuses on the white-box reconstruction attacks" and not other types of privacy attacks such as label attacks._**
>
>
>
> - **White-box attacks are widely accepted in the literature as the correct setting to lower-bound worst-case reconstruction.**
>    - AT-AT's novelty is not that we seek a practical defense that works in various different domains of privacy research (white-box attacks, label attacks e.g. differential privacy, etc.). **Instead, AT-AT's novelty is that it provides the first training regime that allows us to causally test whether non-robust features cause privacy against hi-res reconstruction under MIA.** The **core contribution of the paper** is the finding that non-robust features cause privacy that prevents MIA reconstructions, whereas we show that the prevailing view that information dependency drives privacy is incorrect, despite widespread belief to the contrary.
>
> - **To test a causal mechanism for _generic worst-case reconstruction_ _privacy_, the correct methodology is to evaluate against the strongest possible attacker.** Thus, as is standard in empirical privacy work, we 'give the attacker every possible advantage' to obtain a **lower bound on worst-case leakage** rather than benchmark individual attack families and the particularities of individual attacks. White-box reconstruction attacks are universally regarded as the **most powerful** reconstruction-style MIAs.
>
> - **Evaluating weaker settings (e.g., black-box attacks) would not meaningfully tighten the lower bound on worst-case reconstruction leakage and would not change our causal conclusions.** For this reason, such evaluations are out of scope for the scientific question we study here.
>
>
>  -  **Note that the MIA defense literature widely agrees---to quote other papers:**
>     - Ho et al., CVPR'24 (TL-DMI paper), p. 3 states: "*This study primarily focuses on **whitebox** MI attacks, which are the most dangerous, and can achieve impressive attack accuracy since the adversary has complete access to the target model.*"
>     - Peng et al., KDD'22, (BiDO paper) p. 5 states: "*we present the experimental evaluation for verifying the efficacy of BiDO against different attacks in the **white-box** setting. Note that, here we **do not focus on the black-box** attack methods which are relatively easy to be defended.*"
>
> **_2\. Re: If a model is fine-tuned post-deployment, the author does not evaluate whether AT-AT's privacy guarantees persist or require retraining._**
>
> - **We are not aware of any MIA defense paper that studies post-deployment fine-tuning followed by re-evaluation of privacy. Can you clarify the specific threat model you're describing?  AT-AT is a generic privacy training regime, so it is possible to run further iterations of AT-AT's training loop (to fine-tune), and doing so does not reduce privacy or cause semantically meaningful features of the pretrained data to "reappear."** More specifically, a key contribution of AT-AT's training regime ensures that such robust, visually meaningful features are **never learned in the first place**. This contrasts with gradient-suppressing defenses such as Trap-MID, which _do_ learn semantically meaningful features and attempt to mask them by adding "trapdoors." Such trapdoors could in theory be unlearned through fine-tuning, potentially exposing the underlying robust features again.
>
> -  AT-AT, by contrast, trains the model to rely on **non-robust features** that are generalizable, visually imperceptible, and semantically meaningless to humans. Our finding is that these properties make non-robust features ideal for privacy-preserving learning: the model achieves high accuracy **without ever encoding visually meaningful information that is vulnerable to reconstruction.**
>
> **We hope this clarifies your remaining reservations and provides the additional experiments you suggested, and we would appreciate it if you would consider improving your score accordingly.**

---

### Official Review · Reviewer_APiD · 2025-11-01

**Soundness:** 3
**Presentation:** 2
**Contribution:** 3
**Rating:** 6
**Confidence:** 3

**Summary:**

The paper argues that training-data leakage is driven by adversarially robust features instead of information dependency, and it supports this claim with experiments that falsify the dependency hypothesis. The authors therefore propose Anti-Adversarial Training (AT-AT) which intentionally shifts learning toward non-robust features to reduce leakage, since these are hard for MIAs to turn into clear pictures.

**Strengths:**

Strong MIA defense, AttAcc@1 fell from ~84% to ~6.5% (Tab. 21). On RN-152 accuracy vs. privacy (Fig. 3), AT-AT outperforms baseline methods across datasets and attacks.


The paper combines causal reframing, quantified trade-off, and practical SOTA method. The authors clearly test and challenge a popular theory using simple, targeted experiments, and turn the insights into AT-AT that lowers SOTA inversion success while keeping accuracy high.

The paper is generally well written, with a decent amount of details provided in the appendix.

**Weaknesses:**

Non-robust cues may still be highly predictive when added up and even sufficient for generalization. That means an adaptive MIA that directly optimizes the target’s representations could still extract identity/membership information. The paper does not systematically evaluate such features or prior-agnostic inversion attacks, so its privacy gains may depend only on the current setup.


Narrow empirical scope. The paper lacks tests with broader attackers, other modalities, and offers no formal guarantees.

**Questions:**

Does AT-AT trade privacy gains for interpretability? The proposed method might help privacy, but it may suffer in terms of interpretability compared to robustly trained models. If the model leans on tiny, high-frequency cues spread across many pixels (rather than dog’s head or a face), attribution maps may become visually unintuitive.

---

> ### Author Response · Authors · 2025-11-21
>
> **We sincerely thank the reviewer for the thoughtful and constructive review, and for highlighting that our paper "*combines causal reframing, quantified trade-off, and a practical SOTA method***," ***clearly test\[s\] and challenge\[s\] a popular theory,"*** ***and achieves*** "***strong MIA defense.***" **We also appreciate your recognition of the paper's integration of conceptual reframing with a concrete defense.**
>
> **Your remaining reservations and the reason for a score of 6 instead of "full accept" are due to questions regarding:**
>
> - **1. Paper contents:**
>      (i) the empirical scope,
>      (ii) the lack of formal guarantees.
> - **2. Conceptual contribution:**
>      (iii) whether non-robust cues could still be exploitable by more adaptive or prior-agnostic MIAs,
>      (iv) whether AT-AT trades privacy for interpretability.
>
> **Below we clarify each of these points and would be grateful if you would consider updating your score accordingly.**

---

> > ### Author Response · Authors · 2025-11-21
> >
> > **1\. Paper Contents:**
> >
> > - **(i) Empirical scope**
> >
> > As noted in the top rebuttal above, our empirical scope is already significantly more extensive than prior work, and considerably broader than MID, BiDO, TL-DMI, or NegLS.
> >
> > Nevertheless, we agree that additional tests strengthen the paper. For the rebuttal, we expanded the empirical scope with a new attack (PPDG) and a new defense (Trap-MID). This brings the total to a field-leading: &nbsp;&nbsp;**9 baselines (incl. NoDef), 3 attacks, 2 architectures, 3 datasets, 4 adversarial attacks to test robustness, plus 3 complete motivating experiments.**
> >
> > All new experiments corroborate our main findings and strengthen the empirical case for AT-AT.
> >
> > - **(ii) "Lack of formal guarantees"**
> >
> > **We are surprised by this comment.** No practically successful MIA defense offers formal guarantees. However, our paper **does contribute theoretically:** it provides one of the only **formal counterexamples** to the core assumption underlying a major recent stream of provable-privacy frameworks.
> >
> > Specifically, we show that even when >97% of training pixels enjoy **_perfect_** unbiased-estimator privacy guarantees (via Hammersley-Chapman-Robbins bounds), modern MIAs (which rely on prior-driven estimators such as GANs) can still produce **devastating reconstructions**. This is motivating experiment #3 ("unseen pixels," starting at line 249, and full details in App. E from line 1235).
> >
> > Thus, recent provable-privacy guarantees, while mathematically elegant, are **security-irrelevant under practical attacks like PPA or IF-GMI**. Identifying this fundamental practical flaw is arguably a more significant theoretic contribution than providing an additional guarantee that still fails under real-world MIAs.

---

> > > ### Author Response · Authors · 2025-11-21
> > >
> > > **2\. Conceptual Contribution**
> > >
> > > Your remaining concerns are:
> > >
> > > - **(iii) Could non-robust cues still be exploitable by more adaptive or prior-agnostic MIAs?**
> > >
> > > - **(iv) Does AT-AT trade privacy for interpretability?**
> > >
> > > **A key clarification resolves both:**
> > >
> > > **Our core contribution is _not_ that AT-AT introduces a new non-robust-feature mechanism.**
> > >
> > > It is that AT-AT reveals that _all_ mainstream generic MIA defenses already rely on the same mechanism, which trades robustness (and as a consequence, interpretability-see, e.g., Engstrom et al. 2019, full citation in our paper) for privacy. Specifically, our core result is that contrary to widespread belief, MID, BiDO, and TL-DMI achieve privacy **not** by reducing information dependency, but by unintentionally shifting models toward non-robust features.
> > >
> > > **AT-AT's novelty is that it allows us to causally test whether non-robust features cause privacy, which is the core contribution.**
> > >
> > > **As such, AT-AT is the instrument that allows us to:**
> > >
> > > - Expose and measure the privacy-robustness frontier,
> > > - Makes this non-robustness -> privacy mechanism that is hidden in all generic privacy defenses **explicit**,
> > > - and for the first time allow practitioners to **control** both
> > >     (a) the balance between robust and non-robust features (λ), and
> > >     (b) the magnitude of the non-robust perturbation (ε).
> > >
> > >
> > >
> > > Naturally, larger λ or smaller ε yields **more privacy but less semantic interpretability**.  AT-AT is simply the first method to make this tradeoff that is inherent in all generic defenses visible, tunable, and rigorously (causally) measurable.
> > >
> > > The fact that this applies also to previous defenses is well-evidenced in the privacy-robustness relationship experiments (Fig. 2, Table 5), which show that:
> > >
> > > - Existing defenses heavily favor non-robust features;
> > > - Non-robust features are highly correlated with privacy;
> > > - Increasing non-robust features _causes_ privacy to increase (tested via a formal randomized controlled experiment);
> > > - Meaning that every effective generic MIA defense (not just AT-AT) depends almost entirely on **non-robust features**, even when motivated by "information dependency reduction."
> > >
> > > Thus the question "could non-robust cues still be exploited?" applies **equally** to all generic defenses, not to AT-AT in particular.
> > >
> > > The same applies to interpretability. Interpretability loss is **intrinsic to the privacy-robustness frontier**, not something introduced by AT-AT. AT-AT is just a novel instrument that allows us to rigorously reveal a tradeoff that is common to all generic defenses.
> > >
> > > Our manuscript states:
> > >
> > > - **Lines 121-126:** "MIA defenses unintentionally make a tradeoff… by shifting towards non-robust features."
> > > - **Lines 141-144:** AT-AT's effect is _tunable_, unlike baseline defenses.
> > > - **Lines 155-157:** Robust representations may be **inherently privacy-vulnerable**.
> > >
> > > In short, we do not claim that AT-AT creates a new mechanism. We show that AT-AT is the first method to reveal, quantify, and control the mechanism that _already governs all generic privacy defenses._
> > >
> > > **Final note.** We appreciate your thoughtful review. We hope this clarifies that:
> > >
> > > - The empirical scope is broad and has now been expanded further;
> > > - The paper **_does_** make a significant theoretic contribution by identifying a fundamental flaw in widely believed privacy-guarantee frameworks;
> > > - Concerns about adaptive MIAs or interpretability apply inherently to **all** generic defenses;
> > > - AT-AT is the first method that gives practitioners control over this unavoidable tradeoff;
> > > - **As such, its core contribution is that it allows us to causally test that non-robust features cause privacy, which is our core result and most impactful finding.**
> > >
> > >
> > >
> > > **We strongly feel that this is a high impact paper. We would gratefully appreciate your consideration of an updated score.**

---

### Author Response · Authors · 2025-11-21

**We thank all reviewers for their thoughtful, constructive, and detailed reviews, and for their comments that the paper delivers an:**

-  **"*Excellent and fundamental contribution*" and “*fundamental, paradigm-shifting insight*” [Reviewer *1QzK*]; that**

-   **"*Clearly test[s] and challenge[s] a popular theory*" [Reviewer *APiD*]; and**

-   **"*Open[s] a new perspective and research direction at the intersection of privacy and robustness*" [Reviewer *K1VG*].**


**We also thank reviewers for their many positive comments that the paper provides "*exceptionally well-designed and counter-intuitive experiments*" [reviewer *1QzK*], with "*experimental design [that] is sound, comprehensive, and statistically validated*" [reviewer *K1VG*].**

**Collectively, the reviews strongly affirm that our paper identifies and rigorously validates a previously unrecognized but foundational mechanism by which robust features drive privacy leakage, establishing a new robustness-privacy frontier for future research.**

**We believe we address any remaining reservations in our individual responses under each review below. We would appreciate it if reviewers would consider updating their scores accordingly.**

**We also include a complete new set of results with an additional recent attack and a new baseline defense suggested by Reviewer *2dtt*:**

-   **PPDG** (a very recent attack);

-   **Trap-MID** (a new defense that does not promise generic privacy, but rather attempts to block gradients used in some recent attacks).



**Link to new PPDG and Trap-MID results plots: (*\*Crosshairs denote additions during the rebuttal*): https://imgur.com/a/hgzJTuH**


**This brings the total to a field-leading: &nbsp;&nbsp;   9** baselines (incl. NoDef), **3** attacks, **2** architectures, **3** datasets, **4** adversarial attacks to test robustness, **plus 3** complete motivating experiments.

**These extensive experiments corroborate our main findings and strengthen the extensive empirical evidence.**

---

### Comment · Area_Chair_qZyq · 2025-11-25

Dear Reviewers,

Thank you for your time and effort in reviewing submissions for ICLR 2026. As we begin the author-reviewer discussion process, we kindly remind you to submit your responses to the author rebuttals by **December 2**.

Your engagement in this discussion phase is crucial to ensuring a fair and thorough evaluation of each submission.

### **Action Required**
- Carefully consider the authors’ rebuttal and any additional evidence they provide.
- Update your review (if applicable) to reflect your revised perspective.
- Discuss with the authors if further details are required

Your AC

---

### Author Response · Authors · 2025-11-30

## **Author Note for New AC (re: Score Reversion Due to OpenReview Bug)**

**Dear New AC, please note that two reviewers (K1VG and 1QzK) submitted significant post-rebuttal updates before the OpenReview reversion; both increased their scores significantly based on the rebuttal and new experiments.**

**These two updates increased the mean score to&nbsp; 7.25 or 7.50  (top 0.54% or 0.29% of submissions, respectively, per CoPilot), with the remaining two reviewers not yet having responded during the discussion phase.**

Below is a concise summary of the actual post-rebuttal reviewer positions, with **all statements verifiable in the discussion thread (and screenshots available upon request):**

---

**Reviewer #1 – APiD:**
• **Pre-rebuttal score:** 6
• **Post-rebuttal response:** *Had not yet responded.*

**Reviewer #2 – 2dtt:**
• **Pre-rebuttal score:** 4
• **Post-rebuttal response:** *Had not yet responded.*

**Reviewer #3 – K1VG:**
• **Pre-rebuttal score:** 8
• **Post-rebuttal change:** **_Explicitly stated they increased their rating._**
  – We did not see whether it was **8→9** or **8→10**, as the ICLR security-bug email was sent right after K1VG commented below, quote:   ***“I will update my review scores accordingly. I believe this paper makes interesting contributions.”***

**Reviewer #4 – 1QzK:**
• **Pre-rebuttal score:** 6
• **Post-rebuttal change:** **_Explicit increase to 10._**
  – Timestamped screenshots of **6→10** available.
  – *1QzK's* quote below: ***“I will increase my rating”*** (followed by confirmation).

---

**\*\*\*Score Summary Before and After Rebuttal (prior to reversion):**
• **Pre-rebuttal mean:** 6.0
• **Post-rebuttal mean:**
  – **7.25** if Reviewer K1VG increased **8→9**
  – **7.50** if Reviewer K1VG increased **8→10**
    (We **did not see which of these increases occurred** due to the timing of the ICLR score reversion.)

Only two reviewers had responded before the security incident, and **both increased their scores to the very top of the scale**, as documented in the discussion threads below.

---

### Meta-Review · Area_Chair_AnmH · 2025-12-26

**Summary:**

APiD: 1) whether non-robust cues could still be exploitable by more adaptive or prior-agnostic MIAs ; 2) whether AT-AT trades privacy for interpretability; 3) narrow empirical scope

2dtt: 1) only focus on white-box attack (need more evaluation on black-box and label-only attacks; 2) baselines are not sufficient; 3) a model is fine-tuned post-deployment (shifting its non-robust features). Need evaluations on whether AT-AT’s privacy guarantees persist or require retraining

K1VG: Lack of causal evidence

1QzK: 1) the AT-AT framework undermines model interpretability and. trustworthiness; 2) potential vulnerability to adversarial attacks

**Reviewer Concerns:**

Most of reviewers' concerns are reasonably addressed. Reviewers K1VG and 1QzK have explicitly acknowledged this and agreed with raise their scores.

**Reviewer Scores:**

APiD & 2dtt: these two reviewers did not explicitly respond. But it is likely that they will increase the scores (or at least maintain their original scores)

K1VG: the reviewer is likely to increate the score  (K1VG explicitly mentioned increasing the score)

1QzK: the reviewer is likely to increase the score (1QzK explicitly mentioned increasing the score)

---

### Decision · Program_Chairs · 2026-01-26

Accept (Poster)